# *Clusia* genomes shed light on the evolution and diversity of crassulacean acid metabolism physiotypes

Hannes M. Kramml [1,14], Johannes B. Herpell [1,2,14], Clara Priemer [1,3], Zoe Wessely [1], Florian Schindler [1], Andreas Berger [4], Maximilian Kellner[1], Stefan Plott [1], Ágnes Dohovits[1], Tamara Schmidt[1], Peter Kerpan [5], Leila Afjehi-Sadat [6], Palak Chaturvedi [1], Arindam Ghatak [1,7], Martin Brenner[1], Iro Pierides[1], Lena Fragner[1], Eva M. Temsch [5], Fabio Trevisan[1], Menriti Ibrahim[1], Felix Fromwald[1], Anke Bellaire[5], Oleg Simakov [8], Werner Huber[5], Ulrich Lüttge[9], Ovidiu Paun [5], Susann Wicke [10], Hanna Weiss-Schneeweiss [5], Gert Bachmann[1] & Wolfram Weckwerth [1,7,11,12,13] ✉

More than 200 years ago, Alexander von Humboldt described a tree of the genus *Clusia* for its ability to perform crassulacean acid metabolism (CAM). This drought-adaptive metabolism allows plants to maintain photosynthesis under water limitation by temporally separating $CO_2$ uptake and fixation. The diversity of CAM physiotypes has fueled a debate about evolutionary constraints and the feasibility of engineering CAM into $C_3$ crops. The genus *Clusia* displays an exceptional diversity of photosynthetic physiotypes, yet genome sequences and genomic mechanisms generating this diversity remain unresolved. Here, we sequence and compare the genomes of three *Clusia* species spanning weak, inducible, and strong CAM. We show that polyploidization followed by transposon-mediated genic diploidization could have shaped CAM-related gene families, particularly those controlling phosphoenol-pyruvate recycling via phosphorolytic leaf starch metabolism. Our results indicate that whole-genome duplication coupled to diploidization might have driven diversification of CAM physiotypes in *Clusia*, providing a genomic framework for understanding CAM diversity and evolution.

In 1800, the famous German naturalist Alexander von Humboldt became one of the first observers of a phenomenon later known as crassulacean acid metabolism (CAM). He submerged the leaf of a tropical tree in water and noted its unusual behavior: There was no release of air bubbles despite its exposure to sunlight[1]. The leaf he was studying belonged to *Clusia rosea* Jacq. (Clusiaceae), a neotropical tree with the ability to perform CAM, a carbon concentrating mechanism (CCM) and physiological adaptation to low water or $CO_2$ availability in plants[2]. CAM plants open their stomata for gas exchange during the night but close them during the day. In this manner, diurnal transpiration is reduced to a bare minimum resulting in a very high water use efficiency (WUE)[3].

In light of global warming and the concomitant aridification of many regions used for food production, there is considerable interest in the prospects for bioengineering CAM into $C_3$ plants[4]. It is argued that CAM does not necessarily require specialized cell architecture or de novo evolution of enzymes or transporters[5] and has evolved independently from $C_3$ plants in at least 38 vascular plant families[6]. Species

in the genus *Clusia* are, however, the only genuine dicotyledonous trees known to perform CAM, which is associated with anatomical adaptations[7]. What is, nevertheless, necessary for CAM is a rewiring of central carbohydrate storage pathways and buffer capacities[8]. This is thought to have occurred in a stepwise iterative fashion, through the regulatory evolution of pre-existing genes and not through the acquisition of neofunctionalized genes following whole-genome or tandem gene duplication[9,10]. Although the details regarding the evolution from C₃ to CAM are elusive, investigating different physiotypes of CAM in close relatives can help gather snapshots of different evolutionary stages from C₃ to CAM but also vice versa. In *Clusia*, a large variety of photosynthetic physiotypes have been described[1–3,7,8,11–18], yet comprehensive and high quality genomic, transcriptomic, metabolomic, and proteomic data are unavailable.

Here, we fill this knowledge gap and use our data to reconstruct the evolutionary origins of subtypes of CAM in this particularly physiologically diverse plant genus. We illustrate that whole-genome duplication (WGD) and subsequent genic diploidization could have played a role in pseudogenization in specific pathways explaining the observed *Clusia* CAM physiotypes. Our findings modernize our understanding of the evolution of CAM physiotypic diversity, the role of polyploidy in its, perhaps recurrent, origin, and illuminate what may be necessary for CAM engineering.

## Results and discussion
### Integrating physiology, morphology, and markers to resolve taxonomic ambiguities

Extensive research throughout the 20th century has established that many *Clusia* species possess an inherent genetic capacity for CAM[19]. To evaluate this genus using contemporary molecular tools, we selected species from the Lüttge collection exhibiting contrasting physiological phenotypes. Over the years, we have utilized this collection extensively to compile a dataset of physiological traits. Based on these data, we chose three representative specimens to cover genomes of a broad range of different photosynthetic physiotypes: *C. rosea*, which constitutively performs CAM[7,15,16]; *C. minor* L. s.l., capable of facultative CAM under specific environmental conditions[7,12,13,17]; and *C. major* L., consistently exhibiting weak facultative CAM like behaviour. This selection aligned well with existing physiological data. To unambiguously clarify species identities, considering the notoriously challenging taxonomy of *Clusia* species, we conducted molecular and morphological analyses of relevant specimens and deposited nuclear marker sequences and digital herbarium vouchers (Fig. 1a, Supplementary Table 1). These analyses reveal a recurring pattern of erroneous species determinations that have confounded otherwise valid physiological observations in previous studies[20] (see Supplementary Discussion 1). In this context, we conducted controlled-environment physiological phenotyping and provide internally consistent validation of species physiotypes. Gas exchange measurements conducted under drought conditions show C₃-like daytime carbon assimilation in *C. major* (Fig. 1b). The lack of nocturnal CO₂ efflux, however, suggests a weak CAM phenotype where respired CO₂ is recycled at night via the CAM cycle. Complementary titratable acidity measurements, performed over a full 24-h cycle under identical environmental conditions and either well-watered or drought treatment (Supplementary Fig. 1), demonstrate that *C. major* exhibits no nocturnal acidification when well-watered, but significant overnight acid accumulation under drought, confirming CAM induction (Fig. 1c). In contrast, *C. rosea* shows strong nocturnal acidification under both watering regimes.

Together, taxonomic and molecular identification, gas exchange, and titratable acidity measurements provide a coherent framework supporting our physiotype classifications (Supplementary Discussion 1). To address these issues and promote clarity in future work, here we provide reference genomes of *Clusia*, establishing a new framework amid persistent taxonomic uncertainty.

### *Clusia* genomes vary in size, chromosome number, and ploidy level

In order to investigate the genomic makeup of CAM physiotypes we determined the karyotypes and genome sizes (1 C value) of our selectees: *C. major* possesses 2n = 60 chromosomes (1.68 pg/1 C), *C. minor* 2n = 90 (3.09 pg/1 C), and *C. rosea* 2n = 120 (3.22 pg/1 C). The chromosome numbers unambiguously show that the genomes differ in their levels of ploidy (Supplementary Fig. 2). The genome size of *C. minor*, however, is not proportional to chromosome numbers (Supplementary Table 2) and varying estimates of C values were reported in literature[21], likely due to taxonomic ambiguities. We would only expect a linear correlation if the event leading to polyploidy was recent and if the species are in a parent-descendant relation. Even then, mechanisms such as genome downsizing could prevent proportionality[22].

The three *Clusia* genomes were sequenced on a PacBio Sequel II system resulting in a raw read coverage of 226 x, 120 x, and 49 x for *C. major*, *C. minor*, and *C. rosea*, respectively (Supplementary Data 1). With up to 18 million reads per plant and an average read length of up to 22 Kb, read lengths were sufficient to span large repetitive regions of the genome. Our first draft assemblies via FALCON[23] spanned 1.76 Gb (*C. major*), 3.40 Gb (*C. minor*), and 3.29 Gb (*C. rosea*) with contig N50s of 1.12 Mb, 580 Kb, and 364 Kb, respectively (Supplementary Data 2). Using BUSCO[24] with the lineage dataset eudicots (odb10) we estimated the completeness of our assemblies: All draft assemblies were well covered with values ranging from 94.7% to 95.1% of BUSCOs present, with high percentages of duplicated BUSCOs only in *C. minor* and *C. rosea* (Supplementary Fig. 3a).

In order to contextualize the differences in chromosome numbers of the three species we assessed the ploidy of our genomes using k-mer frequency spectra[25] of error corrected long read data[26]. Although the meiotic behavior of the chromosomes remains unknown, all species show diploidized genomes of polyploid origin, as the estimated levels of heterozygosity and allele topology suggest (Supplementary Fig. 3b–d). To illustrate: *C. major* reads display a high frequency of the tetraploid k-mer AABB, which is only matched by the most abundant diploid k-mer AB, indicating cytological diploidization. The preferential pairing of A with A and B with B—rather than mixed AAAB groupings—could either result from the existence of parental (sub)genomes (homoeologs) that were already different before hybridization occurred, or from ongoing cytological diploidization (i.e., re-establishment of bivalent pairing and thus fertility)[27].

Overall, karyotype and sequencing analyses suggest that *C. major* represents a diploidized tetraploid, here treated as functional pseudo-diploid. *C. minor* seems to be a diploidized hexaploid, hereafter referred to as pseudo-triploid, and *C. rosea* may represent a diploidized octo-ploid, referred to as pseudo-tetraploid (Supplementary Fig. 3).

### The *C. major* genome serves as a reference

We sought to obtain a chromosome level assembly to perform proper genomic analyses. We used the least complex and most highly covered genome, namely that of *C. major*, for Hi-C chromatin conformation capture. After reassigning highly heterozygous sequences and phasing pseudo-haplotypes[28,29], we nearly halved our draft assembly to 1989 primary contigs, of which 99.5% could be anchored to 30 linkage groups[30] (Supplementary Data 3). Position and orientation of scaffolds was manually curated, thereby breaking and correcting more than 200 chimeric scaffolds[31,32]. Finally, we obtained a haploid chromosome level assembly with an N50 of 53 Mb and almost 97% complete BUSCOs, comprising 29 of the 30 chromosomes and three small bins of unanchored contigs/scaffolds (Supplementary Data 4). The *C. major* reference genome assembly adds up to a total length of 1.5 Gb (Fig. 2).

Through a combination of de novo and genome-guided transcriptome assembly, protein alignments, ab initio gene model prediction, repeat identification and masking, and thorough removal of genic fragments we identified 35,612 gene models and 23,194 alleles in

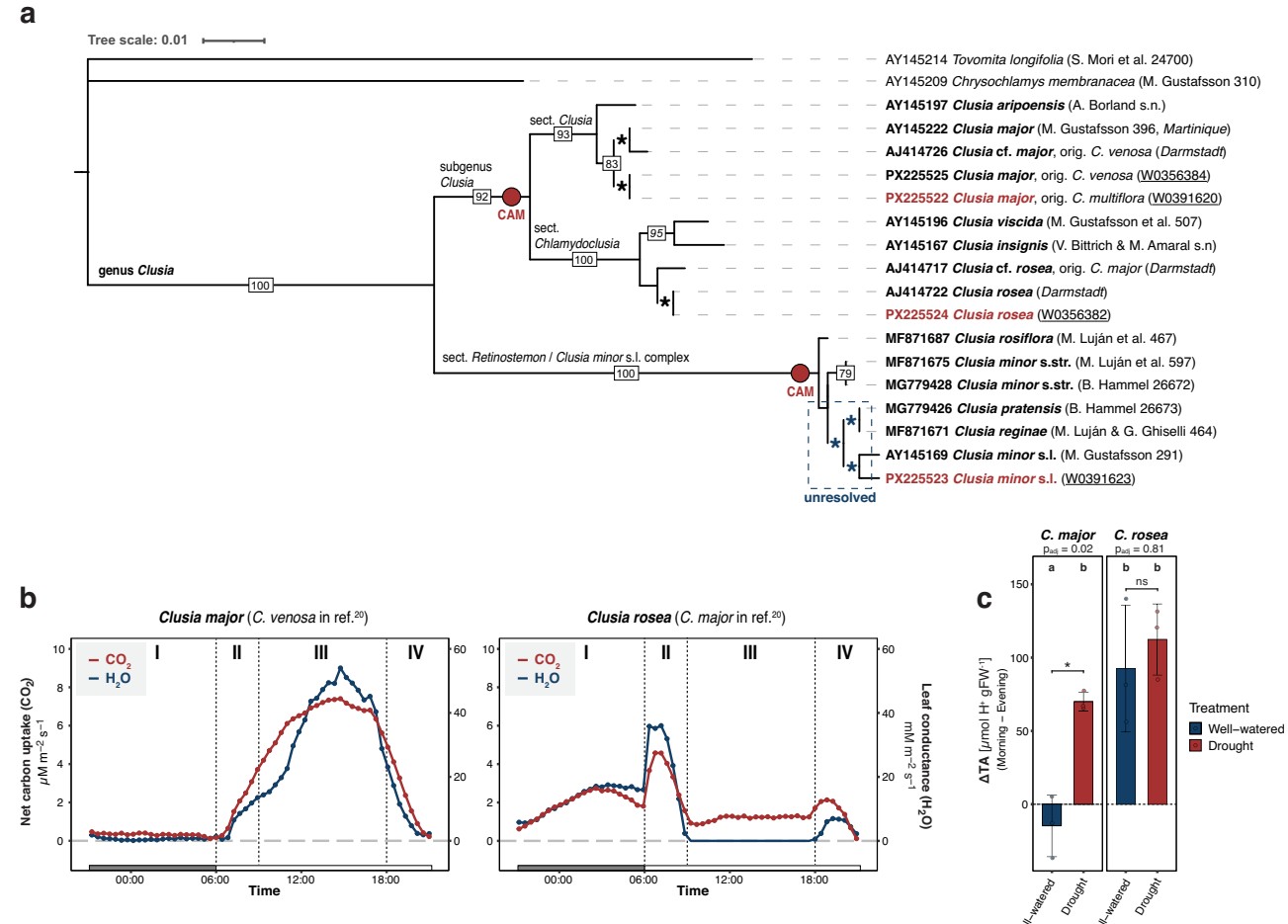

**Fig. 1 | Taxonomic ambiguities and physiology of *Clusia* spp. a** Maximum like-lyhood analysis of nrITS marker sequences of selected specimens (voucher) revealed taxonomic incongruities (see original identification in specimen labels). Consensus tree constructed from 1000 bootstrap trees, support values above 70% are indicated on the branches. Branches marked with an asterisk* exhibit sequence identity (branches shifted for visualization purposes). ITS markers are insufficient to resolve the *C. minor* s.l. complex, likely due to high rate of hybridization. Red labeled specimens were used for whole-genome sequencing. Red circles show generally recognized origins of CAM. **b** Net $CO_2$ fixation and stomatal conductance over a 24-h period under drought conditions. Dark bars indicate nighttime. Nega-tive values of leaf conductance have been corrected. *C. major* (left) exhibits a $C_3$-like mode of photosynthesis but with a slightly positive nocturnal carbon uptake, indicating weak CAM. *C. rosea* (right), conversely, opens its stomata and fixes carbon during the night (c). The phases II and IV are relatively pronounced showing substantial $CO_2$ fixation at dusk and dawn. These findings are consistent with the revised species investigated in the literature[20] (see Supplementary Discussion 1). **c** Nocturnal changes in titratable leaf acidity (TA) under well-watered and drought conditions, measured in technical triplicates for three biological replicates per species ($n = 3$). Significant nighttime acid accumulation under drought conditions demonstrates an induction of CAM in *C. major*. Letters denote statistically sig-nificant differences (two-sided ANOVA, and Tukey's post-hoc test, $p_{adj} < 0.05$). The $p_{adj}$-values refer to within species comparisons of treatments. Error bars represent the ±standard deviation. Experimental conditions and measurements over a full 24-h cycle are provided for all sequenced species in Supplementary Fig. 1. Source data are provided as a Source Data file.

the genome, of which 97% have been assigned a putative functional annotation based on homology within either the SwissProt or EggNOG databases. Gene and repeat densities are inversely correlated: Coding sequences (CDSs) and pseudogenes are most abundant in distal chromosome regions and repeat density is highest towards proximal regions (Fig. 2).

One apparent pattern we observed was the extensive collinearity between two sets of chromosomes, allowing us to assign homo-eologous chromosome pairs. We could not identify putative sub-genomes via transposon signatures[33] because the pairings did not show significant separation (Supplementary Fig. 4). Therefore, we introduced the groups H1 (homoeolog 1) and H2 (homoeolog 2) to distinguish between homoeologous genes located on the size-ordered syntenic chromosome pairs (Fig. 2). We hypothesize that the differ-ences between the groups reflect divergence triggered by diploidiza-tion, which may have been facilitated by reinforcement of bivalent pairing and disomic inheritance.

## WGD and diploidization shaped homoeolog landscape and pseudogenization

We investigated the polyploid nature of the *C. major* genome and the potential connection to the mixed CAM/$C_3$-like physiotypes in *Clusia*. The macrosyntenic relationships between H1 and H2 in the haploid genome representation unambiguously show a polyploid origin, as evidenced by the presence of duplicates of each chromosome (Fig. 3a). Despite some rearrangements between homoeologous chromosomes, they display a high degree of synteny when scaled by gene rank order. However, collinearity is often (but not always) restricted to regions with high gene density along chromosomal arms (Fig. 2). On a physical scale, proximal accumulation (or elimination) of repetitive elements appears to strongly differentiate homoeologous chromosome pairs by size (Supplementary Fig. 5).

Since we found *C. major* to be a diploidized tetraploid, we won-dered how diploidization may have affected transposable elements (TEs) and the overall gene content in terms of pseudogenization and

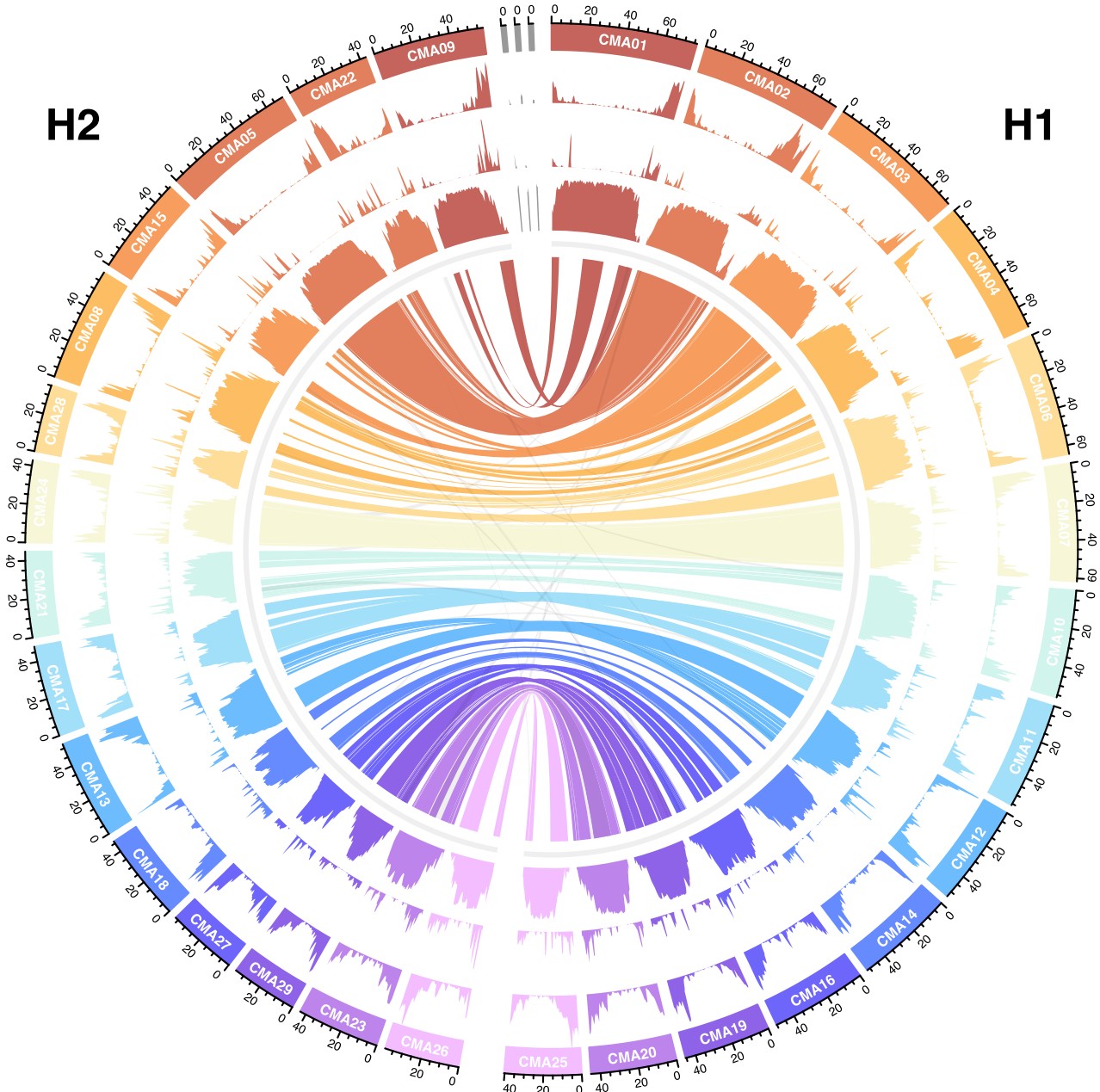

**Fig. 2 | The *C. major* genome.** Circos plot of the haploid chromosome-level assembly comprising 29 anchored pseudo-chromosomes with a size ranging from 76 Mb (CMA01) to 35 Mb (CMA29). The three remaining scaffolds are placed at the end (top). The tracks from the outer to the inner ring show the nucleotide densities in 1 Mb windows of genes, pseudogenes, and repetitive elements, respectively. Syntenic/colinear links between all chromosomes are shown in the innermost circle. Pairs (or triplets) of homoeologous chromosomes with strong collinearity are colored together accordingly, whereas other smaller links are shown in gray. Homoeologous pairs of chromosomes and their respective genes (homoeologs) are further grouped into H1 (right) and H2 (left) based on the size of the syntenic chromosomes affected by repetitive elements.

fragmentation. The distribution of repetitive elements in *C. major* along an evolutionary time scale shows one major wave of Class I transposable elements accumulation, except for LINEs showing a second, older peak. LTR-retrotransposons and other unannotated retrotransposons account for 70% of this massive genome expansion (Fig. 3b). The Ty3/*Gypsy* superfamily members are found to be most abundant in *Clusia* genomes, and known to preferentially localize in putative (peri)centromeric regions in *Populus trichocarpa* Torr. & A. Gray ex Hook., where they might affect recombination rates[34]. Transposition bursts and differential insertion/presence of active TE families in plant genomes can impact gene loss and fractionation. In fact, gene deletion by illegitimate recombination, either involving TEs or not, is

considered to be the predominant mechanism of fractionation and, together with unequal recombination, genome size reduction following WGDs in flowering plants[22,35].

We selected 18,306 high-confidence non-pseudogenized genes (51 % of all gene models) in the haploid assembly based on similarity to and coverage of manually curated proteins, to investigate CDS pseudogenization. Using a homology-based algorithm[36], we identified 11,175 pseudogenes (Fig. 2, Fig. 3c, Supplementary Data 5). Hence, a minimum of 27 % of all putative former genes in *C. major* have, over time, lost their function. The vast majority consists of unprocessed pseudogenes (Fig. 3c). These originate from duplicated parent genes (DUP), e.g. homoeologs, that have lost their primordial protein

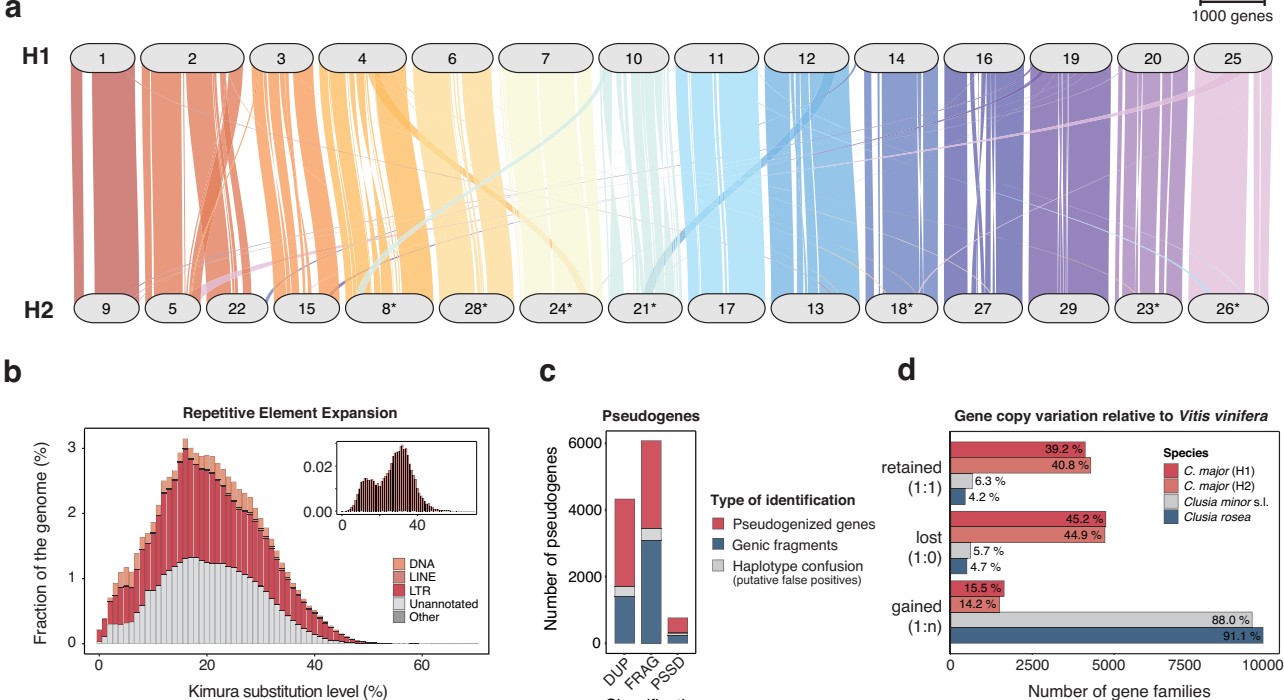

**Fig. 3 | The *Clusia major* genome is undergoing genic diploidization. a** High collinearity between homoeologous chromosomes clearly shows the polyploid origin of the species. Groups H1 and H2 comprise 17,806 and 17,800 syntenic genes, respectively (99 % of all gene models). Pseudo-chromosomes are scaled by gene rank order rather than physical positions. Inverted chromosomes are marked with an asterisk*. There is only one major structural rearrangement on chromosome 2, besides some inversions such as between the chromosomes 4/8 or 16/27. **b** Recent burst of transposable element (TE) amplification explains genome expansion. Copy-divergence analysis of TE classes based on Kimura distances (X-axis) and their represented fraction in the genome of *C. major* (Y-axis). Older copies are located on the right side of the graph while rather recent expansions are located to the left. The interspersed repeat landscape of LINE (small box) displays two waves of expansions, with an evolutionary more ancient major peak. DNA: DNA transposons (Class II), LINE: Long interspersed nuclear elements (Class I), LTR: Long terminal repeat retrotransposons (Class I). **c** High amount of pseudogenes cause homoeolog

fragmentation in *C. major*. Shown are the number (Y-axis) and classification (X-axis) of pseudogenes that accumulated premature stop codons, frameshifts and/or a 3′ polyadenine tail based on high confidence parent genes. DUP: duplicated pseudogenes, FRAG: highly fragmented pseudogenes (low coverage to parent gene/homoeolog but high amino acid identity), PSSD: retrotransposed (processed) pseudogenes. More than half of the identified pseudogenes overlap with annotated gene models, indicating non-functionalization (pseudogenized genes). Numerous genic fragments remain in the genome (not meeting annotation criteria to be classified as genes). 7 % of the identified pseudogenes were filtered for erroneous mixing, collapsing, or switching between haplotypes. **d** Gene retention analysis reveals the loss of one of each homoeolog in *C. major*. 11,257 gene families were selected (16 % from eudicots sampling), containing around 75% of all putative genes in *Clusia* spp. and 18,720 genes (65 %) in the outgroup *Vitis vinifera*. Source data are provided as a Source Data file.

function because of premature stop codons, frameshifts and/or gene fractionation (FRAG)[37,38].

We wanted to assess the evolutionary fates of ancestrally duplicated genes in all three *Clusia* species and, therefore, combined information from contig phasing[29], protein similarities[39], synteny[40], and *Clusia*-specific nucleotide alignments[41], enabling us to create a high-resolution map of gene origins and sorting of orthologs/paralogs into gene families and (positional) homoeologs. This approach suggests that gene families have experienced large expansions only in *C. minor* and *C. rosea*, which can be attributed to their higher levels of ploidy (Fig. 3d). While only a few families gained genes in *C. major*, e.g., through tandem duplications in one of the homoeologous chromosomes, many homoeologs remain as duplicates in relation to the diploid outgroup. Such retained gene pairs, together with functionally annotated but already pseudogenized genes (Fig. 3c), contribute to the high collinearity between homoeologous chromosomes (Fig. 3a). Most remarkable, however, is that the majority of gene families have lost one of each homoeolog in the putative *C. major* subgenomes (Fig. 3d).

Considering our data, genome-wide, ongoing diploidization by means of differential TE amplification, pseudogenization, or loss of one homoeologous copy of a duplicate gene pair becomes evident in *C. major* (Fig. 3). Diploidization processes involve a combination of

chromosomal rearrangements, TE amplification or elimination, pseudogenization and neo-/sub-functionalization of duplicated genes, as well as changes of the epigenetic landscape[35,42]. Depending on the mode of origin, age of polyploids, and pace of diploidization, which can differ among plant groups, many polyploids represent transitory states of diploidization. This may lead to a partly diploidized genome with a mosaic of gene landscapes with different copy numbers and varying origins of individual genes/genic regions. We believe that these differences in genic landscapes might be connected to the different physiotypes in the genus *Clusia*, with respect to their photosynthetic behavior.

## Intronic repeats and fractionation affect CAM-related genes

With the aim of connecting our genomic observations to plant physiological processes, we analyzed homoeolog fractionation and TE insertions within intronic regions in a large set of ancestrally related CAM genes. Based on a literature review[5,43–49], we compiled a list of 72 gene families comprising 1,010 enzymes, 235 transporters and 434 transcription factors directly or indirectly involved in processes related to CAM photosynthesis in *Clusia* spp. These were subdivided into their contribution to specific pathways, namely starch biosynthesis, glycolytic breakdown of carbohydrates, gluconeogenesis, carboxy-, decarboxylation, malate turnover, vacuolar storage, and regulation

and their respective functionality was analyzed (Supplementary Data 6). In *C. major*, gene families linked to the GO terms of circadian rhythm (e.g. *CCA*, *GATA*), carbon utilization (e.g. *BCA*), and starch/glucan/polysaccharide metabolic processes (e.g. *AGPase*, *SS*, *GWD*, *PGMP*) show strong enrichment of large TE insertions and genic diploidization (Supplementary Fig. 6).

Genic diploidization or fractionation is the process of gene removal and loss following polyploidization via molecular mechanisms such as pseudogenization and gene deletion by recombination[35]. Illegitimate recombination can occur through unequal pairing between intrachromosomal, dispersed homologous, or homoeologous regions, ultimately contributing to genome size reduction[50–54]. Many duplicated gene copies involved in starch biosynthesis and carbon breakdown seem to have preferentially returned to single copy state (Supplementary Fig. 7). According to their level of fractionation (100% compared to the respective homoeologous gene/region/chromosome), homoeologs of starch synthase (*SS/SBE*) and amylase (*AMY/BAM*) gene families, for instance, appear to have been lost via illegitimate recombination.

In areas of the genome that do not experience elevated post-WGD recombination rates, genes may persist and accumulate deleterious mutations, leading to a higher frequency of pseudogenization[37,55]. In contrast to fractionation caused by illegitimate recombination, pseudogenes are not physically removed from the genome and may remain as genic fragments for long periods of time after non-functionalization. Essential CAM enzymes (e.g. *PPDK*, *BCA*) and transporters (e.g. *GPT2*, *ALMT9*) show evidence of pseudogenization such as partly fractionated gene models that accumulated premature stop codons or frameshifts (Fig. 4a). A pyruvate, phosphate dikinase (*PPDK*) gene/genic fragment contains a stop codon and is already 82% fractionated (Supplementary Data 6). Interestingly, the remaining genic fragment (Cma17.g469) also carries a high number of intronic repeats (genome-wide z-score of 5.4), which is, importantly, not the case in the other species. Protein alignments reveal that the parent gene on the same chromosome (Cma17.g462) is flanked by TEs and lacks its first exon, which corresponds to the chloroplast transit peptide in *Arabidopsis thaliana* (L.) Heynh. Therefore, both homoeologs in group H2 appear to be non-functional, which is also reflected by its transcriptional expression and protein abundance (see below).

In general, genes associated with plastidic starch recycling and carbon breakdown seem to be most frequently affected by pseudogenization, fractionation, and intronic transposon insertions (Fig. 4a, Supplementary Figs. 6, 7). Starch-storing CAM plants are thought to utilize phosphorolytic degradation of leaf starch to provide phosphoenolpyruvate (PEP) as the primary organic substrate for nocturnal assimilation of $CO_2$. These plants usually re-route chloroplastic starch degradation from the amylolytic/hydrolytic route (associated with $C_3$) to the phosphorolytic route. This process is catalyzed by a starch phosphorylase (*PHS1*), resulting in the production of Glucose 6-phosphate (G6P) and its export via *GPT2*[5,47,48,56–58] (Fig. 4b). Both, *PHS1* and *GPT2* genes show highly fractionated homoeologs (75–80%) in addition to premature stop codons and frameshifts (Fig. 4a, Supplementary Data 6). Both homoeologs of alpha-glucan-water dikinase (*GWD2*) are also severely affected by intronic repeats as is the phosphoglucan water dikinase (*PWD/GWD3*) and the plastidic phosphoglucomutase (*PGMP/PGM*). In fact, on a genome wide scale, these genes are among the top hits of the genomic loci with multiple long repetitive insertions (Fig. 4c).

Mobilized transposable elements are known to disrupt gene functions and thereby promote neo- or sub-functionalization in a variety of ways[59–61]. *PGMP* is one such example: Two homoeologous loci exist in the haploid genome, one on chromosome four (H1) and one on chromosome eight (H2). Each of these loci has an allelic variant and all four gene products are targeted to the chloroplast due to

thylakoid transit peptides (Supplementary Data 7). Interestingly, the four *PGMP*-encoding alleles differ with respect to large repetitive element insertions within their introns and their promoter *cis*-regulatory motifs (Fig. 4d). Both alleles in H2 share the same, evolutionary older, LINE insertions (Fig. 3b) but only the predominant variant (Cma08.g106) experienced the additional and more recent insertion of a large ~19 Kb region consisting of many LTR-retrotransposons that almost all belong either to the LTR/Gypsy superfamily or an unknown/unclassified retrotransposon family. Such differences in TE content between alleles of the same group are a very rare phenomenon in the investigated CAM-related gene families, even for genes with very long inserts (Fig. 4c).

Promoter alignments also show that Cma08.g106 is the only H2 gene in *Clusia* spp. lacking the evening element at the particular position (Fig. 4d, Supplementary Fig. 8). The pattern of certain *cis*-regulatory motifs being found in almost all *Clusia* genes of one homoeologous group but not the other has also been observed in other CAM-related gene families such as *PEPC4*, PEPC-kinase and *PHS2* (Supplementary Fig. 8, Supplementary Data 8). As shown in the following sections, it appears that the evening element in H2-*PGMP*s also causes an altered expression pattern compared to its sister genes in H1. Therefore, the presence of the evening element and the corresponding shift to nocturnal expression indicate that the enzymes have undergone sub-functionalization (and, perhaps, neo-functionalization: only 89 % protein identity to H1 homoeologs).

Together, these data clearly show that progressive accumulation of transposon insertions in and around coding regions as well as dynamics of genic diploidization such as homoeolog fractionation and pseudogenization caused the loss of essential plastidic CAM and starch degradation functions. The loss of homoeologs over time is not necessarily harmful for plants and the long-term return to disomic inheritance allows for duplicated networks to be rewired[62]. We hypothesize that genic variability in the phosphorolytic starch degradation pathway is related to physiological changes in plastidic starch metabolism in *C. major* and may be important for carbon fixation and stomatal behavior[63]. Our objective is not to claim that diploidization uniquely targets CAM genes to the exclusion of other pathways, nor that CAM evolution required exceptional genome-wide behavior. Rather, our goal is to point out that key CAM-associated genes did undergo fractionation and pseudogenization during diploidization, and that these changes are plausibly linked to the observed CAM physiotype diversity.

## CAM signatures under close-to-real-world conditions

In order to establish a causal link between disruptions in starch metabolic genes and the plants' photosynthesis mode, we tested the degree to which the three species sequenced here differ on a molecular level: In an 'open greenhouse' experiment, we exposed the plants to conditions mimicking native habitats under (a) canopy environments (shaded + well-watered) as well as (b) adverse conditions (high-light + drought), to see how they cope with variable environmental conditions (Fig. 5a–d, Supplementary Table 3). We made use of natural sunlight and, therefore, light exposure was subject to a certain degree of variability (Fig. 5b, c) but we determined ΦNPQ—the efficiency of regulated non-photochemical quenching—and observed that both shaded groups displayed little to no heat dissipation while both exposed groups had high ΦNPQ values (Fig. 5d), reflective of the exposure treatment. We performed transcriptomic, proteomic, and metabolomic analyses on the same set of samples from this experiment (Figs. 5, 6). First, we realized that the separation of transcriptome data along principal components followed a similar trajectory in both the strong CAM plant *C. rosea* and in the $C_3$-like plant *C. major* under shaded conditions (Fig. 5e, f), suggesting some overall temporal similarities between gene expression profiles. We also performed these experiments for the facultative CAM plant *C. minor*, but the

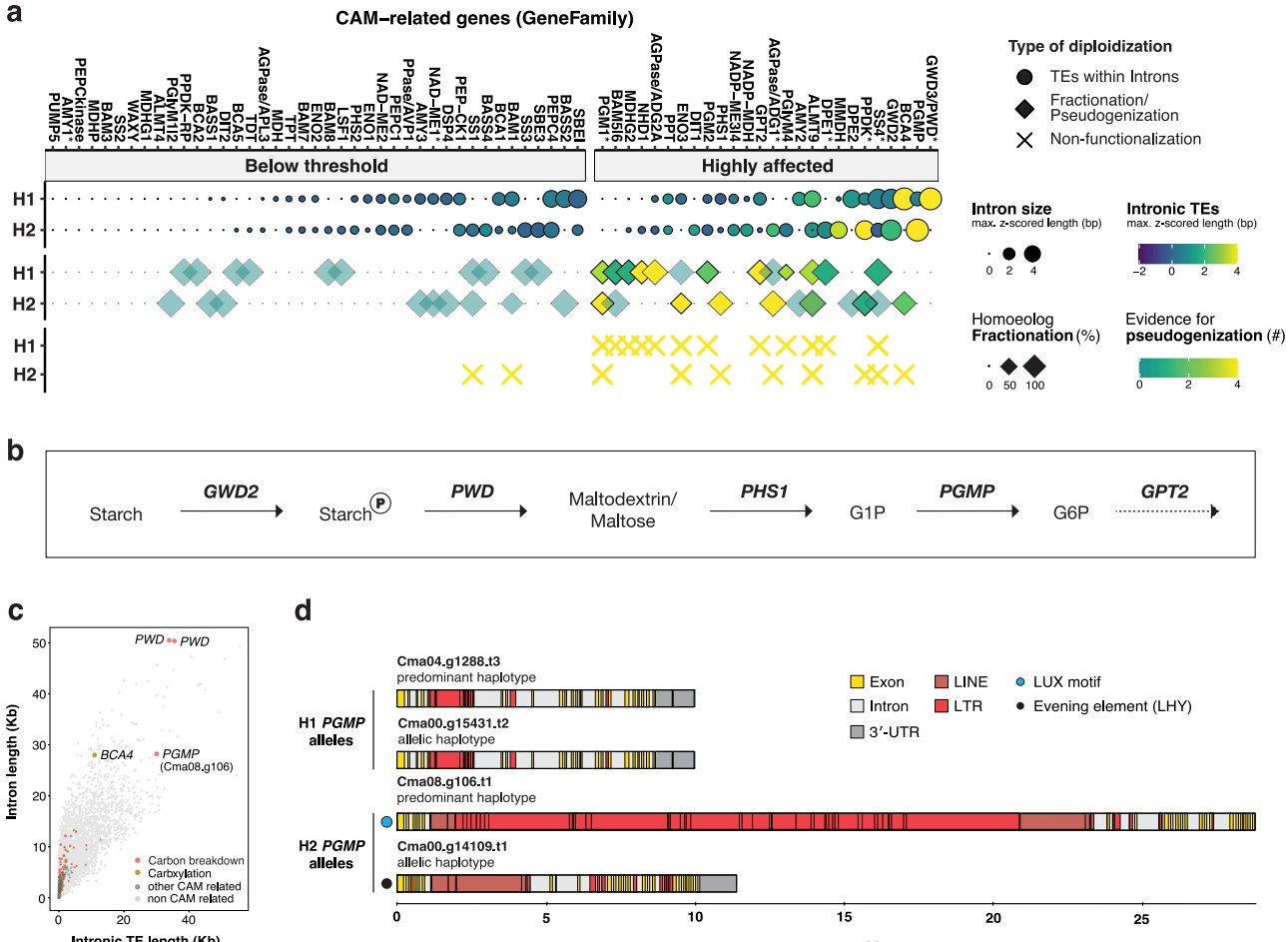

**Fig. 4 | Intronic TEs and homoeolog fractionation affect plastidic CAM and starch degradation pathways in *C. major*. a** Signals of diploidization in CAM-related gene families (X-axis). Individual gene copies within H1/H2 are summarized to their maximum intron/repeat length (z-scored bp) and mean percent of fractionation compared to the corresponding homoeolog (Y-axis). "Highly affected" means that either of the homoeologs exhibits a genome-wide z-score of intronic repeats >=1 and/or evidence of pseudogenization >0 (premature stop codons, frameshifts, polyA-tails). Gene families labeled with an asterisk are non-positional (non-syntenic) homoeologs. Data are sorted by intron size. **b** Proposed/schematic pathway of the phosphorolytic route of chloroplastic starch degradation. **c** Genome-wide intron and intronic repeat lengths (Kb) of all genes/alleles. CAM-related pathways are highlighted. **d** All genes encoding for plastidic phosphoglucomutase (PGMP), separated into a predominant and allelic haplotype as well as their positional homoeologs. Genomic repeat insertions and motif occurrences illustrate their individual properties. The models are aligned according to the transcriptional start site (TSS). Source data are provided as a Source Data file.

recalcitrant nature of the plant prohibited us from obtaining clean transcriptomic data.

We specifically focused on genes and proteins involved in essential CAM processes (Supplementary Figs. 9–12). We observed time-dependent shifted expression of genes and a differential abundance of proteins within both the carboxylation and decarboxylation pathways across timepoints. These generally followed the same diurnal oscillations in *C. major* and *C. rosea* (Fig. 5g-i). *C. major* appears to re-fix respiratory $CO_2$ during the night via the CAM-specific isoform *PEPC1*[18,64] (Fig. 5g). Like in *C. rosea*, *PEPC* protein abundance in *C. major* follows a strong diurnal pattern and the same *PPCK* ortholog is used in both species for its activation, as we conclude from its high level of transcript expression in the evening and during the night (Fig. 5h). The transcriptionally controlled *PPCK* phosphorylates *PEPC*, which is less sensitive to inhibition by malate in its phosphorylated form and has higher substrate affinity. During the day, *PPCK* expression disappears altogether. Organic acids need to be decarboxylated to free the carbon fixed on to them. The main decarboxylation modules in *Clusia* seem to be PEP carboxykinases (*PEP-CK*): Many different isoforms are highly expressed, and protein abundance follows a diurnal oscillation pattern, which peaks in the evening (Fig. 5i). Although diurnal oscillation

patterns are similar in *C. rosea* and *C. major*, the overall relative protein abundance of *PEP-CK* is significantly higher in *C. rosea*, matching higher organic acids levels.

CAM associated metabolite profiling via targeted GC-MS further confirmed that *C. major* performs some type of $C_3$ + CAM. In CAM, RuBisCO is supplied with $CO_2$ via decarboxylation of malate and, possibly, other organic acids, such as citrate[15,65]. Until the morning, organic acids accumulate in both *C. major* and *C. rosea* under control conditions, but not in the facultative CAM plant *C. minor* (Fig. 6e, Supplementary Fig. 13). In all three species, nocturnal malate accumulation is highly induced under adverse conditions. Here, even under well-watered conditions, *C. major* accumulates organic acids, suggesting that this species expresses the CAM cycle constitutively (albeit at a low level). *C. major*, therefore, displays all molecular hallmarks of a constitutive $C_3$ + CAM physiotype in a close-to-real-world scenario.

It is generally recognized that the subgenus *Clusia* to which both *C. major* and *C. rosea* belong has ancestrally developed CAM[66] (Fig. 1a). On a physiological level, both plants clearly have different ways of fixing carbon. In contrast to *C. rosea*, *C. major* opens its stomata during the entire day to directly fix $CO_2$ via RuBisCO (Fig. 1b). However, its high WUE (water leaf conductance versus $CO_2$ net assimilation rate)

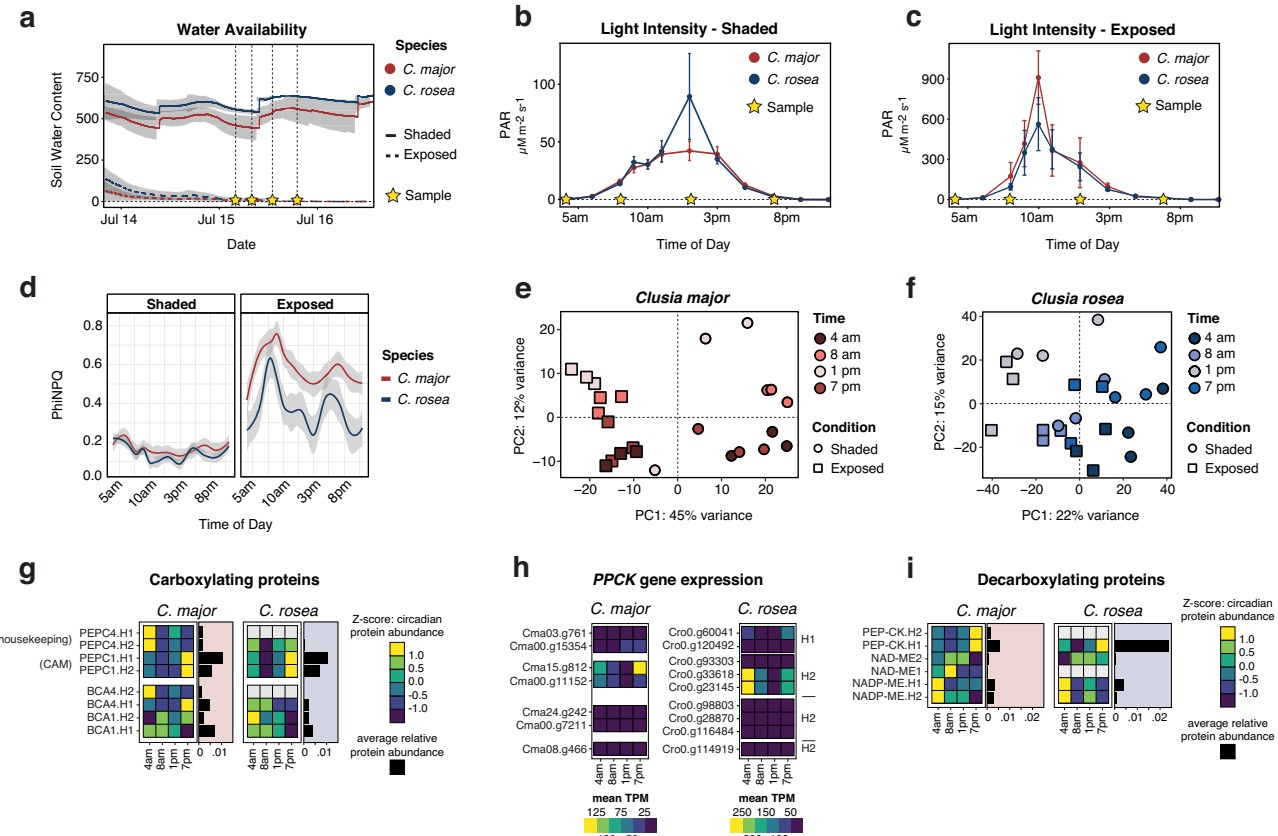

**Fig. 5 | *C. major* and *C. rosea* both display CAM-like gene expression and protein activity patterns in an open greenhouse experiment using natural light.** *Clusia* species were exposed to varying environmental conditions to investigate stress and CAM-type performance. **a** Soil water content as relative arbitrary digital output units. Stars indicate timepoints when multiomics samples were taken. Three sensors per species/treatment combination ($n = 3$). Light intensities measured via PhotosynQ for shaded (**b**) and high-PAR-exposed (**c**) plants. One replicate represents a single measurement per cutting. Groups: *C. major*/shaded ($n = 6$), *C. major*/exposed ($n = 7$), *C. rosea*/shaded ($n = 10$), *C. rosea*/exposed ($n = 10$). For **b** and **c**, error bars indicate the standard error of the mean (SEM). **d** PhiNPQ values (quantum yield of regulated non-photochemical energy loss in PSII) measured via PhotosynQ across the sampling day for both conditions, same replication as **b** and **c**. For **a** and **d**, lines represent loess-smoothed trends; shaded ribbons indicate the 95% confidence interval of the smoothed mean. Principal component analyses (PCA) of RNAseq data for *C. major* (**e**) and the obligate CAM plant *C. rosea* (**f**) across a diel cycle at four timepoints (4 am, 8 am, 1 pm, 7 pm) under two conditions ($n = 3$ per timepoint/condition pair). Heatmaps of carboxylating (**g**) and decarboxylating (**i**) enzyme abundances from proteomics for *C. major* (left) and *C. rosea* (right), displayed as z-score centred values across four timepoints. Bar charts indicate average relative protein abundance across all timepoints. **h** PEPC kinase (PPCK) gene expression (mean TPM) from RNAseq across the four timepoints for *C. major* (left) and *C. rosea* homologs (right). Source data are provided as a Source Data file.

similar to that of *C. rosea* represents an important phenotypic indicator of some degree of CAM (PEPC activity). Significant nocturnal $CO_2$ fixation could not be enforced in any setting, instead *C. major* likely supplies additional $CO_2$ via weak CAM, which would explain the $C_3$ + CAM behavior we observe. Our data implies that *C. major* either lost or never fully acquired the ability to perform strong CAM as a result of ongoing genic diploidization that affected genes involved in CAM and starch-related processes. Whether this is a response to long-term environmental conditions of its specific habitat/distribution area needs further investigation. But what has led to the physiological reprogramming that caused the difference between these plants?

## CAM physiotypes in *Clusia* are associated with the starch/sugar-malate cycle

Starch metabolism has a strong impact on nocturnal malate accumulation and, consequently, on stomata opening[48,58,67]. Our data on diploidization, pseudogenization, fractionation, and non-functionalization illustrated that genes involved in starch degradation were significantly affected by these processes (Fig. 4, Supplementary Fig. 6). The amylolytic/hydrolytic route ($C_3$) appears to be prominent and functional in the *Clusia* species studied here. Chloroplastic ß-amylase 3 (*BAM3*) genes are fully functional in both homoeologs and are highly expressed at night (Fig. 6a). However, the

expression of *BAM3* in H2 largely ceases in stressed *C. rosea* and, simultaneously, the enzyme abundance of the phosphorolytic marker gene *PHS1* strongly increases before sunrise (Fig. 6b). Depending on the condition, *C. rosea* apparently shifts from daytime synthesis to nighttime activity (degradation mode), while *PHS1* is generally strongly induced in stressed *C. minor*. Such diurnal differences are absent in *C. major*, where all genes of the putative CAM-associated phosphorolytic route show highly fractionated homoeologs in addition to the accumulation of intronic TEs. The *cis*-regulatory evening element of plastidic phosphoglucomutase (*PGMP/PGM*) in H2 alters its expression pattern compared to its homoeologs in H1, where both alleles lack this motif and are activated in the morning (Fig. 6c). This aligns with the expected role of plastidic *PGM* in daytime starch biosynthesis. In contrast, the sub-functionalized genes in H2, especially the predominant allele most affected by transposable elements (Fig. 4d), shift towards evening expression, supporting the proposed phosphorolytic pathway of chloroplastic starch degradation.

As a conclusive piece of evidence, we investigated starch pool levels in the three species. We identified a higher starch pool in *C. major* at the end of the night compared to *C. rosea* and *C. minor* (Fig. 6d). Decreased nocturnal starch degradation would provide less substrate for nocturnal malate accumulation and, subsequently, a lower $CO_2$ internal partial pressure in the leaves during the day, forcing

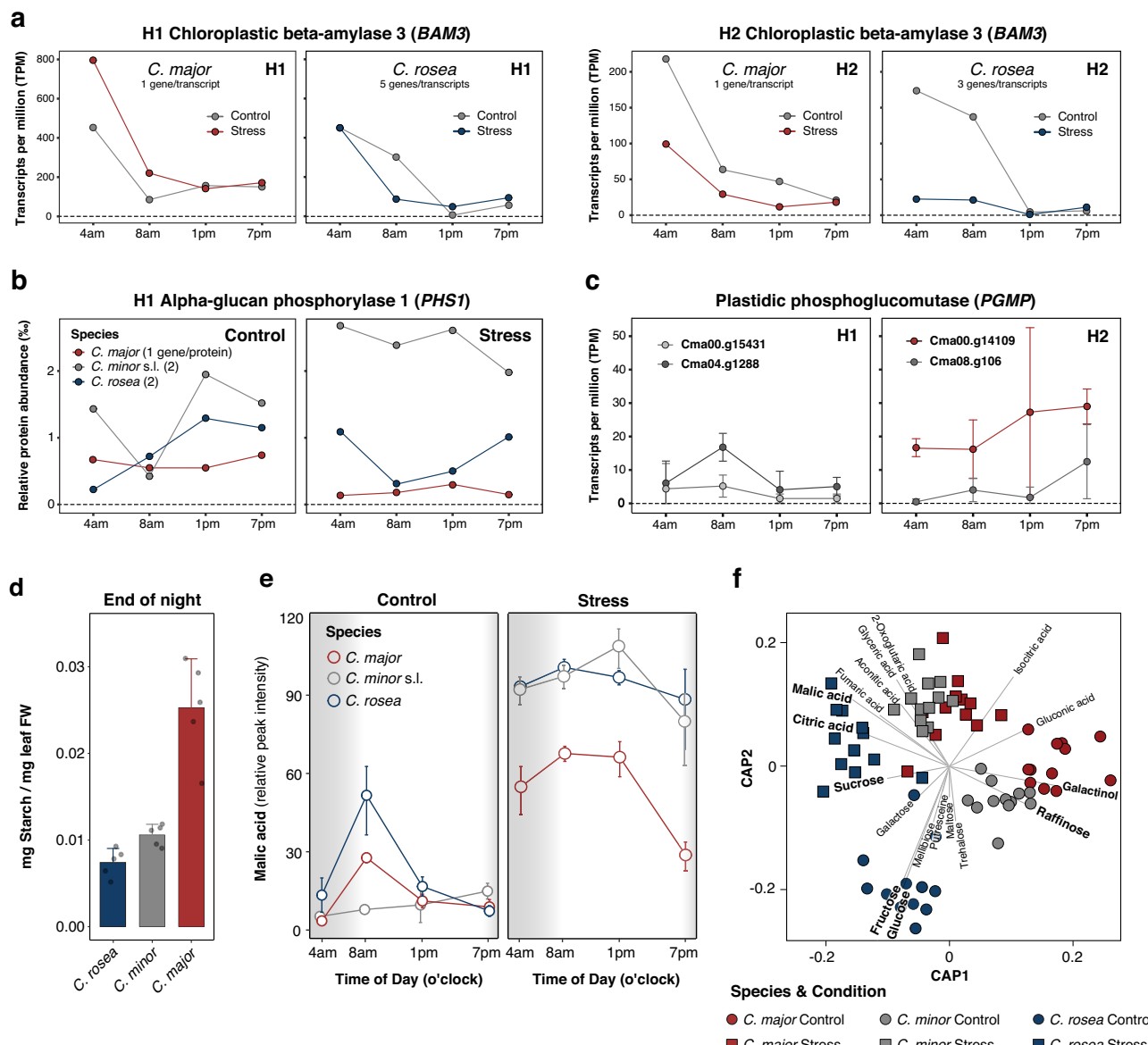

**Fig. 6 | The starch/sugar-malate cycle is associated with the physiotypic diversity of CAM in *Clusia*.** Chloroplastic leaf starch and various soluble sugars are degraded to supply carbohydrates for additional carbon fixation. **a** Overall gene expression of BAM3 in *C. major* and *C. rosea* shows that the C₃-associated amylolytic/hydrolytic route of starch degradation predominates. Transcripts are averaged over replicates ($n = 3$) and paralogs are summarized into H1 (left panels, $n = 6$) and H2 (right panels, $n = 4$). **b** PHS1 protein abundance averaged over replicates ($n = 3$) of the five genes/proteins summarized in H1 (no mRNA/protein activity in genes/genic fragments of H2). **c** Mean transcript expression under control

conditions of the four alleles encoding for PGMP in *C. major* ($n = 3$ per transcript per timepoint). Error bars represent the ±standard deviation. **d** Quantification of soluble starch in leaves of *Clusia* spp. Samples were collected at the end of the dark period ($n = 5$). Error bars represent the standard deviation. **e** Malic acid levels during a 24-h cycle under stress (right panel) and control (left panel) conditions ($n = 3$ per species/condition/timepoint group). Error bars represent the ±standard deviation. **f** Canonical analysis of principal coordinates (CAP) analysis of metabolites in three *Clusia* species grown under control and stress conditions ($n = 12$). Source data are provided as a Source Data file.

*C. major* to open its stomata during the day as we observed (Fig. 1b). In line with this hypothesis, *C. major* also shows lower malate accumulation during the night compared to *C. rosea* (Fig. 6e). Interestingly, recent studies with mutants in starch metabolism (e.g., *PGMP* knockout) in *Kalanchoë fedtschenkoi* Raym.-Hamet & H. Perrier and *A. thaliana* revealed the plasticity of the plants to channel assimilates into soluble sugars compensating for deficiency in transitory starch synthesis and degradation[56,63,68,69]. In certain CAM plants these soluble sugars are degraded instead of starch to produce PEP[58,70]. The degradation of the soluble sugars fructose, glucose, and sucrose was found to be predominantly associated with nocturnal malate synthesis in *Clusia* species based on their oscillations[71]. In fact, while more stressful

conditions induce CAM-related organic acid accumulation in all species to varying extents, their metabolic strategies in response to water limitation, however, differ drastically (Fig. 6f, Supplementary Fig. 13). *C. rosea* switches from high fructose and glucose pools to accumulating sucrose, which has osmo-protective properties and serves as an energy reserve for nocturnal CAM metabolism when photosynthetic activity is diminished under stress[72]. Remarkably, and in contrast to *C. rosea* and *C. minor*, *C. major* accumulates high levels of the potent compatible solutes raffinose and its precursor galactinol[73] (Fig. 6f, Supplementary Fig. 13), indicating an alternative route for soluble sugar-storing CAM plants or an intermediate stage between C₃ and CAM. The RFO sugar family is well described in C₃ and C₄ plants as an

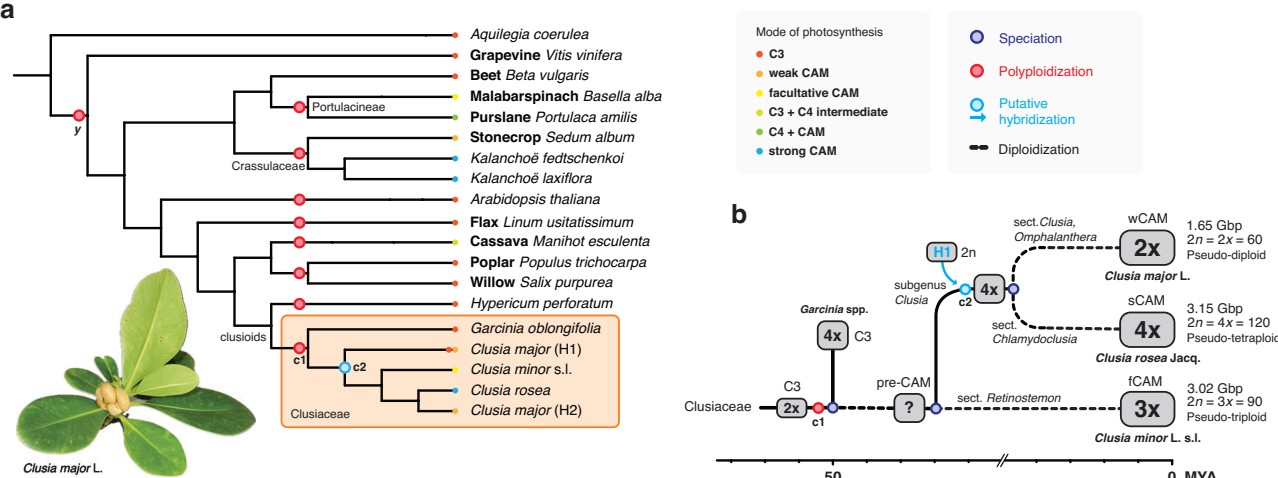

**Fig. 7 | Cladogram of selected eudicots and the evolution of CAM physiotypes in *Clusia*. a** Species tree (cladogram) of selected eudicot lineages inferred from 1845 gene trees via orthogroups. Sample size and hidden paralogy largely prevent any consideration as resolved phylogeny. Circles on the terminal branches illustrate the mode of photosynthesis of corresponding species. Known events of polyploidization cluster in close proximity to the Cretaceous-Paleogene (K-Pg) boundary[79,81] (c1). The placement of the putative subgenomes of *C. major* (H1/H2) into different nodes indicates hybridization/allopolyploidization (c2) of a yet unknown origin/physiotype. **b** Schematic illustration of the evolution of CAM in Clusiaceae. CAM is only reported in the most species-rich genus of *Clusia*[6]. The ancestral physiotype of a putative monophyletic pre-CAM ancestor remains elusive. Different sections/subclades of our selectees allow independent long-term adaptation towards weak, facultative, and strong CAM. The c2-WGD event is likely independent in the subgenus *Clusia* given the phylogenetic age and linear correlations of both the chromosome numbers and genome sizes (Gbp/1 C), eventually resulting in descending dysploidy for *C. major* (i.e., chromosome number reduction). However, since the phylogenetic placement of speciation and polyploidization requires broader selections of species, it could also have occurred prior to the radiation of subclades as clustered in the species tree (**a**).

osmo-protectant and scavenger of oxidative stress during drought, heat and cold[69,73–75].

## Diploidization of ancient polyploids underlies ecophysiological diversity of CAM

*Clusia* is the only known genus containing trees with typical dicotyledonous secondary growth that perform CAM. The genus is considered monophyletic but with a polyphyletic evolution of CAM, as the typical $C_3$ physiology has been observed exclusively within one subclade[76]. CAM is therefore thought to have evolved convergently, at least twice (Fig. 1). However, taxonomic misidentifications and physiological inconsistencies (Supplementary Discussion 1) as well as the imprecise diagnostic methodology of plants performing any type of $C_3$ + CAM subsequently biased the reconstruction of ancestral traits[6,77]. The question arises whether CAM evolution in *Clusia* resembles that in Crassulaceae or Portulacineae where CAM evolved monophyletically from a single $C_3$-photosynthesis ancestor during radiation[78] (Fig. 7a).

WGD events near the Cretaceous–Paleogene boundary in the clade of clusioids have been proposed as a response to periods of environmental and ecological upheaval[79], coinciding with the putative origin of the family Clusiaceae[80]. In fact, $K_S$ anchor pairs (synonymous substitutions in gene coding regions of syntenic paralogs) reveal that another, older, polyploidization event (hereafter named c1) occurred before the more recent one (c2, H1/H2). Two independent rounds of WGD are thus supported by the presence of evolutionarily distant collinear quadruplicated regions across all chromosomes in the haploid genome representation (Supplementary Fig. 14a). Relative timings of speciation and polyploidization events further indicate that the c1-WGD is shared within Clusiaceae (Supplementary Fig. 14b). Ongoing diversification could have conferred a selective advantage to some polyploids due to increased genetic variation and competitiveness, leading to their initial short-term survival and establishment[81]. Such polyploid formation may promote speciation and be accompanied by changes in gene expression and epigenetic remodeling, which in turn could have guided the key evolutionary innovation of (pre-)CAM in *Clusia*, as a trait distinct from $C_3$ (Fig. 7b).

It has repeatedly been proposed that many polyploids have a higher stress tolerance and environmental robustness than diploids, which is supported by the observation that present-day polyploids often thrive in newly created, disrupted, or harsh environments[81]. Fractionation is a particularly important component of diploidization and post-polyploidy genome evolution because nearly all polyploids that persist experience gene removal and the resolution of duplicate gene networks. Although the forces and mechanisms that drive genic diploidization are not yet fully understood there is plenty of evidence that the process is generally not random with respect to parental subgenomes and types of genes that are retained or lost (i.e., biased fractionation)[35,62].

All species/individuals investigated in this study seem to be functional diploids (i.e., polyploid organisms that functionally mimic diploids in a way that, despite having multiple sets of chromosomes, genes may be silenced or expressed in a balanced manner) of a tetra-, hexa-, or octoploid origin (Supplementary Fig. 3). Chromosomal studies[82] also indicate $x = 30$ as the base chromosome number for the genus *Clusia*, suggesting that the original genome(s) of initial polyploids have been extensively diploidized. Two ancient rounds of WGD would allow for two separate cycles of diploidization under different atmospheric/climatic conditions (Supplementary Fig. 15). We therefore propose that polyploidization contributed to increased evolutionary plasticity within *Clusia*, whereby lineage-specific diploidization differentially shaping CAM-associated traits, without necessitating a present-day correlation between genome size and CAM strength, consistent with previous findings[21]. Given the c2-WGD event during genus evolution (with or without hybridization of taxa that are the donors of the putative subgenomes H1/H2), we hypothesize that at least the most recent diploidization cycle may have played a crucial role in the evolution of CAM physiotypes in *Clusia*, by directing subclade-specific, independent long-term adaptation towards weak, facultative, and strong CAM (Fig. 7b).

Our selected species are not closely related and each represents a different section within the genus (Fig. 1a) with distinct distribution areas across a wide range of ecosystems[2,83,84]. Distinct distribution

areas are indicative of different ecological niches or selective pressures that the ancestors of our extant taxa were exposed to. As post-WGD adaptation proceeds, selection to maintain dosage balance and stoichiometry will relax over time, allowing duplicated networks to be rewired and to evolve novel functionality and increase biological complexity[81]. While *C. rosea* can be found in harsh, dry, and salty environments[85], which might explain its high WUE and strong CAM phenotype, *C. major* is endemic to the islands of the Lesser Antilles[86], which, in turn, may represent the absence of a selective pressure to evolve or maintain a strong CAM phenotype. When duplicated gene copies within specific pathways were adapted (sub-/neofunctionalization), phased out (intronic TEs) or lost (non-functionalization via fractionation), these could no longer contribute to an efficient CAM metabolism. Eventually, this could have forced *C. major* back to an energetically favorable $C_3$-like behavior but with mixed CAM elements remaining intact.

Whether an ancestral physiotype originated as strong CAM (and experienced several reversals[76]) and/or as a $C_3$ + CAM intermediate[78,87] remains an open question in the genus. However, the consecutive occurrence of polyploidization and diploidization would provide a general mechanism for the nascent expression of (pre-)CAM elements in $C_3$ plants[5,87] which, once established, emerges a qualitatively distinct phenotype as required for the metabolic reprogramming that links breakdown of storage carbohydrates to the process of net dark $CO_2$ fixation in strong CAM plants[8].

Future research on accurate diagnosis, well-resolved phylogenies, and trait evolution in large surveys across CAM/CCM lineages together with comparative genome evolution, physiological transitions, and multiomics in closely related species or even populations may further clarify the evo-ecophysiological diversity and origin of CAM, or even other CCMs[6,88,89]. This could transform our current understanding through the discovery of remnants of the proposed mechanisms in agriculturally valuable crops. *Clusia* is an ideal model system for studying the remarkable plasticity and diversity of these neotropical trees and their $C_3$ + CAM subtypes or discrete phenotypic phases[87] of evolutionary trajectories of CAM.

## Methods

### Origin, documentation and identification of plant material

The studied *Clusia* accessions were obtained from the collection of the Molecular Systems Biology lab at the Department of Functional and Evolutionary Ecology, University of Vienna, including the living collection originally curated by Ulrich Lüttge. Our laboratory maintains a backup of this *Clusia* collection, with the primary collection still housed at the Botanical Garden of the Technical University Darmstadt, Germany. Cuttings of the entire collection were transferred to the greenhouse of the Department of Functional and Evolutionary Ecology, University of Vienna and cultivated and propagated there. Initially the greenhouses were located at UZA1, Althanstrasse 14, 1090 Vienna, and as of 2021, the plants were transferred to the new greenhouse at the UBB, Schlachthausgasse 43, 1030 Vienna.

Voucher specimens of the cultivated species were prepared following standard herbarium practice and preserved as dried specimens and/or flowers in 70% ethanol, and permanently stored in the herbarium of the Natural History Museum Vienna. Digital images of the voucher specimens and photos of the corresponding living plants are available at the international herbarium database JACQ. Herbarium vouchers for the three sequenced specimen are W0391620 (https://w.jacq.org/W0391620, *C. major* L.), W0391623 (https://w.jacq.org/W0391623, *C. minor* L. s.l.), and W0356382 (https://w.jacq.org/W0356382, *C. rosea* Jacq.). All other relevant specimen vouchers including links to virtual herbaria sources are documented in Supplementary Table 1.

Identification of *Clusia* is notoriously difficult, especially when sterile, but the studied plant species *C. major*, *C. minor* s.l. and *C. rosea*

could successfully be identified by consulting relevant floras, revisions and other literature[85,86,90] and by comparison with authentic specimens in the collection of the herbarium of the Natural History Museum Vienna. See Supplementary Discussion 1 for detailed information on the identification of these plants.

### Molecular barcoding

Leaf material from individuals of our *Clusia* collection were physically homogenized by grinding in liquid nitrogen with the addition of 2–3% m/m PVPP. DNA extraction was performed according to the DNeasy Plant Pro Extraction Kit from Qiagen N.V., Germany (ID 69204). PCR amplification of the ITS locus was done using the primer pairs of ITS1 (5´- TCC GTA GGT GAA CCT GCG G – 3´) and ITS4 (5´- TCC TCC GCT TAT TGA TAT GC – 3´), which flank the ITS1, 5.8S rDNA, and ITS2 sequences of a plants ribosomal DNA. Reactions were performed according to the protocol of the Phire Hot Start II Polymerase (Thermo Fisher Scientific, MA, USA) and additionally treated with 3% of dimethyl sulfoxide (DMSO). PCR products were purified using the silica-based PCR Clean-Up Kit (GeneMark, Hopegen Enterprise, Taiwan). A qualitative analysis of DNA yield and purity was conducted using NanoDrop, Qubit and gel electrophoresis.

Sanger sequencing of the amplified probes was performed in both directions using the Mix2Seq-Kit NightXpress (Eurofins Scientific SE, Luxembourg). Quality-controlled consensus sequences were aligned together with existing markers using MUSCLE (v3.8.1551)[91] and trimmed for phylogenetic analysis via trimAl (v1.5)[92]. IQ-TREE (v2.4.0)[93] was accelerated for maximum likelihood tree generation by using the GTR + G model with 1000 bootstrap replicates. The BASH script can be found on GitHub/Zenodo[94]. GenBank accession numbers for newly created ITS markers and all reference sequences used for the analysis are documented in Supplementary Table 1.

### DNA extraction and whole-genome sequencing

For the extraction of high molecular weight (HMW) gDNA the Nanobind Plant Nuclei Big DNA Kit from Circulomics (PacBio, CA, USA) was used according to the manufacturer's instructions (v0.18; 10/2018). In brief, plant nuclei were pre-isolated from snap frozen plant tissue, which was ground into fine powder in $N_{2(l)}$ and submerged in freshly prepared nuclear isolation buffer (NIB). The homogenate was gravity filtered through five layers of Miracloth and subsequently centrifuged at 3000 ×g (*C. major*) or 2600 × g (*C. minor*; *C. rosea*), according to the estimated genome sizes, to obtain a homogeneous nuclei suspension, which was washed three times using NIB. After the enzymatic removal of proteins and RNA (Proteinase K, RNase 1), the nuclei were incubated for lysis in a nuclear lysis buffer. Free nucleic acids were bound to magnetic nanodisks, washed three to five times, and eluted in the elution buffer supplied with the kit. Because of low initial yields, DNA extractions from *C. rosea* were repeated and included the same protocol for nuclei pre-isolation as described above but relied on a self-made nuclear envelope lysis buffer (5 mM HEPES, 1.5 mM $MgCl_2$, 0.2 mM EDTA, 0.5 mM DTT, 300 mM NaCl, pH = 7.9) and a standard phenol/chloroform/isoamyl-alcohol (PCI) extraction protocol instead of the magnetic disk-based extraction. Chloroform phase separation was repeated three times to ensure thorough removal of residual phenol. DNA was precipitated in 0.3 M NaCl and isopropanol at room temperature (RT) for 1 h and finally eluted in 50 μL nuclease free water. DNA quality and quantity were determined via measurement on a NanoDrop photospectrometer and Qubit fluorometer. Gel electrophoresis and a fragment analyzer (Thermo Fisher Scientific, MA, USA) were used to assess DNA integrity and fragment length. DNA was stored at 4 °C until sequencing. Detailed results can be found in Supplementary Data 9.

Library preparation and DNA sequencing was conducted by the Next Generation Sequencing Facility at Vienna BioCenter Core Facilities (VBCF, Vienna, Austria) using a PacBio Sequel II instrument

(Pacific Biosciences of California, Menlo Park, CA, United States) and following library preparation methods for continuous long-reads (CLR). Sequencing/raw read statistics are provided in Supplementary Data 1.

## HiC extraction and sequencing

For DNA crosslinking, finely ground leaf powder was suspended in 50 mL 1% paraformaldehyde in 1× phosphate buffered saline solution (PBS; pH = 7.4) and incubated for 20 min at RT under periodic vortexing. In order to stop the fixation process, 0.5 g glycine was added and the homogenate was incubated at RT for 15 min. Samples were washed three times with PBS and centrifugation at $1000 \times g$ for 1 min. Approximately 10 g of crosslinked leaf powder were transferred to a clean 15 mL tube, frozen, and shipped to Phase Genomics (Phase Genomics, WA, USA) for further extraction and sequencing. DNA extractions, proximity ligations, library preparations and sequencing were performed by Phase Genomics (WA, USA).

## Genome assembly

First draft assemblies were carried out via FALCON-kit (v1.8.1) and FALCON-unzip (v1.3.7)[23] using all raw reads longer than 500 bp. A second polyploid assembly was performed with Canu (v2.1.1)[26] to acquire error corrected reads for ploidy estimations and to check for haplotype confusions. Details regarding configuration parameters can be found on GitHub/Zenodo[94]. All quality assessments were conducted using the tools BUSCO (v5.2.2)[24] with the orthologous dataset eudicots (odb10). For *C. major*, duplicated primary contigs with genomic regions of high heterozygosity and artefacts were removed from the haploid primary assembly using purge_haplotigs (v1.1.2)[28]. PacBio long-reads were first aligned to the draft assembly with minimap2[41] and read-depth based identification of purge candidates was performed using low, middle, and high cutoff values of 30, 145, and 300, respectively. Repeat annotations were further used to assist in the haplotig reassignment process. The curated primary contigs and associated haplotigs (including reassigned sequences) were phased via FALCON-phase (v1.2.0)[29] by utilizing HiC proximity data.

For scaffolding, we created a comprehensive pipeline taking the predominant pseudo-haplotype assembly and HiC reads as input. The process of contig partitioning and scaffolding was accomplished with ALLHiC[30], a method to assemble allele-aware chromosomal-scale genomes of polyploid plants. The final manual curation of our genome including re-mapping of HiC reads was conducted using a modified version of the software suite Juicer[31,32]. The pipelines are written in BASH and are deposited on GitHub/Zenodo[94]. Intermediate results, HiC contact maps and overall statistics are provided in Supplementary Data 2–4. Final assemblies are deposited at NCBI and on Figshare[95].

## Repeat modeling and masking

For repeat identification, a de novo repeat library was created via RepeatModeler (v2.0)[96] used for identification and classification of TEs based on the draft genomes and, finally, sequences were masked via RepeatMasker (v4.1.2)[97]. For *C. major*, a pipeline written in BASH handled the serial annotation of repeats using several libraries with different levels of priority, including simple/low complexity reads, well-curated repeats from Ensemble nrTEplants (v0.3)[98], and classified and unknown *Clusia*-specific repeats via de novo library. The resulting repeat orders and superfamilies were reclassified[99]. Custom scripts were created or adapted to calculate Kimura 2-Parameter substitution levels and intronic repeat lengths via 'bedtools intersect (v2.31.0)' (calcDivergenceFromAlign.pl, ClusiaDB.repeats.R). Results are stored on Figshare[95].

## Gene annotation

Ab initio gene model prediction was done on softmasked draft genomes via BRAKER2 (v2.1.5-m)[100] using curated proteins from OrthoDB of all land plants (Viridiplantae). For *C. major*, several pipelines handled the preparation of various extrinsic sources and structural gene annotation of both the primary assembly and associated haplotigs. RNA-seq based preparation was done using HISAT2[101] and StringTie[102] for genome-guided transcript assembly in addition to Trinity[103] for stranded de novo transcriptome assembly. Full-length transcripts were further processed using PASA[104] and ORF prediction was conducted via TransDecoder (v5.5.0). Homology-based gene prediction was accomplished via GenomeThreader using protein sequences from Viridiplantae. EVidenceModeler (EVM)[105] was used to compute weighted consensus gene annotations from all combined data sources. UTR annotations and alternatively spliced isoforms were added to the predicted gene models using PASA. About 10,000 models were filtered out based on minimum protein length (>= 50 bp), spanning gaps, or hits to known transposable elements by utilizing the algorithm of funannotate (v1.8.15). More detailed instructions and all pipelines in this section can be found on GitHub/Zenodo[94].

Functional gene annotation was conducted on the basis of a custom BASH pipeline complemented with Python scripts making use of the publicly available databases uniport swissprot, interpro and eggnog. Gene models were further filtered based on functional annotation: (a) keep all PASA models having any transcript evidence, (b) keep valid curated functions (uniprot hits from non-PASA models), and, from the rest, (c) keep only those models having protein similarity (pident) >40%, query and subject coverage >50% based on Viridiplantae orthologs from eggnog. The final annotation files (GFF3, exons, CDS, translated proteins) are deposited on Figshare[95].

## Pseudogene identification

High confidence genes were selected from the primary assembly based on similarity ( > 40%) and query/subject coverage ( > 50%) to manually curated proteins from uniport swissprot. Based on these parent genes, the identification of pseudogenes were conducted using an altered version of PseudoPipe[36]. Overlaps to annotated gene models were intersected via bedtools (v2.31.0) and further validated for downstream analysis. Both the pseudo-haplotypes and CANU assembly were mapped to the pseudogenized genes to screen for haplotype confusions - particularly within CAM-related loci. Pseudogenic regions lacking CANU alignment or exhibiting large deletions were filtered out as evidence of erroneous mixing, collapsing, or switching between haplotypes. Data integration and filtering were done via R as part of the *Clusia* database (ClusiaDB.pseudogenes.R). A list of identified pseudogenes is provided in Supplementary Data 5.

## Flow cytometry

Fresh leaf material of both, the *Clusia* spp. and standard samples, were co-chopped in Otto´s isolation buffer[106–108]. The resulting isolate was filtrated through a nylon mesh (30 μm pore size) and incubated at 37 °C with RNase (final concentration 0.15 mg/mL, *Sigma-Aldrich GmbH*) for 30 min. Propidium iodide supplemented Otto´s staining buffer was added and measurement took place after an about one hour incubation period at refrigerator temperature. *Pisum sativum* (4.42 pg/1 C)[109], *Solanum pseudocapsicum* (1.295 pg/1 C)[110], and *Capsicum flexuosum* (7.44 pg/1 C) served as internal standard organisms. For measurement two Partec/Sysmex flow cytometers were used, either a CyFlow ML (100 mW, 523 nm, Cobolt Samba Laser, Cobolt AB, Stockholm, Sweden), or a CyFlow SL (30 mW, 532 nm laser, Partec, Muenster, Germany). The *Clusia* spp. C-values were calculated from the peak´s mean fluorescence intensities ( = peak position on the x-axis, FI) as follows:

$$1C - value_{Object} = (G_{0/1}\,FI\,peak\,position_{Object}$$
$$/G_{0/1}\,FI\,peak\,position_{Standard})*1C - value_{Standard} \quad (1)$$

## Karyotyping

Chromosome numbers were established based on the analysis of chromosome spreads (several preparations and at least five intact spread cells): Actively growing root meristems were pretreated with 8-hydroxyquinoline for 3 h at RT and 3 h at 4 °C, fixed in 3:1 ethanol:glacial acetic acid, and stored at −20 °C. An enzymatic digestion method was used to prepare chromosomal spreads[111]. Chromosomes were stained with DAPI (4′,6-diamidino-2-2phenylindole; 2 ng/μL) and mounted in Vectashield antifade medium (Vector Laboratories, Burlingame, USA). Preparations were analyzed with an AxioImager M2 epifluorescent microscope (Carl Zeiss, Vienna, Austria) equipped with a CCD camera to capture images using AxioVision (v4.8) (Carl Zeiss, Vienna, Austria). Selected images were used to prepare karyotypes in Corel PhotoPaint X8.

## Genome profiling and read-based ploidy level estimation

The basis for this analysis was built on the error-corrected trimmed long-reads assembled from Canu[26]. K-mer frequencies were counted using a specific version of the tool KMC[112]. We counted the 21-mer coverages between the range of 1–1,000,000 and extracted the histogram values for GenomeScope2 and smudgeplot[25]. For smudgeplot, the k-mer databases were reduced to a range between lower (filter erroneous k-mers) and upper bounds (filter highly repetitive sequences), visually gathered by inspecting the plots from GenomeScope. For *C. major* and *C. minor*, the lower bound value from automatic detection was overruled and decreased to L = 20 because we have sufficient coverage. For *C. rosea*, a lower limit of L = 10 was set as a trade-off between covering the heterozygous peak while potentially including more erroneous k-mers.

## Transposon-mediated subgenome separation

Kmer-based mapping of subgenomes in the putative allopolyploid primary assembly were conducted using the "Kmer-based-Subgenome-Mapping" algorithm[33] with a k-mer size of 13 and a cutoff of 100 for *C. major* and 150 for the positive control of allotetraploid cotton (*G. hirsutum*). Homoeolog pairs were set as colored in Fig. 2. For comparative genomics, we introduced the arbitrary groups H1 (homoeolog 1) and H2 (homoeolog 2) by assigning each syntenic chromosome pair based on the size affected by the transposons. This facilitates the handling of homoeologs in further analyses but does not allow global subgenome-dependent interpretations.

## Synteny-based WGD analysis and $K_S$ age distributions

The analysis was conducted using the tools bundled in wgd (v2)[113]. Divergence time estimation among seven species was done using PAML/MCMCTREE with LG+Gamma model for protein alignments. Fossil calibrations and time restrictions were taken from TimeTree[114]. The results were incorporated in Supplementary Figs. 14 and 15. More detailed instructions and all pipelines can be found on GitHub/Zenodo[94].

## Comparative genomics

Orthogroup clustering and species tree inference was conducted with OrthoFinder2[39] on 51 plant species across the clade of vascular plants. Coding sequences (CDS) and their translated protein sequences were downloaded from NCBI Genbank/RefSeq, Phytozome or genome specific repositories. The groups H1/H2 of *C. major* were treated as different species in order to later distinguish between homoeologs and other paralogs[115]. A comprehensive list of species/accessions used for orthogroup sampling as well as results of all comparative genomic analyses are included on Figshare[95].

Syntenic orthogroups and pangenes among *Clusia* species were inferred via GENESPACE (v1.2.3)[40] using *Vitis vinifera* as outgroup and H1/H2 as separate species to connect positional homoeologs of the putative subgenomes of *C. major*. A second run was performed without outgroup and group separation for within species synteny and riparian plots. Configuration parameters are documented on GitHub/Zenodo[94] (script: genespace.R). In brief, GENESPACE compares protein similarity scores into syntenic blocks using MCScanX[116] and again uses Orthofinder[39] to search for orthologs/paralogs within synteny constrained blocks. The forked and slightly modified software was also utilized to create riparian plots (ClusiaDB.synteny.R).

Genome-wide nucleotide alignments were performed with minimap2[41] using the pseudo-haplotypes and CANU contigs of *C. major* as well as the draft genomes of the other species as individual sources against the primary chromosome level assembly as reference. Modes asm5 and asm10 were defined for the contigs/haplotypes and *C. minor/C. rosea*, respectively, to account for different extents of sequence divergence. MSAs of selected genes/promoters of interest were generated using the ClustalOmega algorithm implemented in the R package msa. For plotting with ggplot2 the R packages msavisr and ggmsa were used with slight modifications. Alignments are provided on Figshare[95].

## *Cis*-regulatory motif analysis

Promoters of CAM-related genes for the three *Clusia* species were included in the motif analysis. Promoter sequences were defined as 1.5 Kb upstream of the predicted transcription start site (TSS). TSS were predicted using TTSFinder[117]. Low complexity regions and simple repeats were hard-masked. Promoters were scanned for transcription factor binding motifs associated with circadian oscillations using the FIMO algorithm implemented in the MEME suite[118] with a 0-order Markov background model generated from a random subset of genes. Position weight matrices (PWMs) for the motifs from *A. thaliana* were downloaded from JASPAR (v9) or PlantTFDB (v4.0). A list of identified motifs is provided in Supplementary Data 8.

## *Clusia* panomics database

A pipeline written in R (v4.2.1) via the package tidyverse handles the resolution of homoeologs, the interspecies pangenome, and selection of CAM-relevant gene families, to account for different genome/annotation qualities, by integrating various genomic data sources and relevant literature. In addition to orthogroups, syntenic pangenes, and functional annotation, we used contig phasing information and sequence alignments to define positional homoeologs and homoeologous groups (syntenic orthogroups having one reference gene) within the primary chromosome level assembly and further map the alleles (haplotypes) of *C. major* and the homologs of *C. minor* and *C. rosea* (handles "non-syntenic"/unresolved genes due to short sequences of draft genomes, gene artefacts, etc.). Resolved/unambiguous gene relationships are then propagated to the next higher level (groups and gene families). Gene families are defined from orthogroups having at least half of all sampled species present to handle lineage/species-specific likely paralogous orthogroups containing neofunctionalized or, more likely, solely fragmented genes.

Automatic selection of CAM-relevant gene families is based on multiple identifier (EC-number, Arabidopsis gene loci, and UniprotID) gathered by inspecting relevant literature and stored along other metadata in a dedicated settings file (metadata.xlsx). The R script (ClusiaDB.init.R) creates the main dataset (data.tsv) which builds the basis to integrate all further multiomics data (see below). The entire pipeline can be executed by running the script ClusiaDB.R.

The *Clusia* database is structured so that certain raw data are processed in twelve R subscripts (notation: ClusiaDB.*.R). These subscripts then organize their own plots and feature matrices like for ClusiaDB.pseudogenes.R: Fig. 3c, Supplementary Data 5. These features are then further integrated within the main data frames and are described elsewhere. Supplementary Table 4 illustrates how the raw datasets[95] are incorporated into the programmatic generation of

figures and tables. The commented R code is available in on GitHub and Code Ocean, where it is built into a dedicated compute capsule.

Processing and visualization of data showing diploidization is organized in a separate R subscript (ClusiaDB.diploidization.R). The script calculates genome-wide z-scores and creates a feature/data matrix of intron length, repetitive elements within introns, pseudogenized gene models, and number of genes per homoeologous group within each gene family, which is incorporated in Fig. 4a, Supplementary Fig. 7, Supplementary Data 6.

The signals of diploidization were used to perform GO term enrichment/over representation analyses (ORA). Inputs were selected based on all CAM-relevant genes and subsets for the individual enrichments "Genic diploidization", "Intron/repeat length", and "Conserved genes", respectively. The orthologs of *A. thaliana* from gene families matching those criteria were used for each ORA with clusterProfiler (v4.12.6)[119]. The org.At.tair.db was utilized as background model with BH adjustment method, 0.05 for $q$-value, and 0.01 as $p$-value cutoff. The script ClusiaDB.enrichment.R handles the calculation and visualization incorporated in Supplementary Fig. 6.

Gene counts per family for each species or homoeologous group are based on the diploid outgroup *Vitis vinifera* L. according to the formular:

$$\text{Gene counts per family for each species or homoeologous group} = 1.0 - (\#Outgroup - \#SpeciesOrGroup) / \#Outgroup \quad (2)$$

Over dispersed gene families having more than 8*ploidy genes per homoeolog/species were filtered out (e.g. transposase) and the outgroup needs to have at least one gene present (script: ClusiaDB.counts.R).

### Plant cultivation and physiological phenotyping

The plants were cultivated under a common garden environment featuring a median temperature of 28 °C during daytime and 22 °C during nighttime, with a median day air humidity of 60% and a global irradiation of 250 PPFD (photosynthetically active photon flux density 400–700 nm) using solar spectrum simulating LEDs (Valoya Inc.). The plants grew in pots with a water supply of approximately 20% water volume and a soil composition of common gardening humic soil (Kranziger), Perlite, Seramis and quarz sand (1:1:1:0.5). Detailed conditions for cultivation as well as all further experimental parameters are listed in Supplementary Table 3.

To evaluate physiological phenotypes, titratable acidity and gas exchange experiments were conducted under highly controlled conditions in a climatic chamber (Fytochamber, PSI, CZ). Microclimatic parameters followed a diurnal cycle as measured in the OpenGreenhouse experiment (i.e., length of photoperiod, average day/night temperatures and relative humidities). Ambient light conditions were set to 300 PPFD at sampling/measuring height and monitored using an Apogee 520 full spectrum quantum sensor to apply just saturating light. We used 50–70 cm high, 5 years old mature plants grown in 24 cm pots from the same collection as the sequenced samples. Soil water content of each individual was measured continuously using ML3 theta probes (Delta T Systems, Cambridge) to facilitate precise daily watering. The exact conditions including ambient VPD (water vapor pressure deficit [kPa]) and DLI (daily light integral [mol m$^{-2}$ d$^{-1}$]) are listed in Supplementary Table 3. Raw data of physiological measurements are deposited on Figshare[95].

### Titratable acidity experiment

Three biological replicates per species were incubated for 6 weeks at 25–30% soil water volume until first sampling took place (well-watered). Irrigation of the same individuals was then gradually reduced to slowly reach a soil water volume of exactly 10%. After two weeks, the

second sampling was performed over a full 24-h cycle (drought conditions). Depending on species and leaf size/thickness, two to four leaf disks from three to five leaves were pooled per sample/replicate and harvested at five timepoints (9 pm, 6 am, 9 am, 6 pm, and 9 pm). A total of 90 samples were then processed using standard hot water extraction and titrated in three technical replicates via photometric assays[120]. 40–50 mg of ground leaf material was incubated in 500 µL MilliQ water at 80 °C for 30 min while shaking. After centrifugation at 21,000 xg for 8 min, the supernatant was collected. Titratable acidity was measured using bromothymol blue (Carl Roth) as a pH indicator: 10 µL of extract was mixed with 90 µL degassed MilliQ and 4 µL bromothymol blue (1 mg/mL stock). Samples were titrated with 10 mM NaOH in 1 µL steps. After each step, absorbance at 445 nm and 615 nm was recorded using the Multiskan plate reader (Thermo Scientific). Based on the 615/445 ratio of buffer standards (pH 4.6–7.8), the pH of analyzed extracts was determined. From the volume of NaOH required to titrate extracts to pH 7.0, the titratable acidity was calculated. Statistical testing was performed using two-way ANOVA and Tukey HSD. Data processing and visualization were done via R (ClusiaDB.experiments.R). The results and environmental conditions were incorporated in Fig. 1c and Supplementary Fig. 1.

### Plant leaf stomatal conductivity and gas exchange

In another phenotyping experiment, gas exchange (net $CO_2$ assimilation and stomatal water conductivity) was measured under identical drought conditions by infrared gas analysis (IRGA), using the WALZ GFS3000 (Walz GmbH, DE). Leaves were enclosed in a leaf clamp chamber where $CO_2$, T°C, rh (relative humidity %), and irradiation were controlled and $CO_2$ net assimilation rate (calculated using the following equation) as well as $H_2O$ evaporation (stomatal water conductance, $GH_2O$) were monitored and logged over a period of more than 24 h at a frequency of 5 min between recordings.

$$CO_2 \text{ net assimilation rate} = CO_2 \text{ assimilation} - CO_2 \text{ respiration} \quad (3)$$

For zero offset corrections, we applied a triple repetition of ZPIrga and ZPCuv, and performed 3 measurements with empty cuvette before and after each measurement run to record initial and emerging deviations. Before the experiment, the gas analysis device was sent to the manufacturer for complete checkup and recalibration. For *C. rosea*, we exhibited artefacts like oscillating and negative values for water stomatal conductance that are likely caused by low evapotranspiration of the upper leaf surface in combination with large differences of air water content and relatively small measurement area (Lycor and WALZ online manuals). Therefore, all plots were averaged over 5 values showing measuring points approximately every 20 min and negative values of $GH_2O$ were manually corrected to zero. Data processing and visualization were done via a custom R script (ClusiaDB.experiments.R).

### Open greenhouse experiment

Plant material for experimentation was generated by producing a total of 48 cuttings from mother plants. *C. major*, *C. minor*, and *C. rosea* were propagated in a clonal fashion from those plants, which we used for DNA extractions and genome sequencing. Cuttings were cultivated in the greenhouse for eight months until all plants reached at least a four-leaf stage, and they were then randomly divided into two groups. The experiment was conducted during a sunny summer week in July 2021. For one week, one group was exposed to direct sunlight. There was no precipitation, and the plants were not irrigated. The other group was placed beneath a shaded cultivation table and each pot received 50 mL of water each day, corresponding to 60% field capacity (approx. 30% soil water). Via soil moisture, photosynthetic active radiation (PAR), and temperature loggers (placed in the shade), connected to Arduinos and a Raspberry Pi, we continuously monitored environmental parameters (GitHub/Zenodo[94], Figshare[95]). Chlorophyll fluorescence (PSII

quantum yield, PSII NPQ non-photochemical quenching) as well as ambient light radiation (PPFD), ambient temperature (°C), relative air humidity (rh%) and leaf differential temperature were monitored on site every other hour using the MultispeQ V2.0 (*PhotosynQ Inc.*). We then selected six uniform cuttings from each plant species based on DLI (three replicates per group). On July 15th 2021, we started sampling leaf tissue on four time points across a 24-h period, at 4 am, 8 am, 1 pm, and 7 pm, respectively. We used sterile punch pliers with a 5 mm diameter to punch four holes into one leaf of each plant. At each time-point a new leaf section (with respect to the main nerve) was used to reduce the amount of injury related signals. The leaf tissue was transferred into 2 mL centrifuge tubes and was immediately snap-frozen in liquid nitrogen and stored at −80 °C for subsequent multiomics analyzes.

### Transcriptomics
Plant material was ground in liquid nitrogen and ~50 mg of leaf powder per sample were used for RNA extraction. All solutions were prepared with DEPC treated water. RNA was extracted using a phenol:chloroform phase separation and LiCl precipitation-based protocol[121] with an adapted extraction buffer (2.5 M NaCl, 2.5% PVP, 100 mM Tris, 25 mM EDTA, 2.5% β-mercaptoethanol and 2% SDS)[122]. RNA concentration and quality were determined via spectral measurements on a Nanodrop-2000 UV spectrophotometer (Thermo Fisher Scientific) and via a fluorimetric assay on an Qubit ™ 4 Fluorometer (Thermo Fisher Scientific) with a RNA High Sensitivity Assay kit (Invitrogen by Thermo Fisher Scientific), according to the manufacturers' instructions. Total RNA was sent to the Vienna Biocenter Core Facilities for further QC, purification using "AMPure" beads, RNA library preparations, and 150 bp paired-end Illumina sequencing on a NovaSeq S4 system.

Sample preprocessing was conducted using fastp (v0.23.2)[123] with the parameters "-q 30 -l 80 --detect_adapter_for_pe". The trimmed paired-end reads for each sample were aligned to repeat-masked *Clusia* assembly. STAR (v2.7.10b)[124] and RSEM (v1.3.3)[125] with custom parameters accomplished read alignment and allele-aware estimation of normalized transcripts per million (TPM) (GitHub/Zenodo[94], Figshare[95]). Data matrices for downstream analyses were constructed and integrated via custom scripts in R (ClusiaDB.transcriptomics.R).

### Proteomics
Proteins were extracted from MCW metabolite extraction pellets[126]. Total protein was dissolved in 6 M urea and 5% SDS and protein concentration was determined using the bicinchoninic acid assay (BCA method)[127]. Proteins were pre-fractionated by 1D SDS-PAGE (40 µg of total protein were loaded onto the gel), trypsin digested, and desalted using Pierce C18 tips (Thermo Fisher Scientific). Purified tryptic peptides, dissolved in 0.1% formic acid (v/v), were separated via reversed-phase HPLC using a Thermo Scientific Dionex Ultimate 3000 nano RPLC system connected to a benchtop Quadrupole Orbitrap mass spectrometer (Q-Exactive Plus). The LC eluent entered the mass spectrometer via an Easy-Spray ion source at 1.9 kV. The LC employed a heated Easy-Spray analytical column (PepMap RSLC C18) at 55 °C, with a flow rate of 300 nL/min. A two-hour LC gradient method was applied, transitioning from 5% to 50% buffer B (79.9% ACN, 0.1% formic acid, 20% ultra-high purity MilliQ water), followed by a rapid shift to 80% buffer B over 5 min. Buffer A contained 0.1% formic acid in MilliQ water. Mass spectra, acquired in positive ion mode using a top fifteen data-dependent acquisition (DDA), had resolutions of 70,000 (full MS) and 17,500 (MS/MS) at m/z 200. For MS/MS fragmentation, normalized collision energy (NCE) was stepped at 27%, 30%, and 33%, with dynamic exclusion at 60 s. Unassigned and +1, +7, +8 and > +8 charged precursors were excluded. Raw mass spectra were correlated against our *Clusia* database containing annotations and decoy/contaminated sequences using the SEQUEST algorithm in Proteome Discoverer version 1.3 (Thermo Fisher Scientific). Peptides were considered a

significant hit when the peptide confidence was high, which is equivalent to a false discovery rate (FDR) of 1%, and the Xcorr threshold was established at 1 per charge (2 for +2 ions, 3 for +3 ions, etc.). Variable modifications were set to acetylation of the N-terminus and methionine oxidation, with a mass tolerance of 10 ppm for the parent ion and 0.8 Da for the fragment ion. Two missed or non-specific cleavages were permitted. There were no fixed modifications, as dynamic modifications were used. The identified proteins were quantitated based on total ion count and normalized using the normalized spectral abundance factor (NSAF) strategy[128]. The reference database as well identified protein quantities are provided on Figshare[95]. Data matrixes for downstream analyses were constructed, integrated and visualized via custom scripts in R (ClusiaDB.proteomics.R).

### Metabolomics
Polar metabolites were extracted from 15 mg of frozen plant tissues using ice-cold methanol, chloroform and water (MCW; 2.5:1:1 (v/v))[126]. 5 µL of 10 mM pentaerythritol (PE) and phenyl-β-D-glucopyranoside (PGP) were added as internal standards. After ultrasonication (20 min, 4 °C) and centrifugation (4 min, 4 °C, 14,000 g) supernatants were transferred to a new 1.5 mL tube. This procedure was repeated twice, supernatants were combined, and 350 µL of water were added to induce phase separation. After centrifugation, the upper phase was transferred to a new 1.5 mL tube and dried by vacuum-centrifugation for 5 h. The dried pellet was dissolved in 20 µL of methoxyamine hydrochloride solution (40 mg/mL in pyridine) and incubated at 30 °C for 90 min with shaking. MSTFA (80 µL) was then added and the mixture incubated at 37 °C for 30 min with vigorous shaking. Prior to GC–MS analysis, an even-numbered alkane solution was spiked into each sample to generate retention indices[129]. An Agilent 7890B gas chromatograph equipped with a LECO Pegasus BT-TOF mass spectrometer (Leco) was used to perform metabolite analysis. 1 µL of derivatized sample was injected into the heated (240 °C) Split/Splitless inlet equipped with an ultra-inert single tapered glass liner with deactivated glass wool (Agilent Technologies) and a split ratio of 1:15 was used. Separation of metabolites was performed on a Restek Rxi-5Sil MS column (length: 30 m, diameter: 0.25 mm, thickness of film: 0.25 µm) using helium as carrier gas with a flow rate of 1.4 mL per minute and a temperature gradient starting with an initial oven temperature of 70 °C held for 1 min, and ramped up with a rate of 11 °C per minute until reaching 340 °C, held for 10 min. After an acquisition delay of 245 seconds, 15 spectra per second were collected with a detector voltage of 2287.5 V in a mass range of 50–600 m/z. A standard solution of n-alkanes (C10-C40) (Sigma Aldrich) was used to calculate the retention index of metabolites. Processing of chromatographic data and annotation of metabolites was performed according to the metabolomics standards initiative (MSI) using ChromaTOF (v5.55.29.0.1187, LECO Corporation) and MS-DIAL (v4.9.221218)[130]. Relative abundances of metabolites were calculated by normalizing peak area to internal standard (PGP) and weight of extracted material for further statistical analysis. Diurnal fluctuations in metabolite pools were analysed by the loess method using R (v4.3.1). A canonical analysis of principal coordinates (CAP) was performed in Primer6 (v6.1.13, PRIMER-E Ltd.) with the PERMANOVA extension (v1.0.3) to identify metabolites mainly discriminating the groups, defined by condition and species. Therefore, a resemblance matrix by Bray-Curtis dissimilarity was created from squareroot-transformed relative abundances in advance. Integrals of annotated peaks of targeted metabolites are provided on Figshare[95].

### Starch assay
Fully developed primary leaves were harvested at the end and the beginning of the photoperiod. Leaves of at least three different individual plants were pooled into one sample. The frozen leaves were ground into a fine powder and 5 mL of 80% methanol was added to

~150 mg tissue. Samples were refluxed at 70 °C for 40 min to remove soluble metabolites and then pelleted by centrifugation for 2 min. The pellet was sonicated and washed four times with water to remove any remaining soluble metabolites. The samples were boiled in 1.25 mL of 0.1 M sodium acetate buffer (pH=4.5) for 30 min. After cooling an equal volume of 0.1 M sodium acetate buffer was added, containing amyloglucosidase (5 U activity in total volume) and α-amylase (0.5 U activity in total volume). For starch digestion, the samples were incubated at 40 °C for 18 h and subsequently centrifuged for 10 min[131]. The supernatant was collected for colorimetric glucose content determination[131,132]: 1 mL of the collected supernatant was combined with 0.025 mL 80% phenol and 2.5 mL of concentrated sulfuric acid. The samples were left to stand for 10 min then mixed and heated for 20 min at 25 °C. Absorbance was measured at 490 nm and the glucose concentration determined via reference to a standard curve consisting of (wheat) starch.

## Reporting summary

Further information on research design is available in the Nature Portfolio Reporting Summary linked to this article.

## Data availability

The DNA/RNA sequencing data generated in this study have been deposited at NCBI under BioProject PRJNA1334428. ITS marker sequences are available in GenBank under accession PX225522 - PX225525. gDNA sequencing libraries (PacBio continuous long-reads, Illumina HiC short-reads) are publicly available within the sequence read archive (SRA) under BioProject PRJNA1183765. The primary chromosome-level assembly of *C. major* (predominant/principal pseudo-haplotype) is deposited under the GenBank accession GCA_056099205.1 [https://www.ncbi.nlm.nih.gov/datasets/genome/GCA_056099205.1/]. The genome for the alternative pseudo-haplotype of *C. major* can be accessed via GCA_056098825.1 [https://www.ncbi.nlm.nih.gov/datasets/genome/GCA_056098825.1/]. The draft genome assemblies of *C. minor* s.l. and *C. rosea* (contig-level) are linked under the BioProject PRJNA1183765. mRNA sequencing data were deposited in SRA under the project accession PRJNA1197736 and in NCBI's Gene Expression Omnibus accessible through GEO Series accession number GSE290226. The mass spectrometry proteomics data have been deposited to the ProteomeXchange Consortium via the PRIDE partner repository with the dataset identifier PXD061385. GC/MS data of targeted metabolites are uploaded to MetaboLights under project accession MTBLS14075. The assembled genomes and annotations as well as (intermediate) pipeline results and multiomics data used for analysis in this article, including raw data of figures and additional work to support the findings, can be found on Figshare [https://doi.org/10.6084/m9.figshare.27599406][95]. An interactive genome browser (https://apps.pph.univie.ac.at/jbrowse/) provides access to the predominant chromosome assembly of *C. major* including gene, pseudogene, and repeat annotations as well as HiC proximity signals and alignments to CANU and the draft assemblies of *C. minor* s.l. and *C. rosea*. Source data are provided with this paper.

## Code availability

The pipelines, scripts and code files used for analyses in this study have been deposited on Zenodo [https://doi.org/10.5281/zenodo.19666426][94], and are also available at GitHub [https://github.com/hanneskramml/Clusia]. A dedicated compute capsule to assess the *Clusia* panomics database is published on Code Ocean [https://doi.org/10.24433/CO.2105665.v2][133].

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

## Acknowledgements

The computational results of this work have been achieved using the Life Science Compute Cluster (LiSC) of the University of Vienna, with special thanks to Thomas Rattei and Florian Goldenberg. The gDNA and mRNA sequencing was performed by the Next Generation Sequencing Facility at Vienna BioCenter Core Facilities (VBCF), member of the Vienna Bio-Center (VBC), Austria. We acknowledge Carmen Czepe, Volodymyr Shubchynskyy, and Andreas Sommer (VBCF) for their work on sample purification and sequencing-related discussions. HiC ligation and sequencing services were provided by Phase Genomics. We acknowledge the excellent work of our gardeners Andreas Schröfl and Thomas Joch. We appreciate Wolfgang Wanek and the staff of the Tropical Research Station La Gamba, the place of inspiration for this study. We specifically acknowledge the work of Christian Lexer – we will always treasure your enthusiasm and support. H.M.K. and S.P. were financially supported by the doctoral program MENTOR (Molecular Mechanisms to Improve Plant Resilience), funded by the Austrian Science Fund (FWF) under project number DOC 111. P.C. was financially supported by the Austrian Science Fund (FWF) under project number DOI 10.55776/I5234. Open access funding provided by University of Vienna.

## Author contributions

W.W., H.M.K., and J.H. conceived the study and jointly wrote the manuscript. H.M.K. and J.H. extracted HMW gDNA and designed the genome sequencing approaches. H.M.K. assembled and annotated the genomes, performed genomic analyses, and took care of data integration across experiments. G.B., H.M.K., and A.D. performed plant physiological phenotyping. H.M.K., J.H., T.S., L.F., S.P., F.S., F.T., A.K.B., W.W. and G.B. designed and conducted greenhouse experiments. M.I., S.P., and J.H. extracted RNA from *Clusia* species, J.H. and H.M.K. performed transcriptome data analyses. P.K., S.P., L.F., M.B., and C.P. extracted metabolites and performed data acquisition. C.P. and J.H. analyzed metabolomics data. F.F., L.A.S., P.C., A.G., and T.S. performed proteomic data acquisition. J.H., H.M.K., and I.P. analyzed proteomic data. F.S. performed starch assays. H.W.S., E.M.T., and T.S. conducted genome size estimation experiments and karyotyping. Z.W. and H.M.K. investigated *cis*-regulatory elements. A.D.B. revised the taxonomy and documented the plant species in the herbarium. M.K. and H.M.K. conducted molecular barcoding. U.L. provided the living plant collection. H.M.K., J.H., G.B., and C.P. plotted all figures. H.W.S., S.W., O.P., O.S., U.L., W.H., and W.W. provided infrastructure essential for experimentation and data acquisition. All authors provided valuable feedback on the manuscript and approved of the final version for publication.

## Competing interests

The authors declare no competing interests.

## Additional information

[1]Molecular Systems Biology (MOSYS), Department of Functional and Evolutionary Ecology, University of Vienna, Vienna, Austria. [2]Department of Plant Microbe Interactions, Max Planck Institute for Plant Breeding Research, Cologne, Germany. [3]Terrestrial Ecosystem Research, University of Vienna, Vienna, Austria. [4]Department of Botany, Natural History Museum Vienna, Vienna, Austria. [5]Department of Botany and Biodiversity Research, University of Vienna, Vienna, Austria. [6]Mass Spectrometry Unit, Research Support Facilities, University of Vienna, Vienna, Austria. [7]Vienna Metabolomics Center (VIME), Vienna, Austria. [8]Department of Neuroscience and Developmental Biology, University of Vienna, Vienna, Austria. [9]Institute of Botany, Technical University of Darmstadt, Darmstadt, Germany. [10]Institute for Evolution and Biodiversity, University of Muenster, Muenster, Germany. [11]Environment and Climate Research Hub (ECH), University of Vienna, Vienna, Austria. [12]Research Network Health in Society, University of Vienna, Vienna, Austria. [13]Artificial Intelligence for Personalized Nutrition (AIPN), University of Vienna, Vienna, Austria. [14]These authors contributed equally: Hannes M. Kramml, Johannes B. Herpell. ✉e-mail: wolfram.weckwerth@univie.ac.at

