## [Peer Review file · Nature Communications]

***Clusia* genomes shed light on the evolution and diversity of CAM physiotypes**

Corresponding Author: Professor Wolfram Weckwerth

Version 0:

Reviewer comments:

Reviewer #1

(Remarks to the Author)

In the manuscript entitled “*Clusia* genomes shed light on the evolution and plasticity of CAM,” the authors describe the sequencing and analysis of *C. multiflora*, *C. minor*, and *C. rosea* genomes, transcriptomes and proteomes to better understand the different types of CAM photosynthesis. The authors develop a compelling hypothesis that it is the return to diploidy that is driving the CAM innovation in this genus. However, the underlying genomes and data are not available to evaluate so it is impossible to provide a full review currently. Below are some general thoughts to improve the overall impact of the manuscript.

While it has become more popular and maybe accepted in scientific literature, starting sentences with “To...” is not proper English grammar.

The authors cite ENA PRJEB67949 for the “project,” but it is not public. Does that include the genomes and gene predictions? Also, no SRA number is cited. Does that include all the transcriptome data? Did the authors submit their analyzed transcriptome data to NCBI GEO? The authors state *C. multiflora* is on phytozome but I could not find it; are the other genomes and gene predictions going to be deposited? Are they on ENA? It would be great to add all three genomes and their gene predictions (gff, protein, CDS) on figshare so that the gene models mentioned in the manuscript can be evaluated directly. Very difficult to evaluate the findings without the genomes. It says in the methods that the transcriptome data is in the supplement (line 642) but I don’t see it there. This would also be a great addition to figshare or as a supplemental table.

Was the circadian clock considered? How are the genome dynamics the authors describe impacting the circadian clock? The authors do look at the evening element but call it the LHY motif and the evening element. It would be more consistent with the literature to call it the evening element.

What is panomics? Is it Pangenomics?

The authors whole thesis is based on polyploidy and whole genome duplication events (WGD). However, there does not appear to be a WGD analysis? When did the last WGD event occur?

The authors present haploid genomes. Did they try to generate phased haplotypes for *C. multiflora* since they have HiC? Or haplotype resolved genomes for the other two with HiFiasm? If not, why not?

The proteomics experiments are mentioned but the experimental approach is not really described in the results. It would be good to introduce these experiments and how they fit into the questions being asked.

(Remarks on code availability)

Reviewer #2

(Remarks to the Author)

This study represents the first sequencing and assembly of three *Clusia* genomes; a (hitherto considered) C3, a facultative and an obligate CAM species. This study uses these genome assemblies, in addition to information about chromosome number and genome size to speculate to the role the polyploidisation played in the evolution of CAM in *Clusia*. Overall, I think that this manuscript is an extremely impressive body of work, and represents a big step forward in the study of CAM evolution. However, I do have some reservations that I outline below.

Comments:

1. I feel that, at times, you are not using the word 'plasticity' with sufficient rigour. It seems like there are points throughout the manuscript where you use this word as a synonym for 'physiologically diverse'. Plasticity is the capacity to change a phenotype in response to environmental stresses. Plasticity itself can vary across a genus – for example consider two closely related species – *Clusia minor* and *Clusia pratensis*. Both exhibit plasticity in their photosynthetic physiology, as both can facultatively induce CAM. However, often the former species exhibits some amount of CAM under well-watered conditions, whereas the latter does not. Therefore *C. pratensis* displays more plasticity than *C. minor*.

#####

2. I think that the chosen species used in this study are excellent, as they represent three widely-distributed, well-studied *Clusia* species. However, studying *Clusia* can be somewhat fraught due to misidentification of species in various botanic gardens and plant collections. For example, *C. rosea* has been found to have extremely diverse carbon isotope ratios in this paper (DOI 10.32615/ps.2022.018). This is either due to misidentification, or extreme intraspecific variation – the answer to this is still unknown. In addition, IKEA sells a species of *Clusia* that they call *C. rosea* – although the leaf shape of this species looks completely different to the *C. rosea* that I am familiar with. Likewise, *C. minor* was only recently shown to be a distinct species from *C. pratensis*, but collections that predate this taxonomic classification may contain the latter, described as *C. minor*. And finally, *Clusia multiflora* is often considered to be a 'species complex' and can look quite different in different locations.

For all of these reasons, I think that it is important that you provide as many photos as possible to accompany the genomic data. These should include photos of flowers, leaves and stems, and should form a supplementary file. This will ensure that future users of your genomes will be able to make good, well-informed use of your data.

#####

3. I think that you have overstated the importance of polyploidy for the evolution of CAM. Many examples exist, such as in the abstract:

“Our findings reveal that polyploidization during genus evolution and subsequent diploidization shaped the emergence of extant C3+CAM subtypes in *Clusia*”

And there are other such statements throughout the text. At best, you have provided a hypothesis that progression across the C3-CAM 'continuum' requires gene(ome) duplication. However, the data you provide is not strong enough to robustly support this claim. Firstly, you have only looked at the genome size/chromosome count for 3 species, from disparate parts of the *Clusia* phylogeny. This makes your analysis subject to biases, as differences in genome size may have nothing to do with CAM. In fact, in the PhD thesis of V.A. Barrera Zambrano, the 2C DNA content for 9 species of *Clusia* was measured, including the three species you have assessed. The link to this thesis is <https://theses.ncl.ac.uk/jspui/handle/10443/1534> - see page 152. They showed that *Clusia rosea* was in fact somewhat of an outlier, and that for other constitutive CAM species, such as *C. hilariana*, there wasn't a particular difference in the 2C DNA content when compared to the C3 or C3/CAM species. I think that it is important that you take the data presented in this thesis into consideration when you frame the interpretations of your data.

Secondly, I am not sure I completely appreciate why a constitutive CAM phenotype would require a larger genome than a facultative CAM phenotype. Surely the capacity to do both C3 and CAM (and to be able to switch depending on the environment that the plant finds itself in) would require from a larger number of gene copies. For this reason, I think it is essential that you critically assess how important genome duplication really is for the evolution of CAM, and how much this might just be a coincidence.

#####

4. I am confused about why your physiology experiments did not employ standardised treatments. Unless I have misunderstood your manuscript, for your greenhouse experiment, you have sampled from 'control' plants that experienced quite a lot of shade, and were watered, and then stress plants that experienced a lot of light, and water was withheld. Also, *C. multiflora* received more light than *C. rosea*. Why is everything so different? Also, in the *C. rosea* 'control' plants there is a peak in light intensity at about 1.30 pm, that is not observable in the *C. multiflora* plants, nor does it correspond to any peak in sunlight that can be inferred from the 'stress' plants (also Fig. 4a). Can you explain why all of these inconsistencies are built into your experimental design?

Also, it would be very useful if Fig. 4a was split based on the 'control' and 'stress' plants, because at the moment the 'stress' plants' PAR is so much higher that is hard to tell how much light the 'control' plants get.

Finally, for the controlled growth chamber experiments (by the way a description of this is missing from the methods), why do the *C. multiflora* plants get less light than the *C. rosea* plants? And why is this the opposite of what was done in Fig. 4a?

Finally, Fig 4a and 4d shows the experimental conditions under which samples were collected. The rest of figure 4 doesn't have any mention of *C. minor*. But then in figure 5, data seems to have been collected for *C. minor*. Please can you explain why data is sometimes present for *C. minor* and sometimes absent?

#####

5. I think that a bit more detail/nuance is needed in your description of the phenotypes of the three species that you studied – especially for *C. multiflora*.

In the case of *C. rosea*, it is worth noting that even this 'constitutive CAM' species does a fair amount of C3 (i.e. has relatively pronounced phases II and IV) under well watered conditions. For example, doi:10.1093/jxb/eru022 showed that *C. rosea* did more C3 assimilation than either *C. hilariana* or *C. alata*, suggesting that this species is somewhat on the edge of being considered a facultative CAM species. Also, as I mentioned above, evidence from carbon isotope surveys have suggested that some individuals in this species may primarily be fixing carbon via C3 (if these herbaria samples are believed to be identified correctly). This ambiguity should also be mentioned in your background, and then you could show that under your growth conditions *C. rosea* displays a pretty strong CAM phenotype (clear phase I, no/little phases II or IV).

In the case of *C. minor*, it is worth noting that often even under well-watered conditions this species does a fair amount of weak CAM. (E.g. doi:10.1093/jxb/eru022). You don't provide gas exchange data for *C. minor*, so it is hard to tell exactly what this species is doing under your growth conditions. This makes interpreting the metabolic data a little trickier. It would be nice if you could add this analysis to your data set.

Finally, I am not convinced by your phenotypic characterisation of *C. multiflora*, as a weak CAM species. I think that to characterise this species in this way, you must first provide compelling phenotypic data, and then the molecular (RNA-seq proteomic etc) data can come in as a reinforcement. At the moment, your physiology data is not sufficiently strong, despite the molecular signals pointing towards some form of weak CAM in this species.

Your gas exchange does not show indications of weak CAM. Typically, when nocturnal respiratory CO₂ is being recycled, you see a small 'bulge' in the nocturnal CO₂ efflux. Sometimes, this 'bulge' pushes assimilation from a net negative to positive value for part of the night, but sometimes it just involves a reduction to CO₂ efflux, peaking at some point during the night. For example, *Clusia pratensis* often does this under well-watered conditions (see doi:10.1093/jxb/eru063, <https://doi.org/10.1016/j.jplph.2024.154185>, and <https://doi.org/10.1071/FP20268>) and also in other taxa (see <https://doi.org/10.1071/FP20151>, <https://doi.org/10.1071/FP20127> and <https://doi.org/10.1093/jxb/ery431>). The gas exchange data you show does not exhibit this tell-tale sign of weak CAM, which makes me question whether it really exists in *C. multiflora*.

Furthermore, your malate content (Fig. 5e) shows some accumulation of malate that coincides with dawn. I have a few comments on this. First, it would be more useful to show malate as a concentration, on a per leaf area and per fw basis. Having both would give a more comprehensive view of these data. Note that in a related species *Clusia tocuchensis*, there is a considerable change in the RWC of the hydrenchma tissue over a 24h period (<https://doi.org/10.1111/pce.14539> – Fig. S8). So there is a possibility that only reporting metabolite concentrations on a FW basis could be biased by this phenomenon. If a combination of low hydraulic conductance and high hydraulic capacitance results in considerable water losses when stomata open in the morning, this could drive the water content of leaves down, and result in an increase in malate content on a fw-basis, without any actual increase to malate content on a leaf-area basis. Maybe this is not the case, but as it stands it is not possible to make that assessment.

Secondly, it is unusual that the accumulation of malate only seems to occur right at the end of the night/around the beginning of dawn (Fig. 5e). These data imply that all the weak CAM activity is limited to the very end of the night, as there doesn't appear to be any difference between 7 pm and 4 am. This is quite interesting, but also makes me wonder whether malate accumulation has occurred in the dark (i.e. between 4 am and 6 am) or only in the first part of the light period (6am to 7am)? I can see from Fig. 4f, that the very beginning of the light period is characterised by an increase in assimilation. Could there

be some PEPC-mediated carboxylation during this time? It is not impossible, because strong CAM species of *Clusia* tend to exhibit PEPC activity for a few hours into the morning. Seeing as your second sampling time (7 am) is 1h after dawn, could you be observing some flux through the CAM cycle that just during the morning?

I think that based on the data presented so far, and also due to the complicated experimental design that I mentioned earlier, it is hard to be confident that *C. multiflora* really is doing weak CAM, and I think you need some more data to reinforce this finding. I would advise you measure malate concentrations over a diel period (including earlier time points in the night). If it were me, I as close to the beginning of the light period as possible, and then 2 hours afterwards, to also see if malate accumulation is limited to the early morning. Also, I would advise that you do this under controlled growth conditions, with each species experiencing the same treatment. If there is a rationale for treating each species differently, this may be ok, but it must be more explicitly stated. I strongly suspect that you will find *C. multiflora* to be doing no CAM, but I would love to be proved wrong! However, as it stands I don't think your data is strong enough to conclusively show that this species does any weak CAM.

(Remarks on code availability)

Reviewer #3

(Remarks to the Author)

Kramml and Herpell et al. report a phased chromosome level genome of the *Clusia multiflora*, and two de novo genomes from other *Clusia*. The authors showed an ancient WGD event and subsequent diploidization processes contributed to CAM in this group, which was first discovered by Alexander von Humboldt. They also used a multiomics approach to show that *Clusia multiflora* performs weak CAM rather than obligate C3 which was proposed by previous studies. Overall, the paper is well-written and I have some minor comments/concerns.

The quality of the genome assembly and annotation of *Clusia multiflora* is solid. This genome seems to be the first chromosome level genome assembly of Clusiaceae. This will be an important resource for researchers studying this family or interested in using *Clusia* genomes for other comparative studies. The genome assembly and annotation are not released on genBank or Phytozome. Please make sure these resources will become publicly available in the next stage.

L134-136 'There are almost no chromosomal rearrangements between homoeologous chromosomes, which is indicative of a relatively recent whole genome duplication event and strong support for a tetraploid state.' This statement is not accurate. Based on Figure 1 and 2a, there are many inversion and inter-chromosomal rearrangements between the homoeologous chromosomes.

L 260 '*C. multiflora* is generally regarded as an obligate C3 and *C. rosea* an obligate CAM plant.' It might be better to mention facultative C3 +CAM plant *C. minor* here given three genomes have been introduced in the previous sections. It makes it unclear why some of the experiments were only performed on the two species.

Fig. 6b provides an important summary of the evolution of CAM and the placement of WGDs and putative hybridization. However, the putative hybridization is not explained in the main text.

Fig. 6b, it is not clear to me what the reference is for the 'Clusiaceae WGD' placement. I assume it is based on Cai et al. 2019. However, Cai et al. 2019 only used two Clusiaceae transcriptomes which probably do not have a very high resolution of this placement. Given that this is the first chromosome level genome assembly of Clusiaceae, the authors should provide some basic synteny inference of this WGD. I will suggest the authors provide a self-self dot plot of the *Clusia multiflora* genome and syntenic depth ratio to *Amborella*, *Aquilegia* or *Coptis*, and *Vitis*. This helps rule out that there is no nested WGD.

Supplemental figures 5, 6, and 7 contain 'NA (NA)' in the Y axis. This is confusing. It does not seem like this provides any useful information.

Supplemental figure 7a, it is unclear to me why there is a 'PEPC4 (OG0001285)' and a 'PEPC4 (NA)'

(Remarks on code availability)

Reviewer #4

(Remarks to the Author)

Dear authors,

I enjoyed reading your manuscript. See my comments below:

- 1) There needs to be a more thorough comparison of the set of genes affected by pseudogenization with a set of genes that are conserved, or on the other hand affected positively by natural selection. Otherwise, it may seem that the genes affected by pseudogenization in starch metabolism were cherry-picked. I can see that some of this emerges in Figure 3 and in the supplementary figures, but it needs to be contrasted properly in the main body of the paper.
- 2) Are there any genes besides the starch/sugar malate cycle that are affected differentially among the different species that are worth discussing? Some works that have looked into the C3-CAM continuum have looked at the evolution in the capacity of PEPC. Decarboxylation is often widely discussed. How are other parts of CAM metabolism affected after diploidization of polyploids?
- 3) Could you briefly elaborate on the significance of the findings with regards to the adaptation of the different species? what selective pressures were pushing those species to move back on the C3-CAM continuum?
- 4) You have mentioned that your findings potentially lead to a paradigm shift in the understanding of the C3 to CAM evolution and that they modernize the understanding of the stepwise evolution of CAM. However you don't discuss both issues any further. If the species studied are product of a further evolution once CAM was reached, doesn't this mean that we still need to study species that are truly evolutionary intermediates? It is necessary to elaborate into the discussion of the C3-CAM continuum if a paradigm shift is to be claimed.
- 5) Some parts need to improve readability. Please improve the writing of lines 96-104 and 321-327.
- 6) See minor corrections in uploaded file.

Best!

(Remarks on code availability)

Version 1:

Reviewer comments:

Reviewer #1

(Remarks to the Author)

Thank you for including the genomes in this round of the review process.

The BUSCO for the scaffolds has a duplication rate of 18%, which is quite high after Purge Haplotigs. This duplication number seems accurate after looking more closely at the genome; there seems to be left over haplotigs in the assembly, which can also be seen in the dot plot provided in the "WGD history of *Clusia multiflora*." Moreover, the genes that have been identified as pseudogenized could be a result of haplotype confusion during assembly, which was a problem during this era of genome assembly with Falcon. Since many of the downstream analyses require accurate assessment of gene content, the current assembly raises some concerns, especially in the context of following a hypothesis regarding genome structure and WGD history. The genome version cited here as uploaded to NCBI(?) is a different version than included in the FigShare (v2.2) Lines 116-117: "The *Clusia multiflora* reference genome assembly (NCBI: CmuV2.3_predom) adds up to a total length of 1.5 Gb (Fig. 1). Is there a difference between these assemblies? Both of the other assemblies are low quality structurally (contig N50 < 1 Mb); what do they add to the manuscript? See below concerning evidence that they have diploidized.

Thank you for providing the document "WGD history of *Clusia multiflora*." While your rationale is understandable for not including this analysis (using previously calculated timing from transcriptome data and a student thesis), this data and integration are pivotal to the argument and the hypothesis put forward in your manuscript. Statements like "Despite some rearrangements between homoeologous chromosomes, they display a high degree of synteny when scaled by gene rank order, which suggests a relatively recent whole genome duplication event and offers strong support for a tetraploid origin," (Lines 138-140) should be bolstered by exactly how recent the WGD occurred. Moreover, you have the data to address this statement: "The large-scale structural similarities (macrosynteny) within the homoeologous groups further suggest that another, older, polyploidization event occurred before the more recent one." (Lines 144-145). Specifically, it would be good to know when genes are discussed later if they are part of the recent tetraploid event or a previous WGD event.

The authors argue that all three genomes are diploidized (several places cited below). *C. multiflora* seems like it could be diploidized since according to the "diploid" genome assembly on FigShare the total size was 2.84 Gb, which must be before Purge Haplotigs since many small contigs are included in that assembly. Supplementary Table 2 does say it assembled to 1.7 Gb so this must have been after Purge Haplotigs? However, the GenomeScope and Smudgeplot in Supplemental

Figure 2 suggest that *C. minor* and *C. rosea* could be triploid and tetraploid respectively; the plots shown are more consistent with that interpretation rather than diploid. Usually, the karyotype would help differentiate these possibilities, but Supplemental Figure 1 is not convincing over the plots in Supplemental Figure 2. Several crops, such as soybean and maize have diploidized after a WGD event, which would be interesting to discuss in terms of their fractionization and genome-trait innovation. Once again, since this is central to the authors central finding, it is essential these points are well supported. Lines 94-96: "Although the meiotic behavior of the chromosomes remains unknown, all species show diploidized genomes of polyploid origin, as the estimated levels of heterozygosity and allele topology suggest (Supplemental Fig. 2b-d)." Lines 103-106: "Overall, karyotype and sequencing analyses suggest that *C. multiflora* represents a diploidized allotetraploid, here treated as functional pseudo-diploid. *C. minor* seems to be a bona fide diploidized hexaploid (hereafter referred to as pseudo-triploid) and *C. rosea* may represent a diploidized octoploid, referred to as pseudo-tetraploid (Supplemental Fig. 2)." Lines 419-420: "All species/individuals investigated in this study are functional diploids with a tetra-, hexa-, or octoploid origin (Fig. 1-3, Supplemental Fig. 1,2,4)."

Lines 185-186 "We believe that these differences in diploidization and, thus, different genic landscapes are connected to the different physiotypes in the genus *Clusia*, with respect to their photosynthetic behavior." The hypothesis that CAM pathways have evolved as a consequence of diploidization after WGD is an intriguing hypothesis. However, this is a general and well-known phenomenon that leads to innovation in plant genomes across an array of phenotypes and lifestyles. Therefore, how is the process different or specific for CAM?

(Remarks on code availability)

Reviewer #2

(Remarks to the Author)

I was reviewer 2 in the first round of this manuscript.

Below, I am attaching my original comments, and your responses. My replies to your responses are indicated by a -> sign, to differentiate them from the first round of reviews.

Unfortunately, overall, I do not feel that the authors have provided satisfactory responses to my comments. I am confused by their response to my comments regarding the link between polyploidy and CAM – and I feel that they contradict themselves in their response.

Furthermore, I have found that there is a major lack of clarity in the manuscript and that I often struggle to assess the conditions that were employed for each experiment. This lack of clarity has been highlighted to a greater extent in the second round of peer review, in part due to the addition of new supplementary figures that do not correspond well with the main manuscript.

Finally, I still do not find that the authors provide sufficiently compelling evidence to show that *C. multiflora* is a weak CAM species. Additionally, after getting permission from the editor, I have sought a second opinion from another scientist with extensive experience working with weak CAM. I include their comments at the bottom.

Reviewer #2 (Remarks to the Author):

This study represents the first sequencing and assembly of three *Clusia* genomes; a (hitherto considered) C3, a facultative and an obligate CAM species. This study uses these genome assemblies, in addition to information about chromosome number and genome size to speculate to the role the polyploidisation played in the evolution of CAM in *Clusia*. Overall, I think that this manuscript is an extremely impressive body of work, and represents a big step forward in the study of CAM evolution. However, I do have some reservations that I outline below.

Comments:

1. I feel that, at times, you are not using the word 'plasticity' with sufficient rigour. It seems like there are points throughout the manuscript where you use this word as a synonym for 'physiologically diverse'. Plasticity is the capacity to change a phenotype in response to environmental stresses. Plasticity itself can vary across a genus – for example consider two closely related species – *Clusia minor* and *Clusia pratensis*. Both exhibit plasticity in their photosynthetic physiology, as both can facultatively induce CAM. However, often the former species exhibits some amount of CAM under well-watered conditions, whereas the latter does not. Therefore *C. pratensis* displays more plasticity than *C. minor*.

Agreed!

Instead of plasticity, we opted to use the terms "physiotype diversity", "ecophysiological diversity", or just "diversity" wherever appropriate.

-> Great!

#####

#####

2. I think that the chosen species used in this study are excellent, as they represent three widely-distributed, well-studied *Clusia* species. However, studying *Clusia* can be somewhat fraught due to misidentification of species in various botanic gardens and plant collections. For example, *C. rosea* has been found to have extremely diverse carbon isotope ratios in this paper (DOI 10.32615/ps.2022.018). This is either due to misidentification, or extreme intraspecific variation – the answer to this is still unknown. In addition, IKEA sells a species of *Clusia* that they call *C. rosea* – although the leaf shape of this species looks completely different to the *C. rosea* that I am familiar with. Likewise, *C. minor* was only recently shown to be a distinct species from *C. pratensis*, but collections that predate this taxonomic classification may contain the latter, described as *C. minor*. And finally, *Clusia multiflora* is often considered to be a ‘species complex’ and can look quite different in different locations.

For all of these reasons, I think that it is important that you provide as many photos as possible to accompany the genomic data. These should include photos of flowers, leaves and stems, and should form a supplementary file. This will ensure that future users of your genomes will be able to make good, well-informed use of your data.

We used the labels that were given to these plants by the Ulrich Lüttge group in the greenhouse at Darmstadt. We most certainly did not use commercially available material.

Additionally, we had several experts in neotropical botany re-examine the plants before setting out to sequence the genomes and we are quite confident in their assessment and the previous identification by the Lüttge group.

Nevertheless, we agree with the reviewer that it will be very helpful for the community and other future research to create less ambiguity. We therefore provide a plethora of selected photos as a supplemental resource deposited on figshare (private link for peer-review: <https://figshare.com/s/ccfabd5785fab31512fc>). Unfortunately, *Clusia rosea* never flowered in the greenhouse during the many years of cultivation. Therefore, we could not include photos of these. We did, however, include photographs of the leaves, stems, and the overall habitus.

-> Thank you for adding these photos. These plants all look nice and healthy too! With regards to *C. rosea*, whilst there are no flowers, it is nice to at least see the shape of the leaves, as this differentiates from the plants being sold as *Clusia rosea* by IKEA. However, I think that it is really important that you get a photo of a fully open *C. multiflora* flower. There is considerable ambiguity regarding the identity of species that are present in European *Clusia* collections. Considering that your genome is primarily based on *C. multiflora*, it is integral that you unambiguously confirm that this is the correct species. You mention that you consulted taxonomists? Are they authors on this paper? Can you provide the criteria that they used to identify these species, particularly *C. multiflora*? Also, were they aware of the recent categorisation of *C. minor* and *C. pratensis* as different species (rather than synonyms) and were they able to confirm that you are definitely working on the former and not the latter? I think more is needed here to unambiguously confirm that *C. multiflora* and *C. minor* are correctly identified.

#####

3. I think that you have overstated the importance of polyploidy for the evolution of CAM. Many examples exist, such as in the abstract:

“Our findings reveal that polyploidization during genus evolution and subsequent diploidization shaped the emergence of extant C3+CAM subtypes in *Clusia*”

And there are other such statements throughout the text. At best, you have provided a hypothesis that progression across the C3-CAM ‘continuum’ requires gene(ome) duplication. However, the data you provide is not strong enough to robustly support this claim. Firstly, you have only looked at the genome size/chromosome count for 3 species, from disparate parts of the *Clusia* phylogeny. This makes your analysis subject to biases, as differences in genome size may have nothing to do with CAM. In fact, in the PhD thesis of V.A. Barrera Zambrano, the 2C DNA content for 9 species of *Clusia* was measured, including the three species you have assessed. The link to this thesis is <https://theses.ncl.ac.uk/jspui/handle/10443/1534> - see page 152. They showed that *Clusia rosea* was in fact somewhat of an outlier, and that for other constitutive CAM species, such as *C. hilariana*, there wasn’t a particular difference in the 2C DNA content when compared to the C3 or C3/CAM species. I think that it is important that you take the data presented in this thesis into consideration when you frame the interpretations of your data.

Secondly, I am not sure I completely appreciate why a constitutive CAM phenotype would require a larger genome than a facultative CAM phenotype. Surely the capacity to do both C3 and CAM (and to be able to switch depending on the environment that the plant finds itself in) would require from a larger number of gene copies. For this reason, I think it is essential that you critically assess how important genome duplication really is for the evolution of CAM, and how much this might just be a coincidence.

We agree with the critical assessment of the reviewer that the role of WGD in the evolution of CAM in general was overstated and, thereby, subject to valid criticism. We amended the text at several instances (Abstract, Introduction,

Discussion and Conclusion) to make sure we no longer overstate. We agree that the data do not provide clear evidence to support the claim that polyploidization was necessary for CAM evolution in the first place, but they allow to formulate such hypothesis that can further be tested.

Also, we do not believe that a constitutive CAM phenotype would require a larger genome than a facultative CAM phenotype. All we mean to say is that by gaining multiple copies of a given pathway at some point in time, this creates "wiggle room" for neo-, sub-, and nonfunctionalization. Especially for strong CAM, polyploidization would provide/enable all necessary mechanisms for the required metabolic reprogramming to evolve, which can barely happen via single or tandem gene duplications alone.

-> Why 'especially for strong CAM'? What would strong CAM require a greater genetic neo- o sub- functionalisation that facultative or weak CAM? They all require the same changes to temporal expression patterns, just some have higher expression than others. I don't really follow this argument. You seem to have directly contradicted yourself here.

Therefore, WGDs are important drivers of added diversity (by increasing copy numbers of the genes) at certain points of an organism's evolutionary history. Since the post-polyploidy evolution of genome size (Cvalues) may either result in up- or downsizing (especially when either TEs or tandem repeats dynamics change), regardless of genic diploidization patterns, we have perhaps not been specific in previous version when referring to "diploidization". We do not refer to genome size changes, but to genic diploidization and accompanying regulatory and metabolic changes. What we emphasize is that the ancient WGD has been instrumental for the extant diversity in physiologies, which was mediated by a concurrence of (genic) diploidization with selective pressures under disparate climatic conditions.

We strongly believe that the combination of WGD and subsequent diploidization processes are the key drivers behind the phenotypic diversity we observe in *Clusia*. To be clear: We only provide evidence for the case of *C. multiflora*. This allows us to formulate a hypothesis to be tested in other species of the genus. We tried to clearly separate speculation from evidence in that respect and hope that the text now reflects our views more accurately.

-> Overall, I am not completely convinced by your response here. You start by saying that you agree with my assessment that the role of WGD was overstated, but then at the end of this paragraph you double down and say that you 'strongly believe that the combination of WGD and subsequent diploidization processes are the key drivers behind the phenotypic diversity we observe in *Clusia*'. Are these two statements not in contradiction of each other? Also, in my last round of review it was essentially this latter statement that I said lacked robust evidence.

-> Why has the updated manuscript not included the thesis I mentioned in my last review as a reference? For me this is one of the most important references for your study and should be addressed directly within the text. It is also worth mentioning that the 1C values that you estimated did not correspond that well with the 2C values in this thesis (particularly for *Clusia minor*). Seeing as the plants studied in this thesis were also from the Luttge collection, a direct comparison between those data and yours should highlight that there can be some ambiguity in the estimation of C values.

-> My main issue is that I still don't accept your interpretation here. You say that WGD events are important for neo-functionalisation of genes – this I think is fair. But you don't provide any data that robustly shows that WGD, or any other structural genomic change, has been important for CAM. Bearing in mind that there is so much genetic redundancy in vascular plants, what's to say that the genomic/genetic conditions required for the evolution of CAM were not present before the WGD event? Just showing that a WGD event happened in the early evolution of *Clusia* and that CAM evolved in this genus is not particularly strong evidence, and statistically would be an $n=1$ situation. To really show that WGD is important for CAM, you would have to show that it occurs more in lineages that evolved CAM, than in lineages that didn't. As it stands, you are simply noticing the co-occurrence of WGD and CAM in a single lineage. In Fig. 6a, you show many occurrences of polyploidisation in lineages that did not evolve CAM. Considering the frequency that WGD occurs in plant evolution, which you highlight in this figure, I don't see how you can make any claims that there is a causal link, based on the data you provide. Also, how many isoforms of PEPC were present in the ancestral genome before the WGD that occurred early in *Clusia* evolution? Can you provide any quantitative data or analyses to support the argument that neofunctionalization could not have led to CAM prior to this WGD event? Unfortunately, based on these considerations, I feel that you do not provide sufficient support for the claim made about WGD playing a role in the evolution of CAM.

#####

4. I am confused about why your physiology experiments did not employ standardised treatments. Unless I have misunderstood your manuscript, for your greenhouse experiment, you have sampled from 'control' plants that experienced quite a lot of shade, and were watered, and then stress plants that experienced a lot of light, and water was withheld. Also, *C. multiflora* received more light than *C. rosea*. Why is everything so different? Also, in the *C. rosea* 'control' plants there is a peak in light intensity at about 1.30 pm, that is not observable in the *C. multiflora* plants, nor does it correspond to any peak in sunlight that can be inferred from the 'stress' plants (also Fig. 4a). Can you explain why all of these inconsistencies are built into your experimental design?

Yes, we can explain the apparent disparities. We conducted two separate sets of experiments. One “open greenhouse experiment” and one “physiological measurements” experiment.

In the first experiment, which displays these differences, and we refer to as the “open greenhouse experiment”, we grew plants under natural sunlight to achieve very high light intensities, which we would not be able to reach with artificial lighting in our climate chambers at the time. The roof of the greenhouse was opened for that purpose, and we were reliant on the environment for the light conditions, so, an occasional cloud and the angle with which the sun hits can influence a given datapoint explaining the different amounts of light that were measured. The natural global irradiation was thereby subject to fluctuations. Additionally, the ‘control’ plants were grown right there, too, but we built a shaded stand for them. The shading mats have slits and openings to let wind pass through. If the sunlight passes directly through one of the openings and hits a sensor, this results in an apparent increase in light which would have affected just a tiny portion of the shaded area but looks like an overall light increase for the condition. The plants and the sensors were, however, always positioned in a randomized fashion. Any differences are mere sensor position coincidence and are, in our opinion, not generally reflective of the actual amount of light all plants received throughout the day.

-> This is not a completely satisfactory answer. Firstly, your explanation does not indicate why there was such a pronounced difference in the light intensity between species. In Fig. 4a, the ‘stressed’ plants of *C. multiflora* consistently get more light than *C. rosea*, which is most pronounced at 10 am (with a staggering difference in PAR of $300 \mu\text{mol m}^{-2} \text{s}^{-1}$). Why is this? This cannot be because of light momentarily passing through shades, because these are the plants that are under high light. Also, does this indicate that your randomised experimental design has failed? If it is because the light sensors used are biased by their positions, then this data is essentially meaningless, and provides no value to the manuscript.

-> Additionally, it is a poor experimental design to change two environmental conditions (water and light) when studying photosynthesis, especially when working with variable conditions imposed by natural sunlight. Is a light intensity of $600\text{--}900 \mu\text{mol m}^{-2} \text{s}^{-1}$ truly stressful to these plants? It's not such a high light intensity, that it is intuitive to me that this will cause light-stress. Can you provide A/Q curves, and NPQ data to show that this light intensity is very stressful for the plants? The reason I ask this, is because as it stands you have changed the light intensity alongside the water availability and then measured photosynthesis at a biochemical/molecular level. But your supposed ‘stress’ conditions also involve giving the plants a huge amount more-light, so surely this will always have an influence on photosynthesis, that has very little to do with stress? For example, if I studied the difference in photosynthetic rate in a plant grown under well watered and salty soils, but I also changed the light regime for these two groups of plants, then how much could I really know that the differences are because of salt stress vs because of light intensity?

-> Unfortunately, the issue raised here regarding the non-standardised experimental design has a knock-on effect on the data presented in Fig. 5, as these data seem to be generated after sampling from the greenhouse experiment. As the experimental design resulted in variable conditions for each species * treatment combination, it is very difficult to assess how much these random effects are contributing to the data vs the effect of the species and/or drought stress. I will comment more on this in my response to your use of molecular data to phenotype *C. multiflora*, below.

-> Overall, I find that this experimental design is very flawed, and leads to difficulties interpreting much of your metabolic data. I also note that you don't have corresponding gas exchange data to go alongside these sampling efforts. Whilst you have done gas exchange under controlled, standardised conditions, you do not have any gas exchange to show what these plants are doing under the less controlled greenhouse conditions. I suggest that you perform gas exchange for each species, under the 4 different experimental conditions that were used for sampling each of the species*treatment combinations. Only then can the reader get a sense of what these plants are doing. However, unfortunately this would still not improve the flawed experimental design used in this experiment, which would continue to limit the robustness of your conclusions.

The “physiological measurements” experiment was conducted in a completely controlled environment (see comments below).

Also, it would be very useful if Fig. 4a was split based on the ‘control’ and ‘stress’ plants, because at the moment the ‘stress’ plants’ PAR is so much higher that it is hard to tell how much light the ‘control’ plants get.

Thank you for the suggestion. For good interpretability, we supply the separate graphs as Supplemental Figure 10.

-> This has not increased my trust of these data. The graph you show for Fig. S10a does not correspond to the data shown in Fig. 4a, in a number of different ways. 1) the peak light intensity in Fig. 4a is at 10 am, whereas the peak in Fig. S10a is at 1 pm. 2) the light intensity experienced by *C. rosea* is greater than that experienced by *C. multiflora* in Fig. S10a, whereas it is the other way round in Fig. 4a. Overall, these are clearly not the same data. Also, I don't understand why your error bars for Fig. S10a go below 0, as this is not physically possible.

Finally, for the controlled growth chamber experiments (by the way a description of this is missing from the methods), why do the *C. multiflora* plants get less light than the *C. rosea* plants? And why is this the opposite of what was done in Fig. 4a?

We apologize for not having included a description of this part in the methods. It must have slipped out in-between versions. We made sure to include these methods in section “Plant cultivation and physiological experimentation” and also list environmental conditions in the new Supplemental Table 8. Thank you for the thorough analysis.

In fact, the differences in light intensity in the controlled experiments have to be corrected. After discussing this, we revised

the original raw data and realized that it was both an axis labelling error, caused by forwarding and copying the tables without the proper SI unit labels, and a data error (wrong sensor for *C. rosea*). We scaled the axes using a multiplier (10x for *C. multiflora*) so the actual value should be 400 $\mu\text{M m}^{-2} \text{s}^{-1}$. For *C. rosea* the wrong values (PARtop instead of PARambient, see below) were supplied. The actual light intensities are in both cases at the same level ($\sim 400 \mu\text{Mm}^{-2}\text{s}^{-1}$). Both plots are now corrected in Figure 4. We have also rechecked all other data to make sure that the data presented are absolutely correct.

-> thank you for correcting this. I suggest that you also tweak these graphs so that the x-axis for both 4c and 4f are the same length. This probably would involve having the x axis a little longer than the true length of the gas exchange, but this extra time can just be left black. I think it would just make it easier to compare these two graphs, if their x-axis aligned. Also, as mentioned above, it would be useful to include a line where $y = 0$.

Finally, Fig 4a and 4d shows the experimental conditions under which samples were collected. The rest of figure 4 doesn't have any mention of *C. minor*. But then in figure 5, data seems to have been collected for *C. minor*. Please can you explain why data is sometimes present for *C. minor* and sometimes absent?

Clusia spp. are quite difficult to work with on a molecular level. Their recalcitrant nature required us to adapt and improve almost every step of every protocol we used for both nucleic acid and protein extractions. To ensure comparability between species, we aimed to establish a protocol for each extraction that would work for all three plant species. In the case of the gDNA, we solved the contaminated DNA issues by pre-isolating nuclei. This allowed us to produce good assemblies for all three species. Even after rigorous testing, however, we were not able to apply the same protocol for RNA isolation to all three species.

We have therefore settled on the protocol that worked well for both *C. multiflora* and *C. rosea*, but was not as efficient for *C. minor*. Unfortunately, we were not able to sequence the resulting cDNA library for *C. minor*, and have therefore lost the sample available. For metabolomics and proteomics, our extraction procedures worked quite well for all three species. We have decided to add the data obtained for *C. minor* as the species is quite interesting in terms of its C3+CAM behavior, and to publish its genome alongside the others. We believe that the metabolomics and proteomics data might be useful for better interpretation of other data in the manuscript and for the community.

-> This is very fair. Some mention of these considerations should be included in the methods section.

```
#####  
#####  
#####
```

5. I think that a bit more detail/nuance is needed in your description of the phenotypes of the three species that you studied – especially for *C. multiflora*.

In the case of *C. rosea*, it is worth noting that even this 'constitutive CAM' species does a fair amount of C3 (i.e. has relatively pronounced phases II and IV) under well watered conditions. For example, doi:10.1093/jxb/eru022 showed that *C. rosea* did more C3 assimilation than either *C. hilariana* or *C. alata*, suggesting that this species is somewhat on the edge of being considered a facultative CAM species. Also, as I mentioned above, evidence from carbon isotope surveys have suggested that some individuals in this species may primarily be fixing carbon via C3 (if these herbaria samples are believed to be identified correctly). This ambiguity should also be mentioned in your background, and then you could show that under your growth conditions *C. rosea* displays a pretty strong CAM phenotype (clear phase I, no/little phases II or IV).

We made sure to mention this ambiguity in the text and cited the referenced article. In fact, we can confirm this observation. With this resubmission, we now provide 3–4-day continuous gas exchange measurements graphs highlighting the transition from cultivation conditions to stress treatment conditions to show the quite striking change in CO₂ uptake behavior. We performed such analyses also for *C. hilariana*. We wanted to discuss these data in a subsequent paper, but we understand that this may be critical for better understanding of the current manuscript. Therefore, we included the findings in the body of the text (lines 318-337) and as part of a new Supplemental Fig. 13.

-> I am really struggling to get the relevant information that is needed to critically assess your physiology data. As an example, I will describe the process I just went through to try to understand what you have done. I read in your comment, above, that you 'provide 3–4-day continuous gas exchange measurements graphs highlighting the transition from cultivation conditions to stress treatment conditions'. So this made me wonder if your gas exchange data in Fig. 4c and 4f, are under well-watered or drought-treated conditions. Are you saying that 3-4 days is enough time to induce a full drought response (it is probably not considering that it was not enough time in <https://doi.org/10.1016/j.jplph.2024.154185> to see any meaningful change). Anyway, I then took another look at fig S13, as your comment above was about *C. rosea*. I noticed that the gas exchange data trace did not resemble that presented in Fig. 4c. So I checked the materials and methods to see what you have said about the conditions for this experiment. But there is insufficient description here, and it sends me to supplementary table 8. When I check this table, there is no mention of any species name, just genus, which is *Clusia* for all experiments. Also, looking at this table makes it seem as though the sampling done for fig. 5 did not come from the 4 time points indicated in Fig. 4a (i.e. the stars) but was in fact done in several different experiments over the course of many years.

Is this the case? I'm sorry to be so difficult, but I am finding the lack of clarity in this manuscript so obstructive that I really cannot understand what it is that you have done. It is extremely difficult for me to highlight all of the missing clarity, because you have loaded the manuscript with so many supplementary figures and tables, many of which are inconsistent with the main manuscript (e.g. S10 is different data to 4a; and the gas exchange traces in s13 do not correspond to those in Fig4). I think you need to heavily edit the entire thing, including the supplementary data and figures, because as it stands I am becoming increasingly aware that it is not possible to understand what you have done.

In the case of *C. minor*, it is worth noting that often even under well-watered conditions this species does a fair amount of weak CAM. (E.g. doi:10.1093/jxb/eru022). You don't provide gas exchange data for *C. minor*, so it is hard to tell exactly what this species is doing under your growth conditions. This makes interpreting the metabolic data a little trickier. It would be nice if you could add this analysis to your data set.

We also provide gas exchange data for *C. minor* as part of Supplemental Figure 13.

-> Why is this data so noisy, and what are the environmental conditions under which it was measured (i.e. when was it watered). This is particularly relevant for this graph, as it looks like assimilation has almost completely stopped for this plant.

Finally, I am not convinced by your phenotypic characterisation of *C. multiflora*, as a weak CAM species. I think that to characterise this species in this way, you must first provide compelling phenotypic data, and then the molecular (RNA-seq proteomic etc) data can come in as a reinforcement. At the moment, your physiology data is not sufficiently strong, despite the molecular signals pointing towards some form of weak CAM in this species.

We performed a couple of additional experiments to answer the questions raised point by point in the sections below, leading us to change the characterization to constitutive C3+CAM.

Your gas exchange does not show indications of weak CAM. Typically, when nocturnal respiratory CO₂ is being recycled, you see a small 'bulge' in the nocturnal CO₂ efflux. Sometimes, this 'bulge' pushes assimilation from a net negative to positive value for part of the night, but sometimes it just involves a reduction to CO₂ efflux, peaking at some point during the night. For example, *Clusia pratensis* often does this under well-watered conditions (see <https://doi.org/10.1093/jxb/eru063>, <https://doi.org/10.1016/j.jplph.2024.154185>, and <https://doi.org/10.1071/FP20268>) and also in other taxa (see <https://doi.org/10.1071/FP20151>, <https://doi.org/10.1071/FP20127> and <https://doi.org/10.1093/jxb/ery431>). The gas exchange data you show does not exhibit this tell-tale sign of weak CAM, which makes me question whether it really exists in *C. multiflora*.

While the mentioned bulge is more pronounced in e.g. *Pilea peperomioides*, we generally do not observe it prominently in *Clusia*. All species show relatively constant dark period CO₂ loss rates just like the C₃ and C₄ reference plants. Instead, the CAM phases II (dawn) and IV (dusk) of PEPC activity are much more decisive as covered by the continuous gas exchange experiment visualized in Supplemental Fig. 13. The proposed emergence of the typical CAM curve during transition is also prevalent in the mentioned papers of *Clusia pratensis*.

-> OK here I must confess that in my last review I missed an essential detail in figure 4F. Whilst there is no 'bulge' in the nocturnal net assimilation trace, there does seem to be a (small) net positive A value during the night, which is also indicative of weak CAM. I think that this would be much clearer if you included a line to indicate y=0 on this graph, as you have done in Fig. S13. I also think that you should explicitly explain that your gas exchange provides some evidence of weak CAM, and make reference to the other *Clusia* papers I mention above, in order to highlight that your evidence differs to much of the published understanding of weak CAM in this genus.

-> That being said, I would also like you to comment on the precision of your gas exchange instrument to pick up on small fluxes, when stomata are closed. How often did you match the IRGAs (and why is this info not included in the materials and methods)? In Fig. 4c, there is a net negative transpiration rate at the start of the night. This is also evident in many of the graphs in Fig S13. Do you think that this is real? And if it is an artifact, can you truly be confident that your assimilation data when fluxes are so small (e.g. *C. multiflora* at night) are robust?

In addition, we believe that the molecular data is amongst the most important pieces of evidence here. The molecular data provide evidence that *C. multiflora* behaves very much like a CAM plant. With several independent experiments (see below) we have now shown that 1) *C. multiflora* accumulates malic acid during the night under controlled conditions and, 2) that organic acid accumulation acidifies the leaves, which is considered to be one of the most important phenotypic traits of CAM. Taking these observations and the molecular data together, we believe this is ample evidence to classify a plant as constitutive C₃+CAM (weak CAM or CAM-cycling).

-> But these data are based on a flawed experimental design, under which each species * treatment combination received different treatments. Also, you have no gas exchange to go alongside these greenhouse sampling efforts, so we don't know what the plants were doing under those conditions.

-> also you put emphasis on the PCA plots in Fig. 4b and e being similar, as evidence that *C. multiflora* does some weak CAM. Firstly, these PCA plots are not so similar. Furthermore, the data largely clusters due to the time/treatment for each species, which just indicates that the experiment has had some impact on the transcript profile, in a consistent way (within each species data set). It doesn't indicate anything about the photosynthetic phenotype of *C. multiflora*.

Furthermore, your malate content (Fig. 5e) shows some accumulation of malate that coincides with dawn. I have a few comments on this. First, it would be more useful to show malate as a concentration, on a per leaf area and per fw basis. Having both would give a more comprehensive view of these data. Note that in a related species *Clusia tocuchensis*, there is a considerable change in the RWC of the hydrenchma tissue over a 24h period (<https://doi.org/10.1111/pce.14539> – Fig. S8). So there is a possibility that only reporting metabolite concentrations on a FW basis could be biased by this phenomenon. If a combination of low hydraulic conductance and high hydraulic capacitance results in considerable water losses when stomata open in the morning, this could drive the water content of leaves down, and result in an increase in malate content on a fw-basis, without any actual increase to malate content on a leaf-area basis. Maybe this is not the case, but as it stands it is not possible to make that assessment.

We always sampled the same leaf area. As described in our methods: “We used sterile punch pliers with a 5 mm diameter to punch four holes into one leaf of each plant.” With this in mind we went back to see whether the biomasses (FW/leaf disc) across a day changed significantly and could find no influence of daytime on the leaf FW for *C. multiflora* (see figure below).

Secondly, it is unusual that the accumulation of malate only seems to occur right at the end of the night/around the beginning of dawn (Fig. 5e). These data imply that all the weak CAM activity is limited to the very end of the night, as there doesn't appear to be any difference between 7 pm and 4 am. This is quite interesting, but also makes me wonder whether malate accumulation has occurred in the dark (i.e. between 4 am and 6 am) or only in the first part of the light period (6am to 7am)? I can see from Fig. 4f, that the very beginning of the light period is characterised by an increase in assimilation. Could there be some PEPC-mediated carboxylation during this time? It is not impossible, because strong CAM species of *Clusia* tend to exhibit PEPC activity for a few hours into the morning. Seeing as your second sampling time (7 am) is 1h after dawn, could you be observing some flux through the CAM cycle that just during the morning?

As a matter of fact, this assessment is only true under non-stress conditions. If exposed to high-light and withheld watering there is a considerable increase in malic acid in *C. multiflora* between 7 pm and 4 am, even more pronounced than that between 4 am and 8 am (Fig. 5e, right side), which would further speak against the water loss hypothesis to explain the malic acid increase and shows that *C. multiflora* has the capacity to produce malic acid before dawn.

To address the comment in more rigorous way, we performed another set of experiments. We combined the measurements of malic acid in *C. rosea* and *C. multiflora* with pH and H⁺ measurements to assess the contribution of organic acid accumulation to leaf acidity. This time we did so under highly controlled conditions with the first sampling timepoint at 6 am (before dawn). The new results show consistently that there is considerable malic acid accumulation between 8 pm and 6 am in both control as well as treatment plants, which reduces during the day. In line with this observation, the pH values are low at 6 am and increase during the day. We incorporated these data into the new version of the manuscript as part of Supplemental Figure 12 and into the text in lines 311-317.

-> This isn't really performing a new set of experiments, but rather combining several data sets that were generated over multiple years and depicting them as a single experiment. Based on my comments above about the lack of transparency/clarity, this is a major issue for me. Were these different years of data collected under the same conditions? Were the plants the same size (this will effect their whole plant transpiration rate and thus their soil water status). Were the pots the same size? Why do the error bars fall below 0? What statistical tests have you done on these data? Considering that Fig. 12g depicts a tiny flux in malate, how confident can you be in the precision of your experiment? I think that combining data from different years is not an appropriate way to identify the presence of CAM when malate fluxes are this low, because these data will be so prone to small differences that could occur one year to the next. Also, why have you presented the 60 % LSD value for *C. multiflora* rather than the 90 % value as was done for everything else. Finally, and please forgive me if I have made a mistake here, but I took a ruler and looked at the error bars in each graph. It seems as though for each graph, every error bar is exactly the same size. This makes alarm bells go off for me, and these data needs to be checked. Regardless of this, unfortunately I think that the aforementioned issues are sufficiently great to make this graph lack any meaningful value in phenotyping *C. multiflora*.

-> My advice remains that to confidently conclude that *C. multiflora* does some CAM, you need to do a SINGLE experiment, under more controlled conditions, in which the sampling is done at a higher time resolution, for a full 24 hours. measurements of TA and malate for such an experiment would be strong evidence of whether *C. multiflora* does weak CAM. At the moment, your data is not strong enough, to make robust conclusions. Also, considering that your subsection entitled 'Diploidization of ancient polyploids underlies ecophysiological diversity of CAM' (starting line 391) relies on *C. multiflora* being C3-CAM for your current interpretation of your data, I would say that a lot of the conclusions in this manuscript are relying on weak data.

I think that based on the data presented so far, and also due to the complicated experimental design that I mentioned earlier, it is hard to be confident that *C. multiflora* really is doing weak CAM, and I think you need some more data to reinforce this finding. I would advise you measure malate concentrations over a diel period (including earlier time points in the night). If it were me, I as close to the beginning of the light period as possible, and then 2 hours afterwards, to also see if malate accumulation is limited to the early morning. Also, I would advise that you do this under controlled growth conditions, with each species experiencing the same treatment. If there is a rationale for treating each species differently, this may be ok, but it must be more explicitly stated. I strongly suspect that you will find *C. multiflora* to be doing no CAM, but I would love to be proved wrong! However, as it stands I don't think your data is strong enough to conclusively show that this species does any weak CAM.

We hope that the new experimental data regarding malate, H⁺ and pH, the reclassification as constitutive C3+CAM instead of weak CAM, the lack of a water evaporation effect, and the continuous gas exchange measurement was sufficient to get the reviewer to agree with our assessment. We would also like to point out that the CO₂ compensation point of around 10 ppm (predawn, postdawn, noon, measured with A/ci curves) for *C. multiflora* is very low compared to C3 plants and represents another important phenotypic indicator of some degree of CAM. And finally, also the WUE (as calculated as net assimilation to H₂O stomatal conductivity relation) is with a 1:5 ratio a lot smaller than that of a C3 plant, estimated at around 1:15 (Supplemental Table 7, Supplemental Fig. 13).

We thank the referee for the very thorough and helpful review that allowed us to improve the manuscript!

Minor comments

Line 54 – I'm not sure it is strictly true to say that CAM doesn't require structural changes in order to function. Even within *Clusia* there is a pretty substantial body of literature that shows that CAM is associated with anatomical adaptations. I think these need to be mentioned, or the text needs to be rephrased. (See <https://doi.org/10.1093/jxb/eru022> ; <https://doi.org/10.1093/botlinnear/boab075> ; <https://doi.org/10.1093/aob/mcad035>)

See

Line 431 – you need a reference to support this statement about environmental niches of these species.

Line 75 – I think you should name the 'physiotype' of *C. minor* (i.e. facultative CAM) and also briefly mention that it does some weak CAM under well-watered conditions.

Comments from a second opinion:

Start of message:

I can't say much about the genomics part of the manuscript, but I can say that the physiology part is not very strong, and this is worrying because it is largely the physiological experimentation that makes the authors conclude that *Clusia multiflora* shows some CAM. If true, this could have important implications for the understanding of CAM evolution in *Clusia*, because in none of the species of the *C. multiflora* complex has CAM been reported previously.

Given past misidentifications in the *Clusia* literature, at this point there is no guarantee that the authors' *C. multiflora* is indeed *C. multiflora*. Plants were received as a gift from another lab, and there is no information in the manuscript that the authors have confirmed species identities and have deposited pressed plant samples with flowers and fruits for future reference.

The presence of CAM typically results in nocturnal net CO₂ uptake and nocturnal increases in tissue acidity, i.e., of malic acid. As expected, *Clusia rosea* shows nocturnal net CO₂ uptake in Fig. 4c but rather unexpectedly doesn't show nocturnal net CO₂ uptake in Supplemental Figure 13c (?). *C. multiflora* doesn't show nocturnal net CO₂ uptake in both Fig. 4f and Supplemental Figure 13a suggesting that net CO₂ uptake is entirely via the C3 pathway. Thus, evidence of presence or absence of CAM in *C. multiflora* relies almost entirely on measurements of malic acid levels. These were conducted between 4 am and 7 pm, but not over a full day-night cycle. Indeed, Fig. 5e demonstrates a decline in malic acid level from 1 pm to 7 pm, but it is not demonstrated that malic acid levels return to the initial 4 am/8am level during the following night. It is important to demonstrate this, because factors other than CAM can lead to a decline in leaf acidity during daytime. Similarly, declines in malic acid levels (with large error bars) are shown in Supplemental Fig. 12g, but again no data are presented that show that malic acid levels increased during the following night.

In some Figures the y axis extends to negative values for malic acid, probably to accommodate the large error bars, but also for PAR (Supplemental Fig. 13) which is not possible.

In summary, there seems to be a possibility of weakly expressed CAM in *C. multiflora*, but the manuscript does not provide convincing evidence.

End of message

(Remarks on code availability)

Reviewer #3

(Remarks to the Author)

The authors addressed my comments/suggestions in the revision.

(Remarks on code availability)

Reviewer #4

(Remarks to the Author)

The work is a notable genomic study on the genomic rearrangements underwent by several species of the *Clusia* genus. The authors have described how these species in fact slid back on the C3-CAM continuum providing both genomic and physiological evidence, and in the revised version of the manuscript, a plausible evolutionary explanation due to their current ecological niche.

The work is quite relevant as it in fact provides a more complex view of the C3-CAM evolution. Among others, the authors mention an instance that I would call C3-CAM-pseudoC3. This research showcases that CAM as every adaptation is not a culprit of evolution both rather an evolutionary state that can move further on to other adaptations depending on selective pressures and availability of ecological niches.

Quite an inspiring work!

(Remarks on code availability)

Version 2:

Reviewer comments:

Reviewer #1

(Remarks to the Author)

The abstract advances an exciting claim: "Through a combination of phased chromosome-level assembly and annotation, comparative multi-omics, and physiological experiments, we demonstrate that diploidization of polyploids explains the phenotypic diversity of CAM." However, the manuscript does not yet provide a causal or even preferential link between diploidization and CAM. Whole-genome duplication followed by fractionation is the rule for angiosperms, so the burden is to show that CAM genes experienced a non-random fate relative to the genomic background and that those fates are tied to CAM phenotypes. At present, the results largely document fractionation/pseudogenization within CAM-related loci, but they do not test whether these patterns exceed genome-wide expectations or predict CAM function.

How does this compare to other paleotetraploids in terms of the fractionation of these CAM genes (even in non-CAM genomes since the GO terms in Supplemental Figure 5 are very similar to what is found in other genomes). I suggested in the last round to look at classic paleotetraploids like maize and soy (but there are many with great data). Or the authors could compare a host of pathways to see if the CAM pathway is preferentially impacted during the diploidization.

A second issue is assembly terminology and phasing. The current reference is a haploid/pseudo-haplotype assembly, yet the text calls it "phased." Phasing, in the strict sense, requires haplotype-resolved sequences and metrics (e.g., switch-error rate). With a single collapsed haplotype, haplotype collapse or mixing could mislead subgenome and fractionation inferences. The rebuttal introduces a CANU build and "manual curation" as validation, but the procedures and criteria are not described, and the statement that CAM pathway genes are "accurate" is undefined.

In short, the study compellingly catalogs CAM-gene fractionation but does not yet show that diploidization specifically shaped CAM beyond background WGD dynamics, and the assembly status needs to be aligned with current standards or terminology.

(Remarks on code availability)

Reviewer #2

(Remarks to the Author)

Clusia genomes shed light on the evolution and diversity of CAM physiotypes 3rd review

I was reviewer 2 in the first 2 rounds of peer review

Major comments –

I am not going to give responses to your responses to my responses, because this would be hard to read. Instead, I outline the 5 major points that were brought up in the first two rounds of peer review and give my responses below.

1. This was addressed in the last round of reviews

2. Fantastic! It is really great to have included a taxonomist in the analysis, and to have a record of what you have used for your genome sequencing – this will save a lot of headaches for future scientists who opt to use your genome as a resource. I am particularly impressed that you have deposited herbarium specimens and I agree that you have established “a new framework amid persistent taxonomic uncertainty”. Whilst you no longer have a C3 comparator in your study, I personally believe that this extra care and consideration in taxonomic identification actually adds more novelty than is lost from the lack of a fully C3 species. I know that there is some debate amongst CAM researchers as to the precise species identification of the first *Kalanchoe* genome that was assembled, so your study can now act as a baseline upon which future work can be based, to avoid such issues.

3. Please accept my apologies if I have been pedantic with regards to your use of English – this was not my goal in these comments – the manuscript is extremely well written.

I feel that the updated manuscript, for the most part, does not overstate the link between WGD and CAM evolution in the problematic way that the last versions did. If I understand correctly, all of your species are ‘functional diploids’ and you are hypothesising that the WGD may have contributed to the evolutionary lability within *Clusia*, rather than genome size being directly proportional to the ability to evolve CAM? If this is what you are saying, then I can get behind this as a hypothesis. I would say that this could be spelled out a bit more clearly, as I am not entirely sure that that I have interpreted this correctly. May I suggest that you use the PhD thesis by B. Zambrano as a comparison in the paragraph starting line 414. Because this thesis shows that genome size does not correlate with CAM strength (Even if there are taxonomic misidentifications (very possible) in this thesis, this actually doesn’t make any difference for this point, because the author has generated an internally consistent estimate of the strength of CAM for these 9 species – so even if they are not the species that they are reported to be, the analysis still holds). But seeing as you have shown that all of your study species are functional diploids then it would follow that if WGD did promote the evolution altered CAM states, this would not necessarily result in a correlation between genome size and CAM strength. I think that if this is the hypothesis that you are trying to present, then it would help to make this a bit more explicit.

That being said, I still do feel that the abstract of your manuscript is overstating the role of WGD in the evolution of CAM. For example, in your current abstract, you say “we demonstrate that diploidization of polyploids explains the physiotype diversity of CAM” – but really you are proposing a hypothesis, rather than a demonstration.

Also I think that line 422 should be changed to ‘we hypothesise that...’ and ‘may have played a crucial role’ – to make this clear that it is a hypothesis.

In a related point, I think that your use of “evolutionary intermediates” (line 45) is problematic, because it implies that weak and/or facultative CAM species are on an evolutionary trajectory towards becoming strong CAM. But this might not be the case and it is also equally parsimonious (based on only 3 species) that a strong CAM phenotype has evolved into a weaker CAM phenotype. I think that you should not implicitly assume that there is a particular direction to the evolution of CAM, because you don’t have data to support this.

4. I appreciate the time that it would take to recreate the greenhouse experiment under controlled conditions, as it would be prohibitively time consuming.

My initial issue stemmed primarily from the fact that you had drawn conclusions about the photosynthetic phenotypes of these species based on a highly variable experimental design, which made it difficult to differentiate G from E when drawing conclusions. I much prefer the way you have framed the data now, with Fig. 1 as the controlled phenotyping and figs 5 and 6 as examples of how you could use your genome sequences to understand the physiology of these species under field-like conditions (i.e. where there will be more variation). I am struck by what you say in the last round of reviewer responses – “In fields where molecular biology and ecology intersect, some degree of environmental variability is expected and, in our view, acceptable” – and I totally agree with you. In fact, I think that once the phenotyping has been done (under controlled conditions) this is a great example of how your genome can provide insights into what is going on under more realistic conditions. That being said, I still think that the manuscript would be improved from more data in the initial phenotyping,

under controlled conditions. At the moment, the only line of evidence that *C. major* is C3/CAM is that there is no detectable net CO₂ efflux at night in your gas exchange (a point that you don't seem to make – and is absent from the legend to Fig. 1 and elsewhere). But seeing as the nocturnal CO₂ assimilation rate is so low in Fig. 1B, it is hard to know that this falls within the precision of your instruments. If you were to measure titratable acidity, in all three species, grown under controlled conditions at dawn and dusk, under well watered and also droughted conditions, with the same light regime, you would be able to provide a second line of evidence that would help to confirm the photosynthetic physiology of the species you use in your study. This would then mean that the second physiology experiment could be used not to determine species genetic capacity for CAM, but how they are behaving under close-to-real world conditions. If you had this structure to the manuscript: i.e. 1) Phenotype the species under controlled conditions; 2) Identify the species ID using taxonomic techniques, 3) sequence the genome and assess WGD; and 4) use the genome in a more variable, close-to-real-world experiment to give direct evidence of the utility of the genome – this would be a really beautiful paper. As it stands it is just short of that, and the ambiguity of what *C. major* is doing leaves the reader with a sense of uncertainty in how to interpret the results presented in Fig 5 and 6, due to the combination of treatments employed in this experiment (i.e. drought and high light are always together).

Whilst I recognise that repeating the gas exchange for all of these plants would take too long, a titratable acidity experiment would be much more manageable, and in my opinion is the missing part to confirm that *C. major* is doing what you say it is. As you have pointed out in your manuscript, the prevalent misidentification of *Clusia* species in the literature makes it difficult to know that any previously published paper is definitely talking about the same species as you have. For this reason, it is integral that you have internally consistent phenotyping data conducted with a robust, highly controlled experimental design. Please note if you do this, that it is important to leave the plants under the controlled conditions to acclimate to those conditions for at least two weeks.

5. Most of what I wanted to say about this is covered in my response to point 4. But I will just add that the subtitle “CAM-like gene expression and protein activity patterns are retained in the C3-type mode of photosynthesis” line 284 does not really hold up because you haven't done sufficient phenotyping under controlled conditions – i.e. see my point above. The gas exchange trace gives some suggestion of weak CAM because there is no nocturnal respiratory efflux of CO₂, but this is not robust enough to know for sure what is going on. For this reason you cannot determine that these plants are truly exhibiting a “C3-type mode of photosynthesis” or if they are constitutive weak CAM under well watered conditions. Considering that there is already considerable debate about the degree to which low level acid fluctuations occur in C3 plants (see <https://doi.org/10.1111/nph.17790>) it is tricky to be calling something “C3-type” when it might be doing weak CAM. I would suggest changing this subtitle to something like “RNA-seq and proteomics to explore the nature of CAM under close to real-world conditions”, and also adding the titratable acidity experiment I outlined above to Fig. 1.

Minor comments:

Line 78 – this statement is unsubstantiated and I don't think it is correct. Whilst it is more than possible that 'most' and maybe even 'all' species of *Clusia* have some capacity of CAM, the data that currently exists in the literature has not shown this to be the case. For example isotopic analyses find many species that do not show CAM signatures (doi: 10.32615/ps.2022.018). Also a meta-analysis on photosynthetic modes in *Clusia* found that several species exhibited no upregulation of CAM even when drought stressed (<https://doi.org/10.1071/FP20268>). I think that as it stands, there is no robust way to quantify the extent to which CAM (especially weak CAM) occurs across the genus. I would advise changing this opening statement to something more like “Extensive research throughout the 20th century has established that many *Clusia* species possess an inherent genetic capacity for CAM”.

Line 165 - Do you mean *C. major* here?

Line 288 – “In an ‘open greenhouse’ experiment, we exposed the plants to conditions mimicking 289 native habitats under canopy environments (shaded) as well as exposed and drought conditions, to see 290 how they cope with adverse conditions” – this makes it seem as though the shade and drought treatments were applied separately, whereas in reality the high-light plants were always experiencing drought – this should be made more clear in this sentence.

Line 438 – are you specifically referring to *Clusia* in this statement – if so you should make this clear

Fig. 1 – Can you really say that “*C. major* exhibits a clear C3 -type of photosynthesis” when there is no nocturnal CO₂ efflux? Does this not suggest a weak CAM phenotype where nocturnally respired CO₂ is recycled via the CAM cycle?

(Remarks on code availability)

Version 3:

Reviewer comments:

Reviewer #2

(Remarks to the Author)

Peer review for *Clusia* Genome paper, March 2026

I was reviewer 2 in previous rounds of peer review. The editor has asked me to also assess whether the responses to reviewer 1's comments are satisfactory. Reviewer 1 wrote 2 paragraphs in the last round. Their first paragraph concerns the interpretation of WGD, and I have added comments to this in my 'major comments'. For the second paragraph of comments from reviewer 1, I do not have sufficient expertise to make a judgement of whether you have satisfied this comment.

Overall, I really enjoyed re-reading the manuscript, and feel that it is a significant contribution to the field.

Major comments:

A major difference in the manuscript is the addition of titratable acidity measurements to phenotype the species under controlled conditions. I thank the authors for adding these data, as I believe that they make a huge difference to the story. The manuscript no longer relies on data from the greenhouse experiment, where the environmental conditions are inherently noisy, to phenotype the three species.

I really appreciate that sufficient time was given to ensure that the plants were truly in a well-watered condition before the experiment began. My main criticism of these new data is that you have made the claim that $n=9$, when in reality $n=3$. The use of multiple technical replicates from each biological rep definitely strengthens your data, but if you use $n=9$, then this is pseudoreplication. Likewise, with the p values that you show – are these based on treating each of the 9 points as an individual replicate? Because if so then this needs to be changed.

In my last round of review, I mentioned that your writing, at times, depicts a directionality to the evolution of CAM – that the C3-CAM phenotypes are evolutionary intermediates between C3 and strong CAM. Whilst this might be the case, there is insufficient information to confirm this, as you are only looking at 3 species. It is also completely possible that *C. minor* and *C. major* evolved their 'intermediate' phenotypes from a strong CAM ancestor, rather than from a C3 ancestor. Because it is not possible to resolve this issue, with your data, more care is needed in your writing. Since the last review, you changed the phrasing in the abstract to say 'phenotypic phases'. However, I think you have not really dealt with the issue, but merely changed the phrasing to a different (more confusing) term. My point in the last review is that you cannot assume that the direction of evolution in your 3 species went C3 -> weak CAM -> strong CAM; because you don't have a strong enough phylogeny to resolve this. So you need to carefully reframe arguments that implicitly assume this direction.

For example, line 66/67 you say "investigating intermediate phenotypes of CAM in close relatives can help gather snapshots of different evolutionary stages". But you don't know the direction that the phenotypes have evolved, so how can you conclude anything about 'evolutionary stages'?

Also, line 73 you write "Our findings modernize our understanding of the "stepwise" evolution of CAM phenotypic diversity, the role of polyploidy in its, perhaps recurrent, origin..." - again, you don't know whether *C. minor* and *C. major* evolved their phenotypes from a C3 or a strong CAM ancestor, so you really cannot make any conclusions about the 'stepwise' nature of CAM evolution.

That being said, I do appreciate that you have engaged with this more directly later on (line 447). The issues are much more pronounced in the introduction and earlier parts of the manuscript.

On a related note, one of the previous criticism I have outlined (as has reviewer 1) was the claim that WGD was important for the evolution of CAM. I think both reviewer 1 and I pointed out that this claim would need to show that there was a higher rate of WGD than would be expected in *Clusia* and that this has led to the evolutionary lability within this genus. I appreciate that the results/discussion section now does outline the WGD idea much more as a hypothesis than as a proven theory, but I do think that you could make this even more explicit, just so this paper doesn't get cited incorrectly. My suggestion is that you include a paragraph from your response letter:

"Our primary objective is not to claim that diploidization uniquely targets CAM genes to the exclusion of other pathways, nor that CAM evolution required exceptional genome-wide behavior. Rather, our goal is to demonstrate that key CAM-associated genes did undergo fractionation and pseudogenization during diploidization, and that these changes are plausibly linked to observed CAM phenotypic diversity. From a functional and evolutionary perspective, it is not necessary to show that CAM genes behaved differently from all other genes in the genome; it is sufficient to establish that diploidization

affected CAM-relevant loci in ways that could influence CAM metabolism and regulation. Even if similar processes also impacted other pathways, this does not diminish their potential relevance for CAM evolution.”

My advice is that you include this after line 459. This will make it clear to the reader exactly the extent of your hypothesis, and will help them understand the nuance of what you are claiming/exploring.

Minor comments:

Line 31 – this sentence needs to be rephrased – maybe replace ‘evolved’ with ‘evolution’?

Line 53 – reference needed here (for the sentence that ends on line 53) – to support the evolutionary relevance and ecological role of CAM in *Clusia*

Line 54 – ‘transpiration’ should be ‘diurnal transpiration’

Line 65 – I think ‘conversion’ should be replaced by ‘evolution’ here.

Line 65 – I am not really sure why you make specific reference to ‘single cell C4’ here, or C4 at all? My advice would be to remove the part in the parenthesis, as it doesn’t add anything to your introduction.

Line 66/67 – see major comments

Line 70 – Where you say “use our data to reconstruct the evolutionary origins of subtypes of CAM in this particularly physiologically plastic plant genus” - I don’t know what you mean.

Line 73 – see major comments

Line 79 – it seems odd to me that you make an effort to highlight U Luttge in the parenthesis and then ref 21 is not one of his papers.

Line 79/80 – have you really ‘evaluated’ the claim that many species in *Clusia* do CAM? I think it would be more appropriate to say ‘To analyse this genus with contemporary molecular tools...’.

Line 85 – all weak CAM species (and all strong CAM species for that matter) do C3 photosynthesis, so it is not really correct to describe *C. major* as “consistently exhibiting C3-like behavior in 86 combination with weak CAM”. It is better to just describe it as a weak facultative CAM species.

Line 86 – “This selection aligned well with existing physiological data, especially when the notoriously challenging taxonomy of *Clusia* species is considered” – I don’t really know what this means.

Line 289 – as a plant physiologist (with limited genomics experience) reading this manuscript, I feel that it would be nice if you could qualify why you make this hypothesis. Is it based on the details that you have outlined before? Is there any quantitative evidence (i.e. comparison of diploidisation extent in starch degradation vs a basal background rate)? Or is it just a guess? I cannot tell what leads you to end this paragraph with this hypothesis, and would benefit from this being explained explicitly.

Line 396 – what do you mean by “Given that CAM originated much earlier in the two best-studied lineages” – are you referring to Crassulaceae and Portulacineae here? I think this sentence could be tighter, as I am not sure what you are referring to.

(Remarks on code availability)

REVIEWER COMMENTS

Reviewer #1 (Remarks to the Author):

In the manuscript entitled “Clusia genomes shed light on the evolution and plasticity of CAM,” the authors describe the sequencing and analysis of C. multiflora, C. minor, and C. rosea genomes, transcriptomes and proteomes to better understand the different types of CAM photosynthesis. The authors develop a compelling hypothesis that it is the return to diploidy that is driving the CAM innovation in this genus. However, the underlying genomes and data are not available to evaluate so it is impossible to provide a full review currently. Below are some general thoughts to improve the overall impact of the manuscript.

While it has become more popular and maybe accepted in scientific literature, starring sentences with “To...” is not proper English grammar.

We removed every occurrence of a sentence starting with “To...”.

The authors cite ENA PRJEB67949 for the “project,” but it is not public. Does that include the genomes and gene predictions? Also, no SRA number is cited. Does that include all the transcriptome data? Did the authors submit their analyzed transcriptome data to NCBI GEO? The authors state C. multiflora is on phytozome but I could not find it; are the other genomes and gene predictions going to be deposited? Are they on ENA? It would be great to add all three genomes and their gene predictions (gff, protein, CDS) on figshare so that the gene models mentioned in the manuscript can be evaluated directly. Very difficult to evaluate the findings without the genomes. It says in the methods that the transcriptome data is in the supplement (line 642) but I don’t see it there. This would also be a great addition to figshare or as a supplemental table.

We tried to prevent the public release of data before the article is published and, unfortunately, ENA does not provide a reviewer token to access it beforehand. Therefore, all data were made available on figshare, which for instance includes all three genomes and their annotations as well as gene expression data and many other things (hundreds of files in total). In fact, the data had been delivered to the editor as a separate dataset for peer-review and we do not know why it has not been made available to the reviewers. We apologize for this inconvenience and offer the following solution: All data has now been merged into the article’s general figshare repository (Supplementary Materials) under the private link: <https://figshare.com/s/ccfabd5785fab31512fc>

Because of ENAs regulations, we also decided to move all DNA/RNA sequencing-related data to NCBI. This includes all raw reads in SRA, draft assemblies in GenBank (C. minor & C. rosea), gene expression data in GEO, and the diploid assembly and annotation of C. multiflora. The data availability statement has been updated to include all the accession numbers mentioned. At time of release, all data will be accessible via a NCBI umbrella project. Unfortunately, the three assemblies have not yet been processed completely due to the “high volume of submissions we have received at GenBank”. All already processed sequencing data can be accessed under the private tokens linked below including an interactive Genome Browser providing easy access to the assembly and features of C. multiflora. Additionally, all requested data (and much more) is deposited on figshare to bridge the time until complete processing through GenBank.

DATA AVAILABILITY for peer-review

Figshare repository (private link)

<https://figshare.com/s/ccfabd5785fab31512fc>

Genome Browser

<https://apps.pph.univie.ac.at/jbrowse/>

Exemplary session of PPK:

<https://apps.pph.univie.ac.at/jbrowse/?session=share-laM8UKgj-P&password=GECMn>

GitHub repository

<https://github.com/hanneskramml/Clusia>

PRJNA1183765 - Whole-genome sequencing of *Clusia* spp. (NCBI SRA)

<https://dataview.ncbi.nlm.nih.gov/object/PRJNA1183765?reviewer=jfmrucfigitq94f4val6crdjk8>

PRJNA1197736 - Transcriptome sequencing of *Clusia* spp. (NCBI SRA)

<https://dataview.ncbi.nlm.nih.gov/object/PRJNA1197736?reviewer=g6d38i94pp7klgpdatt2od73f8>

GSE290226 - NCBI's Gene Expression Omnibus (reviewer token: urkvcioktrcxdsn)

<https://www.ncbi.nlm.nih.gov/geo/query/acc.cgi?acc=GSE290226>

PXD061385 - ProteomeXchange via PRIDE (reviewer token: ZNsZc9D3Uyca)

<https://www.ebi.ac.uk/pride/login>

Was the circadian clock considered? How are the genome dynamics the authors describe impacting the circadian clock? The authors do look at the evening element but call it the LHY motif and the evening element. It would be more consistent with the literature to call it the evening element.

We amended the text: all occurrences of “LHY motif” have been changed to “evening element”.

The circadian clock has been considered. Our considerations were the reason for including the promotor sequence analysis targeting TF binding sites known to be involved in circadian rhythms. The section is included in the chapter: “*Intronic transposon insertions and homoeolog fractionation affect CAM photosynthesis and starch metabolism in C. multiflora*”. In addition, we now deliver a GO term enrichment analysis, which shows that e.g. the CCA transcription factor (circadian rhythm) is highly affected by genic diploidization (Supplemental Fig. 5).

What is panomics? Is it Pangenomics?

We changed “panomics” to “multiomics”. With this term we refer to a cascade of different omics techniques, most importantly genomics, transcriptomics, proteomics, and metabolomics on the same set of samples.

The authors whole thesis is based on polyploidy and whole genome duplication events (WGD). However, there does not appear to be a WGD analysis? When did the last WGD event occur?

Although we understand the criticism, the body of work was already so extensive that we originally decided not to include any more analyses at that time but rather formulated a hypothesis based on relevant literature (e.g. Cai et al. 2019) and findings of this manuscript (e.g. high collinearity, chromosome numbers, proposed levels of polyploidy). However, in the meantime, we conducted a synteny-based WGD analysis including relative timings and absolute phylogenetic dating of speciation and polyploidization. The analysis reveals that *C. multiflora* is a former octoploid and the two rounds of WGD are estimated at a posterior mean of 36.7 mya (~ genus origin) and 52.6 mya (~ Clusiaceae radiation), allowing for two separate cycles of diploidization under different atmospheric/climatic conditions.

This perfectly fits the literature-based assumptions made in the manuscript (Fig. 6). We plan to publish these results in an upcoming paper (together with *C. rosea* as part of a PhD roadmap). Therefore, we offer to provide the analysis for peer-review only and forward it to the editor. We hope this is an acceptable solution. We also added a self-dot plot to Supplemental Figure 4 and amended the figure legend of Fig. 6 to make clear on which literature the WGD assumptions are based on.

The authors present haploid genomes. Did they try to generate phased haplotypes for C. multiflora since they have HiC? Or haplotype resolved genomes for the other two with HiFiasm? If not, why not?

The assembly was already made in 2021/2022 and technologies and algorithms are evolving quickly. At that time, we used the best approach available considering that we were uncertain about the type and level of polyploidy. The contigs used for scaffolding of the principal/predominant haplotype are phased but the final chromosome-level genome is not, they contain switch errors and in minor cases both haplotypes were wrongly integrated into the primary assembly. This does not affect the underlying analyses (due to manual curations) but will be improved in a follow-up study as we are already working on a fully phased haplotype-resolved genome of *C. multiflora* by utilizing the software GreenHill. Moreover, it is already a very extensive work with many different experiments, so we decided to publish only one high quality genome and focus on this species, while the others will follow in a subsequent paper. HiFiasm, however, cannot be used in our case since we have PacBio CLR reads rather than HiFi reads. And it will probably be a monoploid/haploid chromosome-level genome assembly of *C. rosea* given even higher level of ploidy of this species and insufficient read coverage to resolve all four or even eight haplotypes.

The proteomics experiments are mentioned but the experimental approach is not really described in the results. It would be good to introduce these experiments and how they fit into the questions being asked.

Proteomic, transcriptomic, and metabolomic data was generated from the same experiment. In the text we state:

“In order to establish a causal link between disruptions in starch metabolic genes and the plants’ photosynthesis mode, we tested the degree to which the three species sequenced here differ on a physiological level by physiological and molecular measurements: In an ‘open greenhouse’ experiment, we exposed the plants to conditions mimicking native habitats under canopy environments (control) as well as high light and drought conditions (stress), to see how they cope with adverse conditions (Fig. 4ad) [...] and performed transcriptomic, proteomic, and metabolomic analyses.”

In order to clarify, we amended the last sentence, which now states: "... and performed transcriptomic, proteomic, and metabolomic analyses **on the same set of samples from the same experiment** (Fig. 4, Fig. 5)."

The experiment itself is further described in Material & Methods under the section "Open greenhouse experiment" and protein extraction, sequencing, peptide identification and analysis are described in the MM section "Proteomics".

We thank the referee for very helpful comments and suggestions that allowed us to improve the manuscript.

Reviewer #2 (Remarks to the Author):

This study represents the first sequencing and assembly of three Clusia genomes; a (hitherto considered) C3, a facultative and an obligate CAM species. This study uses these genome assemblies, in addition to information about chromosome number and genome size to speculate to the role the polyploidisation played in the evolution of CAM in Clusia. Overall, I think that this manuscript is an extremely impressive body of work, and represents a big step forward in the study of CAM evolution. However, I do have some reservations that I outline below.

Comments:

1. I feel that, at times, you are not using the word ‘plasticity’ with sufficient rigour. It seems like there are points throughout the manuscript where you use this word as a synonym for ‘physiologically diverse’. Plasticity is the capacity to change a phenotype in response to environmental stresses. Plasticity itself can vary across a genus – for example consider two closely related species – Clusia minor and Clusia pratensis. Both exhibit plasticity in their photosynthetic physiology, as both can facultatively induce CAM. However, often the former species exhibits some amount of CAM under well-watered conditions, whereas the latter does not. Therefore C. pratensis displays more plasticity than C. minor.

Agreed!

Instead of plasticity, we opted to use the terms “physiotype diversity”, “ecophysiological diversity”, or just “diversity” wherever appropriate.

#####

2. I think that the chosen species used in this study are excellent, as they represent three widely-distributed, well-studied Clusia species. However, studying Clusia can be somewhat fraught due to misidentification of species in various botanic gardens and plant collections. For example, C. rosea has been found to have extremely diverse carbon isotope ratios in this paper (DOI 10.32615/ps.2022.018). This is either due to misidentification, or extreme intraspecific variation – the answer to this is still unknown. In addition, IKEA sells a species of Clusia that they call C. rosea – although the leaf shape of this species looks completely different to the C. rosea that I am familiar with. Likewise, C. minor was only recently shown to be a distinct species from C. pratensis, but collections that predate this taxonomic classification may contain the latter, described as C. minor. And finally, Clusia multiflora is often considered to be a ‘species complex’ and can look quite different in different locations.

For all of these reasons, I think that it is important that you provide as many photos as possible to accompany the genomic data. These should include photos of flowers, leaves and stems, and should form a supplementary file. This will ensure that future users of your genomes will be able to make good, well-informed use of your data.

We used the labels that were given to these plants by the Ulrich Lüttge group in the greenhouse at Darmstadt. We most certainly did not use commercially available material. Additionally, we had several experts in neotropical botany re-examine the plants before setting out to sequence the genomes and we are quite confident in their assessment and the previous identification by the Lüttge group.

Nevertheless, we agree with the reviewer that it will be very helpful for the community and other future research to create less ambiguity. We therefore provide a plethora of selected photos as a supplemental resource deposited on figshare (private link for peer-review: <https://figshare.com/s/ccfabd5785fab31512fc>). Unfortunately, *Clusia rosea* never flowered in the greenhouse during the many years of cultivation. Therefore, we could not include photos of these. We did, however, include photographs of the leaves, stems, and the overall habitus.

#####

3. I think that you have overstated the importance of polyploidy for the evolution of CAM. Many examples exist, such as in the abstract:

“Our findings reveal that polyploidization during genus evolution and subsequent diploidization shaped the emergence of extant C3+CAM subtypes in Clusia”

And there are other such statements throughout the text. At best, you have provided a hypothesis that progression across the C3-CAM ‘continuum’ requires gene(ome) duplication. However, the data you provide is not strong enough to robustly support this claim. Firstly, you have only looked at the genome size/chromosome count for 3 species, from disparate parts of the Clusia phylogeny. This makes your analysis subject to biases, as differences in genome size may have nothing to do with CAM. In fact, in the PhD thesis of V.A. Barrera Zambrano, the 2C DNA content for 9 species of Clusia was measured, including the three species you have assessed. The link to this thesis is <https://theses.ncl.ac.uk/jspui/handle/10443/1534> - see page 152. They showed that Clusia rosea was in fact somewhat of an outlier, and that for other constitutive CAM species, such as C. hilariana, there wasn’t a particular difference in the 2C DNA content when compared to the C3 or C3/CAM species. I think that it is important that you take the data presented in this thesis into consideration when you frame the interpretations of your data.

Secondly, I am not sure I completely appreciate why a constitutive CAM phenotype would require a larger genome than a facultative CAM phenotype. Surely the capacity to do both C3 and CAM (and to be able to switch depending on the environment that the plant finds itself in) would require from a larger number of gene copies. For this reason, I think it is essential that you critically assess how important genome duplication really is for the evolution of CAM, and how much this might just be a coincidence.

We agree with the critical assessment of the reviewer that the role of WGD in the evolution of CAM in general was overstated and, thereby, subject to valid criticism. We amended the text at several instances (Abstract, Introduction, Discussion and Conclusion) to make sure we no longer overstate. We agree that the data do not provide clear evidence to support the claim that polyploidization was necessary for CAM evolution in the first place, but they allow to formulate such hypothesis that can further be tested.

Also, we do not believe that a constitutive CAM phenotype would require a larger genome than a facultative CAM phenotype. All we mean to say is that by gaining multiple copies of a given pathway at some point in time, this creates “wiggle room” for neo-, sub-, and non-functionalization. Especially for strong CAM, polyploidization would provide/enable all necessary mechanisms for the required metabolic reprogramming to evolve, which can

barely happen via single or tandem gene duplications alone. Therefore, WGDs are important drivers of added diversity (by increasing copy numbers of the genes) at certain points of an organism's evolutionary history. Since the post-polyploidy evolution of genome size (C-values) may either result in up- or downsizing (especially when either TEs or tandem repeats dynamics change), regardless of genic diploidization patterns, we have perhaps not been specific in previous version when referring to "diploidization". We do not refer to genome size changes, but to genic diploidization and accompanying regulatory and metabolic changes. What we emphasize is that the ancient WGD has been instrumental for the extant diversity in physiologies, which was mediated by a concurrence of (genic) diploidization with selective pressures under disparate climatic conditions.

We strongly believe that the combination of WGD and subsequent diploidization processes are the key drivers behind the phenotypic diversity we observe in *Clusia*. To be clear: We only provide evidence for the case of *C. multiflora*. This allows us to formulate a hypothesis to be tested in other species of the genus. We tried to clearly separate speculation from evidence in that respect and hope that the text now reflects our views more accurately.

#####

4. I am confused about why your physiology experiments did not employ standardised treatments. Unless I have misunderstood your manuscript, for your greenhouse experiment, you have sampled from 'control' plants that experienced quite a lot of shade, and were watered, and then stress plants that experienced a lot of light, and water was withheld. Also, *C. multiflora* received more light than *C. rosea*. Why is everything so different? Also, in the *C. rosea* 'control' plants there is a peak in light intensity at about 1.30 pm, that is not observable in the *C. multiflora* plants, nor does it correspond to any peak in sunlight that can be inferred from the 'stress' plants (also Fig. 4a). Can you explain why all of these inconsistencies are built into your experimental design?

Yes, we can explain the apparent disparities. We conducted two separate sets of experiments. One "open greenhouse experiment" and one "physiological measurements" experiment.

In the first experiment, which displays these differences, and we refer to as the "open greenhouse experiment", we grew plants under natural sunlight to achieve very high light intensities, which we would not be able to reach with artificial lighting in our climate chambers at the time. The roof of the greenhouse was opened for that purpose, and we were reliant on the environment for the light conditions, so, an occasional cloud and the angle with which the sun hits can influence a given datapoint explaining the different amounts a light that were measured. The natural global irradiation was thereby subject to fluctuations. Additionally, the 'control' plants were grown right there, too, but we built a shaded stand for them. The shading mats have slits and openings to let wind pass through. If the sunlight passes directly through one of the openings and hits a sensor, this results in an apparent increase in light which would have affected just a tiny portion of the shaded area but looks like an overall light increase for the condition. The plants and the sensors were, however, always positioned in a randomized fashion. Any differences are mere sensor

position coincidence and are, in our opinion, not generally reflective of the actual amount of light all plants received throughout the day.

The “physiological measurements” experiment was conducted in a completely controlled environment (see comments below).

Also, it would be very useful if Fig. 4a was split based on the ‘control’ and ‘stress’ plants, because at the moment the ‘stress’ plants’ PAR is so much higher that is hard to tell how much light the ‘control’ plants get.

Thank you for the suggestion. For good interpretability, we supply the separate graphs as Supplemental Figure 10.

*Finally, for the controlled growth chamber experiments (by the way a description of this is missing from the methods), why do the *C. multiflora* plants get less light than the *C. rosea* plants? And why is this the opposite of what was done in Fig. 4a?*

We apologize for not having included a description of this part in the methods. It must have slipped out in-between versions. We made sure to include these methods in section “Plant cultivation and physiological experimentation” and also list environmental conditions in the new Supplemental Table 8. Thank you for the thorough analysis.

In fact, the differences in light intensity in the controlled experiments have to be corrected. After discussing this, we revised the original raw data and realized that it was both an axis labelling error, caused by forwarding and copying the tables without the proper SI unit labels, and a data error (wrong sensor for *C. rosea*). We scaled the axes using a multiplier (10x for *C. multiflora*) so the actual value should be $400 \mu\text{M m}^{-2} \text{s}^{-1}$. For *C. rosea* the wrong values (PARtop instead of PARambient, see below) were supplied. The actual light intensities are in both cases at the same level ($\sim 400 \mu\text{M m}^{-2} \text{s}^{-1}$). Both plots are now corrected in Figure 4. We have also rechecked all other data to make sure that the data presented are absolutely correct.

*Finally, Fig 4a and 4d shows the experimental conditions under which samples were collected. The rest of figure 4 doesn't have any mention of *C. minor*. But then in figure 5,*

data seems to have been collected for C. minor. Please can you explain why data is sometimes present for C. minor and sometimes absent?

Clusia spp. are quite difficult to work with on a molecular level. Their recalcitrant nature required us to adapt and improve almost every step of every protocol we used for both nucleic acid and protein extractions. To ensure comparability between species, we aimed to establish a protocol for each extraction that would work for all three plant species. In the case of the gDNA, we solved the contaminated DNA issues by pre-isolating nuclei. This allowed us to produce good assemblies for all three species. Even after rigorous testing, however, we were not able to apply the same protocol for RNA isolation to all three species. We have therefore settled on the protocol that worked well for both *C. multiflora* and *C. rosea*, but was not as efficient for *C. minor*. Unfortunately, we were not able to sequence the resulting cDNA library for *C. minor*, and have therefore lost the sample available. For metabolomics and proteomics, our extraction procedures worked quite well for all three species. We have decided to add the data obtained for *C. minor* as the species is quite interesting in terms of its C3+CAM behavior, and to publish its genome alongside the others. We believe that the metabolomics and proteomics data might be useful for better interpretation of other data in the manuscript and for the community.

#####

5. I think that a bit more detail/nuance is needed in your description of the phenotypes of the three species that you studied – especially for C. multiflora.

In the case of C. rosea, it is worth noting that even this ‘constitutive CAM’ species does a fair amount of C3 (i.e. has relatively pronounced phases II and IV) under well watered conditions. For example, doi:10.1093/jxb/eru022 showed that C. rosea did more C3 assimilation than either C. hilariana or C. alata, suggesting that this species is somewhat on the edge of being considered a facultative CAM species. Also, as I mentioned above, evidence from carbon isotope surveys have suggested that some individuals in this species may primarily be fixing carbon via C3 (if these herbaria samples are believed to be identified correctly). This ambiguity should also be mentioned in your background, and then you could show that under your growth conditions C. rosea displays a pretty strong CAM phenotype (clear phase I, no/little phases II or IV).

We made sure to mention this ambiguity in the text and cited the referenced article. In fact, we can confirm this observation. With this resubmission, we now provide 3–4-day continuous gas exchange measurements graphs highlighting the transition from cultivation conditions to stress treatment conditions to show the quite striking change in CO₂ uptake behavior. We performed such analyses also for *C. hilariana*. We wanted to discuss these data in a subsequent paper, but we understand that this may be critical for better understanding of the current manuscript. Therefore, we included the findings in the body of the text (lines 318-337) and as part of a new Supplemental Fig. 13.

In the case of C. minor, it is worth noting that often even under well-watered conditions this species does a fair amount of weak CAM. (E.g. doi:10.1093/jxb/eru022). You don’t provide gas exchange data for C. minor, so it is hard to tell exactly what this species is doing under your growth conditions. This makes interpreting the metabolic data a little trickier. It would be nice if you could add this analysis to your data set.

We also provide gas exchange data for *C. minor* as part of Supplemental Figure 13.

Finally, I am not convinced by your phenotypic characterisation of *C. multiflora*, as a weak CAM species. I think that to characterise this species in this way, you must first provide compelling phenotypic data, and then the molecular (RNA-seq proteomic etc) data can come in as a reinforcement. At the moment, your physiology data is not sufficiently strong, despite the molecular signals pointing towards some form of weak CAM in this species.

We performed a couple of additional experiments to answer the questions raised point by point in the sections below, leading us to change the characterization to constitutive C3+CAM.

Your gas exchange does not show indications of weak CAM. Typically, when nocturnal respiratory CO₂ is being recycled, you see a small 'bulge' in the nocturnal CO₂ efflux. Sometimes, this 'bulge' pushes assimilation from a net negative to positive value for part of the night, but sometimes it just involves a reduction to CO₂ efflux, peaking at some point during the night. For example, *Clusia pratensis* often does this under well-watered conditions (see doi:10.1093/jxb/eru063, <https://doi.org/10.1016/j.jplph.2024.154185>, and <https://doi.org/10.1071/FP20268>) and also in other taxa (see <https://doi.org/10.1071/FP20151>, <https://doi.org/10.1071/FP20127> and <https://doi.org/10.1093/jxb/ery431>). The gas exchange data you show does not exhibit this tell-tale sign of weak CAM, which makes me question whether it really exists in *C. multiflora*.

While the mentioned bulge is more pronounced in e.g. *Pilea peperomioides*, we generally do not observe it prominently in *Clusia*. All species show relatively constant dark period CO₂ loss rates just like the C₃ and C₄ reference plants. Instead, the CAM phases II (dawn) and IV (dusk) of PEPC activity are much more decisive as covered by the continuous gas exchange experiment visualized in Supplemental Fig. 13. The proposed emergence of the typical CAM curve during transition is also prevalent in the mentioned papers of *Clusia pratensis*.

In addition, we believe that the molecular data is amongst the most important pieces of evidence here. The molecular data provide evidence that *C. multiflora* behaves very much like a CAM plant. With several independent experiments (see below) we have now shown that 1) *C. multiflora* accumulates malic acid during the night under controlled conditions and, 2) that organic acid accumulation acidifies the leaves, which is considered to be one of the most important phenotypic traits of CAM. Taking these observations and the molecular data together, we believe this is ample evidence to classify a plant as constitutive C₃+CAM (weak CAM or CAM-cycling).

Furthermore, your malate content (Fig. 5e) shows some accumulation of malate that coincides with dawn. I have a few comments on this. First, it would be more useful to show malate as a concentration, on a per leaf area and per fw basis. Having both would give a more comprehensive view of these data. Note that in a related species *Clusia tocuchensis*, there is a considerable change in the RWC of the hydrenchma tissue over a 24h period (<https://doi.org/10.1111/pce.14539> – Fig. S8). So there is a possibility that only reporting metabolite concentrations on a FW basis could be biased by this phenomenon. If a combination of low hydraulic conductance and high hydraulic capacitance results in considerable water losses when stomata open in the morning, this could drive the water content of leaves down, and result in an increase in malate content on a fw-basis, without

any actual increase to malate content on a leaf-area basis. Maybe this is not the case, but as it stands it is not possible to make that assessment.

We always sampled the same leaf area. As described in our methods: “We used sterile punch pliers with a 5 mm diameter to punch four holes into one leaf of each plant.” With this in mind we went back to see whether the biomasses (FW/leaf disc) across a day changed significantly and could find no influence of daytime on the leaf FW for *C. multiflora* (see figure below).

Secondly, it is unusual that the accumulation of malate only seems to occur right at the end of the night/around the beginning of dawn (Fig. 5e). These data imply that all the weak CAM activity is limited to the very end of the night, as there doesn't appear to be any difference between 7 pm and 4 am. This is quite interesting, but also makes me wonder whether malate accumulation has occurred in the dark (i.e. between 4 am and 6 am) or only in the first part of the light period (6am to 7am)? I can see from Fig. 4f, that the very beginning of the light period is characterised by an increase in assimilation. Could there be some PEPC-mediated carboxylation during this time? It is not impossible, because strong CAM species of *Clusia* tend to exhibit PEPC activity for a few hours into the morning. Seeing as your second sampling time (7 am) is 1h after dawn, could you be observing some flux through the CAM cycle that just during the morning?

As a matter of fact, this assessment is only true under non-stress conditions. If exposed to high-light and withheld watering there is a considerable increase in malic acid in *C. multiflora* between 7 pm and 4 am, even more pronounced than that between 4 am and 8 am (Fig. 5e, right side), which would further speak against the water loss hypothesis to explain the malic acid increase and shows that *C. multiflora* has the capacity to produce malic acid before dawn.

To address the comment in more rigorous way, we performed another set of experiments. We combined the measurements of malic acid in *C. rosea* and *C. multiflora* with pH and H⁺ measurements to assess the contribution of organic acid accumulation to leaf acidity. This time we did so under highly controlled conditions with the first sampling timepoint at 6 am (before dawn). The new results show consistently that there is considerable malic acid accumulation between 8 pm and 6 am in both control as well as treatment plants, which reduces during the day. In line with this observation, the pH values are low at 6 am and increase during the day. We incorporated these data into the new version of the manuscript as part of Supplemental Figure 12 and into the text in lines 311-317.

I think that based on the data presented so far, and also due to the complicated experimental design that I mentioned earlier, it is hard to be confident that *C. multiflora* really is doing weak CAM, and I think you need some more data to reinforce this finding. I would advise you measure malate concentrations over a diel period (including earlier time points in

the night). If it were me, I as close to the beginning of the light period as possible, and then 2 hours afterwards, to also see if malate accumulation is limited to the early morning. Also, I would advise that you do this under controlled growth conditions, with each species experiencing the same treatment. If there is a rationale for treating each species differently, this may be ok, but it must be more explicitly stated. I strongly suspect that you will find *C. multiflora* to be doing no CAM, but I would love to be proved wrong! However, as it stands I don't think your data is strong enough to conclusively show that this species does any weak CAM.

We hope that the new experimental data regarding malate, H⁺ and pH, the reclassification as constitutive C₃+CAM instead of weak CAM, the lack of a water evaporation effect, and the continuous gas exchange measurement was sufficient to get the reviewer to agree with our assessment. We would also like to point out that the CO₂ compensation point of around 10 ppM (predawn, postdawn, noon, measured with A/ci curves) for *C. multiflora* is very low compared to C₃ plants and represents another important phenotypic indicator of some degree of CAM. And finally, also the WUE (as calculated as net assimilation to H₂O stomatal conductivity relation) is with a 1:5 ratio a lot smaller than that of a C₃ plant, estimated at around 1:15 (Supplemental Table 7, Supplemental Fig. 13).

We thank the referee for the very thorough and helpful review that allowed us to improve the manuscript!

Reviewer #3 (Remarks to the Author):

Kramml and Herpell et al. report a phased chromosome level genome of the Clusia multiflora, and two de novo genomes from other Clusia. The authors showed an ancient WGD event and subsequent diploidization processes contributed to CAM in this group, which was first discovered by Alexander von Humboldt. They also used a multiomics approach to show that Clusia multiflora performs weak CAM rather than obligate C3 which was proposed by previous studies. Overall, the paper is well-written and I have some minor comments/concerns.

The quality of the genome assembly and annotation of Clusia multiflora is solid. This genome seems to be the first chromosome level genome assembly of Clusiaceae. This will be an important resource for researchers studying this family or interested in using Clusia genomes for other comparative studies. The genome assembly and annotation are not released on genBank or Phytosome. Please make sure these resources will become publicly available in the next stage.

Unfortunately, ENA does not provide a reviewer token to access it beforehand. Therefore, we moved all DNA/RNA sequencing-related data to NCBI. This includes all raw reads in SRA, draft assemblies in GenBank (*C. minor* & *C. rosea*), gene expression data in GEO, and the diploid assembly and annotation of *C. multiflora*. The data availability statement has been updated to include all accession numbers. At the time of release, all data will be accessible via an NCBI umbrella project. Unfortunately, the three assemblies have not yet been processed completely due to “a high volume of submissions we have received at GenBank”. But all already processed sequencing data can be accessed under the private tokens linked below. All requested data and much more is also deposited on figshare under the following links including an interactive genome browser providing easy access to the genome of *C. multiflora*.

DATA AVAILABILITY for peer-review

Figshare repository (private link)

<https://figshare.com/s/ccfabd5785fab31512fc>

Genome Browser

<https://apps.pph.univie.ac.at/jbrowse/>

Exemplary session of PPK:

<https://apps.pph.univie.ac.at/jbrowse/?session=share-laM8UKgj-P&password=GECMn>

GitHub repository

<https://github.com/hanneskramml/Clusia>

PRJNA1183765 - Whole-genome sequencing of Clusia spp. (NCBI SRA)

<https://dataview.ncbi.nlm.nih.gov/object/PRJNA1183765?reviewer=jfmrucfigitq94f4val6crdjk8>

PRJNA1197736 - Transcriptome sequencing of Clusia spp. (NCBI SRA)

<https://dataview.ncbi.nlm.nih.gov/object/PRJNA1197736?reviewer=g6d38i94pp7klgpddat2od73f8>

GSE290226 - NCBI's Gene Expression Omnibus (reviewer token: urkvcioktrcxdsn)

<https://www.ncbi.nlm.nih.gov/geo/query/acc.cgi?acc=GSE290226>

L134-136 'There are almost no chromosomal rearrangements between homoeologous chromosomes, which is indicative of a relatively recent whole genome duplication event and strong support for a tetraploid state.' This statement is not accurate. Based on Figure 1 and 2a, there are many inversion and inter-chromosomal rearrangements between the homoeologous chromosomes.

We amended the statement. It now says: "Despite some rearrangements between homoeologous chromosomes, they display a high degree of synteny when scaled by gene rank order, which suggests a relatively recent whole genome duplication event and offers strong support for a tetraploid origin."

L 260 'C. multiflora is generally regarded as an obligate C3 and C. rosea an obligate CAM plant.' It might be better to mention facultative C3 +CAM plant C. minor here given three genomes have been introduced in the previous sections. It makes it unclear why some of the experiments were only performed on the two species.

Sadly, we only got good quality data for two out of the three species regarding our transcriptomic dataset. We added a disclaimer to the text reading:

"We also performed these experiments for the facultative CAM plant C. minor, but the recalcitrant nature of the plant prohibited us from obtaining clean transcriptomic data."

Clusia spp. are quite difficult to work with on a molecular level. Their recalcitrant nature required us to adapt and improve almost every step of every protocol we used for both nucleic acid and protein extractions. To ensure comparability between species, we aimed to establish a protocol for each extraction that would work for all three plant species. In the case of the gDNA, we solved the contaminated DNA issues by pre-isolating nuclei. This allowed us to produce good assemblies for all three species. Even after rigorous testing, however, we were not able to apply the same protocol for RNA isolation to all three species. We have therefore settled on the protocol that worked well for both C. multiflora and C. rosea, but was not as efficient for C. minor. Unfortunately, we were not able to sequence the resulting cDNA library for C. minor, and have therefore lost the sample available. For metabolomics and proteomics, our extraction procedures worked quite well for all three species. We have decided to add the data obtained for C. minor as the species is quite interesting in terms of its C3+CAM behavior, and to publish its genome alongside the others. We believe that the metabolomics and proteomics data might be useful for better interpretation of other data in the manuscript and for the community. Therefore, we tried to incorporate these data into the text, too.

Fig. 6b provides an important summary of the evolution of CAM and the placement of WGDs and putative hybridization. However, the putative hybridization is not explained in the main text.

We now mention the putative hybridization event in the text (lines 422-423).

Fig. 6b, it is not clear to me what the reference is for the 'Clusiaceae WGD' placement. I assume it is based on Cai et al. 2019. However, Cai et al. 2019 only used two Clusiaceae transcriptomes which probably do not have a very high resolution of this placement. Given that this is the first chromosome level genome assembly of Clusiaceae, the authors should

provide some basic synteny inference of this WGD. I will suggest the authors provide a self-self dot plot of the *Clusia multiflora* genome and syntenic depth ratio to *Amborella*, *Aquilegia* or *Coptis*, and *Vitis*. This helps rule out that there is no nested WGD.

Cai et al. 2019 was indeed an important source as well as the general clustering of events of polyploidization according to Van de Peer et al. 2017, and timings of speciation/radiation in Malpighiales (Xi et al. 2012). Together with the findings of this manuscript (e.g. high collinearity, chromosome numbers, proposed levels of polyploidy) we made a most likely assumption and formulated a hypothesis for further discussion and future work.

Although we understand the criticism in view of the low resolution of the placement despite the new genome, the body of work is already so extensive that we had decided not to include any more analyses at the time of first submission. However, in the meantime, we conducted a synteny-based WGD analysis including relative timings and absolute phylogenetic dating of speciation and polyploidization. The analysis reveals that *C. multiflora* is of octoploid origin and the two rounds of WGD are estimated at a posterior mean of 36.7 mya (~ genus origin) and 52.6 mya (~ Clusiaceae radiation), allowing for two separate cycles of diploidization under different atmospheric/climatic conditions.

This perfectly fits the literature-based assumptions made in the manuscript (Fig. 6). However, this is ongoing and future work as part of a PhD roadmap. Therefore, we provide the results of this analysis for peer-review only and also forward it to the editor. We hope this is an acceptable solution. We added the requested self-dot plot to Supplemental Figure 4 and amended the figure legend of Fig. 6 to make clear on which literature the WGD assumptions are based on. And there are also additional riparian-/dotplots published as part of Supplementary Data.

Supplemental figures 5, 6, and 7 contain 'NA (NA)' in the Y axis. This is confusing. It does not seem like this provides any useful information.

Thank you for noticing. We improved the algorithm for homoeolog resolution and manually curated remaining genes/genic fragments. This eliminated the NAs from the supplemental figures.

Supplemental figure 7a, it is unclear to me why there is a 'PEPC4 (OG0001285)' and a 'PEPC4 (NA)'

This pertains to the gene family concept we introduced to plot these data. The PEPC4 (NA) is a PEPC4 that belongs to a different orthogroup other than the global gene family, attributed to gene fractionation. This issue has also been resolved with the improved version.

Thank you for all comments and suggestions, it helped us to improve the paper.

Reviewer #4 (Remarks to the Author):

Dear authors,

I enjoyed reading your manuscript. See my comments below:

1) There needs to be a more thorough comparison of the set of genes affected by pseudogenization with a set of genes that are conserved, or on the other hand affected positively by natural selection. Otherwise, it may seem that the genes affected by pseudogenization in starch metabolism were cherry-picked. I can see that some of this emerges in Figure 3 and in the supplementary figures, but it needs to be contrasted properly in the main body of the paper.

We conducted extensive literature research to identify a subset of about 70 gene families that, according to current knowledge, are in some way related to CAM (ref. 4, 40-46). The complete list is included in the Supplementary Code (metadata.xlsx) as well as together with all evidence pertaining to diploidization in Supplementary Table 4. Within the CAM-related subset, we see clear differences in respective genes or pathways involved. To illustrate this further, we have now performed individual GO term enrichments of pseudogenization/fractionation, intron/repeat length, and conserved genes, respectively. We amended the analysis in the main body as part of a new Supplemental Fig. 5.

2) Are there any genes besides the starch/sugar malate cycle that are affected differentially among the different species that are worth discussing? Some works that have looked into the C3-CAM continuum have looked at the evolution in the capacity of PEPC. Decarboxylation is often widely discussed. How are other parts of CAM metabolism affected after diploidization of polyploids?

We do appreciate this excellent comment. PEPC seems intact/conserved and fully functional (Fig. 3a, Fig. 4g) and we partly addressed evolutionary considerations in terms of shared sub-functionalization among our selected species (Supplemental Fig. 9b). Currently we only provide evidence of diploidization for *C. multiflora*. This inference allows us to formulate a hypothesis to be tested later in other species of the genus.

To comparatively cover the evolution of PEPC or other key CAM genes we would greatly benefit from further improvements of the assemblies/annotations of the other species, that are currently planned for. This would allow us to disentangle the events that have accompanied each of the two hypothesized rounds of polyploidization as well as their effects on transposon-mediated sub-, neo-, and non-functionalization (i.e. genic diploidization). Although, we can already see some patterns based on genomic alignments visualized in the new interactive GenomeBrowser. Here we provide an example of PPDK as mentioned in the manuscript: <https://apps.pph.univie.ac.at/jbrowse/?session=share-laM8UKgj-P&password=GECMn>

3) Could you briefly elaborate on the significance of the findings with regards to the adaptation of the different species? what selective pressures were pushing those species to move back on the C3-CAM continuum?

We added a brief section commenting on the selective pressures of each plant in lines 427-438: *“Distinct distribution areas are indicative of different selective pressures that the ancestors of our extant plants were exposed to. [...] While C. rosea can be found in harsh, dry, and salty environments, which might explain its high WUE and stronger CAM phenotype, C. multiflora is mostly found within canopy shaded rainforests, which, in turn, may represent the absence of a selective pressure to evolve or maintain a strong CAM phenotype. When duplicated gene copies within specific pathways were adapted (sub/neofunctionalization), phased out (intronic TEs) or lost (non-functionalization via fractionation), these could no longer contribute to an efficient CAM metabolism. Eventually, this forced C. multiflora back to an energetically favorable C3-like behavior but with mixed CAM elements remaining intact.”*

4) You have mentioned that your findings potentially lead to a paradigm shift in the understanding of the C3 to CAM evolution and that they modernize the understanding of the stepwise evolution of CAM. However you don't discuss both issues any further. If the species studied are product of a further evolution once CAM was reached, doesn't this mean that we still need to study species that are truly evolutionary intermediates? It is necessary to elaborate into the discussion of the C3-CAM continuum if a paradigm shift is to be claimed.

Thank you very much for this valuable comment. We hope that our study will stimulate the investigations on these truly evolutionary intermediates. In this revised version we do discuss the significance of our findings in more detail (see comment above and the whole paragraph “Diploidization of ancient polyploids underlies ecophysiological diversity of CAM”), but we’ve toned down our claims regarding the paradigm shift.

5) Some parts need to improve readability. Please improve the writing of lines 96-104 and 321-327.

We improved the two sections. Now lines 94-106 and 362-368.

6) See minor corrections in uploaded file.

Thank you for all corrections, they have now been amended.

Best!

We thank the referee for very helpful suggestions and corrections that allowed us to improve the manuscript.

REVIEWER COMMENTS

Reviewer #1 (Remarks to the Author):

Thank you for including the genomes in this round of the review process.

The BUSCO for the scaffolds has a duplication rate of 18%, which is quite high after Purge Haplotigs. This duplication number seems accurate after looking more closely at the genome; there seems to be left over haplotigs in the assembly, which can also be seen in the dot plot provided in the “WGD history of *Clusia multiflora*.” Moreover, the genes that have been identified as pseudogenized could be a result of haplotype confusion during assembly, which was a problem during this era of genome assembly with Falcon. Since many of the downstream analyses require accurate assessment of gene content, the current assembly raises some concerns, especially in the context of following a hypothesis regarding genome structure and WGD history. The genome version cited here as uploaded to NCBI(?) is a different version than included in the FigShare (v2.2) Lines 116-117: “The *Clusia multiflora* reference genome assembly (NCBI: CmuV2.3_predom) adds up to a total length of 1.5 Gb (Fig. 1). Is there a difference between these assemblies? Both of the other assemblies are low quality structurally (contig N50<1 Mb); what do they add to the manuscript? See below concerning evidence that they have diploidized.

-> We performed an additional assembly using CANU at the time, applying polyploid-aware parameters specifically designed to minimize haplotype collapse. This alternative assembly was used to screen the final chromosome-level version for incorrectly integrated haplotypes. In cases where haplotype redundancy was detected—particularly within CAM-related loci—manual curation (or exclusion from downstream analyses) was performed to confirm our discussed set of CAM genes and corresponding pseudogenes. Overall, it is possible that the total number of pseudogenes might be slightly overestimated but the refined set of CAM-genes, on which our hypothesis is based upon, is accurate. The CANU-based alignment is now available and can be explored in the Genome Browser (Link: <https://apps.pph.univie.ac.at/jbrowse/>). Furthermore, we included the requested WGD analysis as part of Supplemental Fig. 11 and a revised Figure 7 (former Fig. 6), see below.

-> There is no structural difference between the NCBI and figshare assemblies. The discrepancy in version numbers refers to the annotation version rather than the assembly itself. The version submitted to NCBI underwent a revised gene annotation process, which included the consolidation of isoforms that only varied in UTRs, in compliance with NCBI's standards. It should be noted that we had to revise the species identification for *C. multiflora* (which was revised to *Clusia major* L., see Supplemental File 1) within this revision process, which complicated data deposition. Paired with lengthy response times of some NCBI services (we have been waiting for over half a year for GenBank's response), we decided to deposit the assembly without annotation (as part of BioProject PRJNA1183765). It should be mentioned that we still introduced a new version of annotation (v2.3) to reflect the revised species, as the gene IDs have been given a new prefix (Cma). All final results have been adapted accordingly. The full annotation is available for review in the figshare repository (Link: <https://figshare.com/s/ccfabd5785fab31512fc>) and also searchable within the Genome Browser (Link: <https://apps.pph.univie.ac.at/jbrowse/>). We will provide publicly accessible links for all repositories (including a public figshare under DIO: 10.6084/m9.figshare.26212895) and all the deposited data will be made public upon acceptance, and we will make sure all cross references detailed in our data availability statement check out.

-> Regarding the inclusion of additional, lower-contiguity assemblies: these were incorporated in direct response to a prior round of reviewer feedback. While these

assemblies are not of chromosome-level quality, they represent complete reference genomes on contig-level that were mapped to the high-quality assembly. They contribute to the manuscript's broader comparative and evolutionary context, as detailed in the various sections (e.g., Fig. 3d, Supplemental Fig. 9, interspecies pangenome creation, transcriptomics/proteomics plots). We also believe that making such resources available is consistent with the FAIR principles, particularly for underrepresented clades.

Thank you for providing the document "WGD history of *Clusia multiflora*." While your rationale is understandable for not including this analysis (using previously calculated timing from transcriptome data and a student thesis), this data and integration are pivotal to the argument and the hypothesis put forward in your manuscript. Statements like "Despite some rearrangements between homoeologous chromosomes, they display a high degree of synteny when scaled by gene rank order, which suggests a relatively recent whole genome duplication event and offers strong support for a tetraploid origin," (Lines 138-140) should be bolstered by exactly how recent the WGD occurred. Moreover, you have the data to address this statement: "The large-scale structural similarities (macrosynteny) within the homoeologous groups further suggest that another, older, polyploidization event occurred before the more recent one." (Lines 144-145). Specifically, it would be good to know when genes are discussed later if they are part of the recent tetraploid event or a previous WGD event.

-> We thank you for your comments and now included the analysis as part of Supplemental Fig. 11. In order to not confuse the reader, we removed the statements in lines 138-140 and 144-145 and summarized the findings in section "Diploidization of ancient polyploids underlies ecophysiological diversity of CAM". The new analysis in combination with the revised species/clade (Supplemental File 1) led to a refinement of the proposed phylogenetic placement of the recent tetraploid event (c2-WGD event, see Figure 7).

The authors argue that all three genomes are diploidized (several places cited below). *C. multiflora* seems like it could be diploidized since according to the "diploid" genome assembly on FigShare the total size was 2.84 Gb, which must be before Purge Haplotigs since many small contigs are included in that assembly. Supplementary Table 2 does say it assembled to 1.7 Gb so this must have been after Purge Haplotigs? However, the GenomeScope and Smudgeplot in Supplemental Figure 2 suggest that *C. minor* and *C. rosea* could be triploid and tetraploid respectively; the plots shown are more consistent with that interpretation rather than diploid. Usually, the karyotype would help differentiate these possibilities, but Supplemental Figure 1 is not convincing over the plots in Supplemental Figure 2. Several crops, such as soybean and maize have diploidized after a WGD event, which would be interesting to discuss in terms of their fractionization and genome-trait innovation. Once again, since this is central to the authors central finding, it is essential these points are well supported. Lines 94-96: "Although the meiotic behavior of the chromosomes remains unknown, all species show diploidized genomes of polyploid origin, as the estimated levels of heterozygosity and allele topology suggest (Supplemental Fig. 2b-d)." Lines 103-106: "Overall, karyotype and sequencing analyses suggest that *C. multiflora* represents a diploidized allotetraploid, here treated as functional pseudo-diploid. *C. minor* seems to be a bona fide diploidized hexaploid (hereafter referred to as pseudo-triploid) and *C. rosea* may represent a diploidized octoploid, referred to as pseudo-tetraploid (Supplemental Fig. 2)." Lines 419-420: "All species/individuals investigated in this study are functional diploids with a tetra-, hexa-, or octoploid origin (Fig. 1-3, Supplemental Fig. 1,2,4)."

-> Regarding ploidy, we emphasize that the term *functional diploid* refers to the genomic architecture behaving as diploid in terms of recombination and gene expression, despite an

underlying polyploid origin (e.g., tetraploid, hexaploid, or octoploid ancestry). This is distinct from cytogenetic $2n = 2x$ diploidy. While the functional diploidization observed in *Clusia major* is backed up by various data (Fig. 2-4, Supplemental Fig. 2, 5, 6), we only make an educated assumption for the other species based on allele frequency distributions, ploidy levels, base chromosome number, proposed placement of events within the genus, and comparative observations in the GenomeBrowser. These points are now more clearly stated and toned down in the text, e.g. lines 414-417: “All species/individuals investigated in this study seem to be functional diploids (i.e., polyploid organisms that functionally mimic diploids in a way that, despite having multiple sets of chromosomes, genes may be silenced or expressed in a balanced manner) of a tetra-, hexa-, or octoploid origin (**Supplemental Fig. 2**)”. We also removed occurrences about the type of polyploidy as there is no solid evidence favoring allo- vs. autopolyploid origin and such inferences are anyway not pertinent to the data interpretation.

-> We greatly appreciate the suggestion to expand the discussion of diploidization with additional CCM lineages, and we encourage this in our outlook (lines 444-448). However, we believe this extends beyond the current scope of the manuscript and we look forward to performing a more detailed analysis in this direction in future work.

-> We also realized from the comment that there may have been some confusion regarding the workflow details (e.g., which assembly state was used for which analysis). We apologize for any ambiguity. To address this, we have compiled a new table that outlines the workflow in detail:

a) Supplemental Table 3, sheet “Assembly (v1)”, presents statistics for all primary assemblies at the contig level (after FALCON Unzip), which are referenced in the second chapter of the manuscript (“Clusia genomes vary in size, chromosome number, and ploidy level”).

b) The new sheet “Workflow stats (v2)” in Supplemental Table 3 now provides step-by-step metrics for all assembly types, including contig/scaffold counts, genome size, N50, and BUSCO scores at each stage (from the draft assembly (v1) to the curated version (v2)).

c) We have also added a README.md file describing all FASTA files included in the Supplementary Data on Figshare (<https://figshare.com/s/ccfabd5785fab31512fc>).

-> This documentation clarifies the distinction between pre- and post-curation genome sizes and explains how the different assembly versions were generated and used throughout the manuscript.

Lines 185-186 “We believe that these differences in diploidization and, thus, different genic landscapes are connected to the different physiotypes in the genus *Clusia*, with respect to their photosynthetic behavior.” The hypothesis that CAM pathways have evolved as a consequence of diploidization after WGD is an intriguing hypothesis. However, this is a general and well-known phenomenon that leads to innovation in plant genomes across an array of phenotypes and lifestyles. Therefore, how is the process different or specific for CAM?

-> There are no differences in terms of their mechanisms. Mechanisms are typically shared, just different mechanism combinations are used by different plant groups. So it depends on the ecological niche or selective pressures for their specific habitat and abiotic conditions at time of evolution (lines 425-428). To address these patterns broader analyses of more CAM-species of the group would have to be compared to make further conclusions.

Reviewer #2 (Remarks to the Author):

I was reviewer 2 in the first round of this manuscript.

Below, I am attaching my original comments, and your responses. My replies to your responses are indicated by a -> sign, to differentiate them from the first round of reviews.

 Our new answers are underlined.

Unfortunately, overall, I do not feel that the authors have provided satisfactory responses to my comments. I am confused by their response to my comments regarding the link between polyploidy and CAM – and I feel that they contradict themselves in their response.

Furthermore, I have found that there is a major lack of clarity in the manuscript and that I often struggle to assess the conditions that were employed for each experiment. This lack of clarity has been highlighted to a greater extent in the second round of peer review, in part due to the addition of new supplementary figures that do not correspond well with the main manuscript.

Finally, I still do not find that the authors provide sufficiently compelling evidence to show that *C. multiflora* is a weak CAM species. Additionally, after getting permission from the editor, I have sought a second opinion from another scientist with extensive experience working with weak CAM. I include their comments at the bottom

Reviewer #2 (Remarks to the Author):

*This study represents the first sequencing and assembly of three *Clusia* genomes; a (hitherto considered) C3, a facultative and an obligate CAM species. This study uses these genome assemblies, in addition to information about chromosome number and genome size to speculate to the role the polyploidisation played in the evolution of CAM in *Clusia*. Overall, I think that this manuscript is an extremely impressive body of work, and represents a big step forward in the study of CAM evolution. However, I do have some reservations that I outline below.*

Comments:

*1. I feel that, at times, you are not using the word 'plasticity' with sufficient rigour. It seems like there are points throughout the manuscript where you use this word as a synonym for 'physiologically diverse'. Plasticity is the capacity to change a phenotype in response to environmental stresses. Plasticity itself can vary across a genus – for example consider two closely related species – *Clusia minor* and *Clusia pratensis*. Both exhibit plasticity in their photosynthetic physiology, as both can facultatively induce CAM. However, often the former species exhibits some amount of CAM under well-watered conditions, whereas the latter does not. Therefore *C. pratensis* displays more plasticity than *C. minor*.*

Agreed!

Instead of plasticity, we opted to use the terms “physiotype diversity”, “ecophysiological diversity”, or just “diversity” wherever appropriate.

-> Great!

#####

2. I think that the chosen species used in this study are excellent, as they represent three widely-distributed, well-studied *Clusia* species. However, studying *Clusia* can be somewhat fraught due to misidentification of species in various botanic gardens and plant collections. For example, *C. rosea* has been found to have extremely diverse carbon isotope ratios in this paper (DOI 10.32615/ps.2022.018). This is either due to misidentification, or extreme intraspecific variation – the answer to this is still unknown. In addition, IKEA sells a species of *Clusia* that they call *C. rosea* – although the leaf shape of this species looks completely different to the *C. rosea* that I am familiar with. Likewise, *C. minor* was only recently shown to be a distinct species from *C. pratensis*, but collections that predate this taxonomic classification may contain the latter, described as *C. minor*. And finally, *Clusia multiflora* is often considered to be a ‘species complex’ and can look quite different in different locations.

For all of these reasons, I think that it is important that you provide as many photos as possible to accompany the genomic data. These should include photos of flowers, leaves and stems, and should form a supplementary file. This will ensure that future users of your genomes will be able to make good, well-informed use of your data.

We used the labels that were given to these plants by the Ulrich Lüttge group in the greenhouse at Darmstadt. We most certainly did not use commercially available material. Additionally, we had several experts in neotropical botany re-examine the plants before setting out to sequence the genomes and we are quite confident in their assessment and the previous identification by the Lüttge group.

Nevertheless, we agree with the reviewer that it will be very helpful for the community and other future research to create less ambiguity. We therefore provide a plethora of selected photos as a supplemental resource deposited on figshare (private link for peer-review: <https://figshare.com/s/ccfabd5785fab31512fc>). Unfortunately, *Clusia rosea* never flowered in the greenhouse during the many years of cultivation. Therefore, we could not include photos of these. We did, however, include photographs of the leaves, stems, and the overall habitus.

-> Thank you for adding these photos. These plants all look nice and healthy too! With regards to *C. rosea*, whilst there are no flowers, it is nice to at least see the shape of the leaves, as this differentiates from the plants being sold as *Clusia rosea* by IKEA. However, I think that it is really important that you get a photo of a fully open *C. multiflora* flower. There is considerable ambiguity regarding the identity of species that are present in European *Clusia* collections. Considering that your genome is primarily based on *C. multiflora*, it is integral that you unambiguously confirm that this is the correct species. You mention that you consulted taxonomists? Are they authors on this paper? Can you provide the criteria that they used to identify these species, particularly *C. multiflora*? Also, were they aware of the recent categorisation of *C. minor* and *C. pratensis* as different species (rather than synonyms) and were they able to confirm that you are definitely working on the former and not the latter? I think more is needed here to unambiguously confirm that *C. multiflora* and *C. minor* are correctly identified.

 We deeply appreciate the reviewer’s comments regarding species identification—this feedback proved to be among the most valuable we received during the review process. *Clusia* is a notoriously complex genus with a rapidly evolving taxonomy, and your observation prompted a thorough re-evaluation of our material.

 In response, we consulted an additional taxonomic expert in Neotropical flora, Dr. Andreas Berger (Natural History Museum of Vienna), who is now a co-author of this revised manuscript. In collaboration, we conducted an in-depth barcoding analysis, sequencing established marker regions across all our studied individuals. These sequences were compared against reference datasets from herbarium material and other curated *Clusia* collections. This reevaluation led to a crucial finding: our genome previously referred to as *Clusia multiflora* has been misidentified. Following Dr. Berger's expert assessment and additional morphological analyses (including photographic documentation, voucher specimens, and comparisons of floral traits), we have now correctly identified the species as *Clusia major* L., not to be confused with *C. rosea* Jacq.

 We recognize the significance of this correction. The challenges of *Clusia* taxonomy—particularly the misapplication of names in cultivated and scientific collections—have become a central theme of our revised manuscript (see Fig 1, lines 76-96 in main document). We discuss this in more detail as part of a new Supplemental File 1, including phylogenetic trees, identification criteria, and trait-based comparisons.

 In response, we now provide what we believe to be the first reference genomes of *Clusia* species with fully documented, voucher-backed identifications. All specimens have been digitized and deposited at the Natural History Museum Vienna (herbarium code W), and their images are available via JACQ.org: *Clusia major* L. (W0391620), *Clusia minor* L. s.l. (W0391623), and *Clusia rosea* Jacq. (W0356382).

 We now discuss the implications of this reclassification in the broader context of *Clusia* systematics. As the reviewer rightly noted, accurate species identification is a prerequisite for interpreting physiological and molecular data. Indeed, *Clusia major* has previously been described by Ulrich Lüttge as a weak C₃/CAM intermediate—a phytotype consistent with both our experimental observations and molecular findings. Prof. Lüttge has also joined the revised author list. Unlike *C. multiflora*, *C. major* is already established in the literature as a C₃+CAM species—making our findings more consistent with existing knowledge. This should address the reviewers' major concern regarding the wrongful CAM classification of a bona fide C₃ plant (for a very detailed explanation please refer to **Supplemental File 1**).

 We believe this integrative correction strengthens our work significantly and raises broader awareness of taxonomic uncertainty in physiological studies. Finally, this corrected identification of *Clusia major* also offers an opportunity: it may help reconcile apparent discrepancies between physiological and molecular assessments of CAM in our previous version. As a weak C₃/CAM species, *C. major* provides a valuable model for exploring intermediate forms of CAM expression.

#####

3. I think that you have overstated the importance of polyploidy for the evolution of CAM. Many examples exist, such as in the abstract:

“Our findings reveal that polyploidization during genus evolution and subsequent diploidization shaped the emergence of extant C₃+CAM subtypes in *Clusia*”

And there are other such statements throughout the text. At best, you have provided a hypothesis that progression across the C3-CAM 'continuum' requires gene(ome) duplication. However, the data you provide is not strong enough to robustly support this claim. Firstly, you have only looked at the genome size/chromosome count for 3 species, from disparate parts of the Clusia phylogeny. This makes your analysis subject to biases, as differences in genome size may have nothing to do with CAM. In fact, in the PhD thesis of V.A. Barrera Zambrano, the 2C DNA content for 9 species of Clusia was measured, including the three species you have assessed. The link to this thesis is <https://theses.ncl.ac.uk/jspui/handle/10443/1534> - see page 152. They showed that Clusia rosea was in fact somewhat of an outlier, and that for other constitutive CAM species, such as C. hilariana, there wasn't a particular difference in the 2C DNA content when compared to the C3 or C3/CAM species. I think that it is important that you take the data presented in this thesis into consideration when you frame the interpretations of your data.

Secondly, I am not sure I completely appreciate why a constitutive CAM phenotype would require a larger genome than a facultative CAM phenotype. Surely the capacity to do both C3 and CAM (and to be able to switch depending on the environment that the plant finds itself in) would require from a larger number of gene copies. For this reason, I think it is essential that you critically assess how important genome duplication really is for the evolution of CAM, and how much this might just be a coincidence.

We agree with the critical assessment of the reviewer that the role of WGD in the evolution of CAM in general was overstated and, thereby, subject to valid criticism. We amended the text at several instances (Abstract, Introduction, Discussion and Conclusion) to make sure we no longer overstate. We agree that the data do not provide clear evidence to support the claim that polyploidization was necessary for CAM evolution in the first place, but they allow to formulate such hypothesis that can further be tested.

Also, we do not believe that a constitutive CAM phenotype would require a larger genome than a facultative CAM phenotype. All we mean to say is that by gaining multiple copies of a given pathway at some point in time, this creates "wiggle room" for neo-, sub-, and nonfunctionalization. Especially for strong CAM, polyploidization would provide/enable all necessary mechanisms for the required metabolic reprogramming to evolve, which can barely happen via single or tandem gene duplications alone.

-> Why 'especially for strong CAM'? What would strong CAM require a greater genetic neo- or sub- functionalisation than facultative or weak CAM? They all require the same changes to temporal expression patterns, just some have higher expression than others. I don't really follow this argument. You seem to have directly contradicted yourself here.

Therefore, WGDs are important drivers of added diversity (by increasing copy numbers of the genes) at certain points of an organism's evolutionary history. Since the post-polyploidy evolution of genome size (C-values) may either result in up- or downsizing (especially when either TEs or tandem repeats dynamics change), regardless of genic diploidization patterns, we have perhaps not been specific in previous version when referring to "diploidization". We do not refer to genome size changes, but to genic diploidization and accompanying regulatory and metabolic changes. What we emphasize is that the ancient WGD has been instrumental for the extant diversity in physiologies, which was mediated by a concurrence of (genic) diploidization with selective pressures under disparate climatic conditions.

We strongly believe that the combination of WGD and subsequent diploidization processes are the key drivers behind the phenotypic diversity we observe in Clusia. To be clear: We

only provide evidence for the case of *C. multiflora*. This allows us to formulate a hypothesis to be tested in other species of the genus. We tried to clearly separate speculation from evidence in that respect and hope that the text now reflects our views more accurately.

-> Overall, I am not completely convinced by your response here. You start by saying that you agree with my assessment that the role of WGD was overstated, but then at the end of this paragraph you double down and say that you 'strongly believe that the combination of WGD and subsequent diploidization processes are the key drivers behind the phenotypic diversity we observe in *Clusia*'. Are these two statements not in contradiction of each other? Also, in my last round of review it was essentially this latter statement that I said lacked robust evidence.

-> Why has the updated manuscript not included the thesis I mentioned in my last review as a reference? For me this is one of the most important references for your study and should be addressed directly within the text. It is also worth mentioning that the 1C values that you estimated did not correspond that well with the 2C values in this thesis (particularly for *Clusia minor*). Seeing as the plants studied in this thesis were also from the Luttge collection, a direct comparison between those data and yours should highlight that there can be some ambiguity in the estimation of C values.

-> My main issue is that I still don't accept your interpretation here. You say that WGD events are important for neo-functionalisation of genes – this I think is fair. But you don't provide any data that robustly shows that WGD, or any other structural genomic change, has been important for CAM. Bearing in mind that there is so much genetic redundancy in vascular plants, what's to say that the genomic/genetic conditions required for the evolution of CAM were not present before the WGD event? Just showing that a WGD event happened in the early evolution of *Clusia* and that CAM evolved in this genus is not particularly strong evidence, and statistically would be an n=1 situation. To really show that WGD is important for CAM, you would have to show that it occurs more in lineages that evolved CAM, than in lineages that didn't. As it stands, you are simply noticing the co-occurrence of WGD and CAM in a single lineage. In Fig. 6a, you show many occurrences of polyploidisation in lineages that did not evolve CAM. Considering the frequency that WGD occurs in plant evolution, which you highlight in this figure, I don't see how you can make any claims that there is a causal link, based on the data you provide. Also, how many isoforms of PEPC were present in the ancestral genome before the WGD that occurred early in *Clusia* evolution? Can you provide any quantitative data or analyses to support the argument that neofunctionalization could not have led to CAM prior to this WGD event? Unfortunately, based on these considerations, I feel that you do not provide sufficient support for the claim made about WGD playing a role in the evolution of CAM.

 On the topic of WGD:

 We appreciate the reviewer's close reading and acknowledge that the phrasing in our previous response may have caused confusion. The use of the word "especially" was indeed imprecise in this context, and we apologize for any lack of clarity. We would also like to note that as non-native English speakers, occasional linguistic missteps may occur in complex argumentative passages, though we take full responsibility for the meaning conveyed.

 To clarify: What our data **do support** is the idea that **diploidization after WGD** plays a significant role in **driving diversification among CAM phenotypes**. In other words, once CAM has evolved—whether in a diploid or polyploid background—the subsequent reorganization of the genome during diploidization creates evolutionary flexibility. This includes possibilities for regulatory divergence, expression shifts, sub-/neofunctionalization,

and even gene loss, which together can result in distinct CAM intensities and physiologies among closely related species.

 This pattern is reflected in our data from *Clusia* species, where we observe that species with a shared polyploid ancestry exhibit clear differences in CAM strength and expression patterns. Thus, we interpret diploidization as a process that contributes to **post-CAM diversification**, rather than to CAM origin.

 Here are passages from our manuscript that underline these statements:

Lines 439-443: “The consecutive occurrence of polyploidization and diploidization would provide a general mechanism for the nascent expression of (pre-)CAM elements in C3 plants, which, once established, emerges a qualitatively distinct phenotype as required for the metabolic reprogramming that links breakdown of storage carbohydrates to the process of net dark CO2 fixation in strong CAM plants.”

Lines 428-430: “As post-WGD adaptation proceeds, selection to maintain dosage balance and stoichiometry will relax over time, allowing duplicated networks to be rewired and to evolve novel functionality and increase biological complexity.”

 In summary, our key point is that:

- **WGD may or may not be involved in the origin of CAM** in a given lineage:
- **But if a CAM lineage undergoes WGD (or is already polyploid), then the process of diploidization can drive divergence in CAM phenotypes** across evolutionary time:
- **This model is supported by our comparative genomic and expression data in *Clusia***, and we now frame it explicitly as a contribution to the understanding of CAM diversification—not CAM origin. This also led us to change the title of the manuscript: “*Clusia* genomes shed light on the evolution and **diversity of CAM phenotypes**”

 **On the Inclusion of the Thesis Reference**

 We included the thesis reference and explicitly acknowledge the discrepancy in 1C values for *C. minor* and highlight potential ambiguities due to taxonomy.

 **On the Generalizability of Our Hypothesis**

 We agree that demonstrating a causal link between WGD and CAM would require broader comparative studies across CAM and non-CAM lineages. Our current data support the hypothesis within *Clusia* and motivate further testing across taxa. We have revised the manuscript to make this distinction clear and avoid overgeneralization.

#####

4. I am confused about why your physiology experiments did not employ standardised treatments. Unless I have misunderstood your manuscript, for your greenhouse experiment, you have sampled from ‘control’ plants that experienced quite a lot of shade, and were watered, and then stress plants that experienced a lot of light, and water was withheld. Also,

C. multiflora received more light than *C. rosea*. Why is everything so different? Also, in the *C. rosea* 'control' plants there is a peak in light intensity at about 1.30 pm, that is not observable in the *C. multiflora* plants, nor does it correspond to any peak in sunlight that can be inferred from the 'stress' plants (also Fig. 4a). Can you explain why all of these inconsistencies are built into your experimental design?

Yes, we can explain the apparent disparities. We conducted two separate sets of experiments. One "open greenhouse experiment" and one "physiological measurements" experiment.

In the first experiment, which displays these differences, and we refer to as the "open greenhouse experiment", we grew plants under natural sunlight to achieve very high light intensities, which we would not be able to reach with artificial lighting in our climate chambers at the time. The roof of the greenhouse was opened for that purpose, and we were reliant on the environment for the light conditions, so, an occasional cloud and the angle with which the sun hits can influence a given datapoint explaining the different amounts of light that were measured. The natural global irradiation was thereby subject to fluctuations. Additionally, the 'control' plants were grown right there, too, but we built a shaded stand for them. The shading mats have slits and openings to let wind pass through. If the sunlight passes directly through one of the openings and hits a sensor, this results in an apparent increase in light which would have affected just a tiny portion of the shaded area but looks like an overall light increase for the condition. The plants and the sensors were, however, always positioned in a randomized fashion. Any differences are mere sensor position coincidence and are, in our opinion, not generally reflective of the actual amount of light all plants received throughout the day.

-> This is not a completely satisfactory answer. Firstly, your explanation does not indicate why there was such a pronounced difference in the light intensity between species. In Fig. 4a, the 'stressed' plants of *C. multiflora* consistently get more light than *C. rosea*, which is most pronounced at 10 am (with a staggering difference in PAR of $300 \text{ } \mu\text{mol m}^{-2} \text{ s}^{-1}$). Why is this? This cannot be because of light momentarily passing through shades, because these are the plants that are under high light. Also, does this indicate that your randomised experimental design has failed? If it is because the light sensors used are biased by their positions, then this data is essentially meaningless, and provides no value to the manuscript.

-> Additionally, it is a poor experimental design to change two environmental conditions (water and light) when studying photosynthesis, especially when working with variable conditions imposed by natural sunlight. Is a light intensity of $600\text{-}900 \text{ } \mu\text{mol m}^{-2} \text{ s}^{-1}$ truly stressful to these plants? It's not such a high light intensity, that it is intuitive to me that this will cause light-stress. Can you provide A/Q curves, and NPQ data to show that this light intensity is very stressful for the plants? The reason I ask this, is because as it stands you have changed the light intensity alongside the water availability and then measured photosynthesis at a biochemical/molecular level. But your supposed 'stress' conditions also involve giving the plants a huge amount more-light, so surely this will always have an influence on photosynthesis, that has very little to do with stress? For example, if I studied the difference in photosynthetic rate in a plant grown under well watered and salty soils, but I also changed the light regime for these two groups of plants, then how much could I really know that the differences are because of salt stress vs because of light intensity?

-> Unfortunately, the issue raised here regarding the non-standardised experimental design has a knock-on effect on the data presented in Fig. 5, as these data seem to be generated after sampling from the greenhouse experiment. As the experimental design resulted in

variable conditions for each species * treatment combination, it is very difficult to assess how much these random effects are contributing to the data vs the effect of the species and/or drought stress. I will comment more on this in my response to your use of molecular data to phenotype *C. multiflora*, below.

-> Overall, I find that this experimental design is very flawed, and leads to difficulties interpreting much of your metabolic data. I also note that you don't have corresponding gas exchange data to go alongside these sampling efforts. Whilst you have done gas exchange under controlled, standardised conditions, you do not have any gas exchange to show what these plants are doing under the less controlled greenhouse conditions. I suggest that you perform gas exchange for each species, under the 4 different experimental conditions that were used for sampling each of the species*treatment combinations. Only then can the reader get a sense of what these plants are doing. However, unfortunately this would still not improve the flawed experimental design used in this experiment, which would continue to limit the robustness of your conclusions.

 Regarding the criticism of the experimental design of our open greenhouse experiment: Conducting the open greenhouse experiment was a conscious decision made to capture realistic physiological responses under close to natural conditions and accordingly has higher environmental fluctuations. The experiment still illuminates the phenotypic diversity of the chosen *Clusia* species. The light intensity differences observed (e.g., in former Fig. 4a, now figure 5 b and c) are a result of a randomized setup: We used the PhotosynQ device for measurements of PAR when sampling a random plant and the sensors may have occasionally been shaded by leaves, structural elements, or clouds. Seeing as we couldn't sample all leaves at exactly the same time there might have been an occasional cloud influencing measurements. Since we selected only three plants per species and condition for downstream analysis stochasticity has a rather pronounced effect on the data shown. While we understand the reviewer's concern, the variation is not evidence of flawed design, but rather a consequence of conducting an outdoor experiment under natural sunlight—a conscious decision made to capture more realistic physiological responses. We lacked access to climate chambers with sufficiently controlled conditions and thus opted for the best available compromise.

 Importantly, for the molecular analyses shown in Fig. 5/6, we specifically selected plants that had experienced similar light conditions for most timepoints. To further address this point, we revisited our data from the PhotosynQ device, which captured Φ NPQ (PhiNPQ) at incident light intensities for all plants used. These data confirm that the samples used for multiomics were under comparable light regimes in terms of the physiological response that they triggered, and we include these plots in the revised manuscript (Fig. 5d). We also mention this in the revised text (lines 290-294).

 We also want to emphasize that this study is primarily genomics-focused. Our aim was to bridge computational predictions with physiological relevance, and we firmly believe that experiments conducted under semi-natural conditions contribute meaningfully to this goal—even if they lack the strict control of growth chambers. In fields where molecular biology and ecology intersect, some degree of environmental variability is expected and, in our view, acceptable for generating hypotheses and biological insight.

 Regarding the request for gas exchange measurements under each of the four greenhouse treatment conditions: this is not feasible. The available instrumentation (a single WALZ GFS3000) allows for only one plant to be connected at a time, and given the scope of

the study (48 plants in the beginning), this level of replication would be prohibitively time-intensive. We believe our new standardized gas exchange data under highly controlled conditions, which now align well with our semi-natural data in the open greenhouse experiment, sufficiently support the core findings, as outlined in more detail below.

The “physiological measurements” experiment was conducted in a completely controlled environment (see comments below).

Also, it would be very useful if Fig. 4a was split based on the ‘control’ and ‘stress’ plants, because at the moment the ‘stress’ plants’ PAR is so much higher that is hard to tell how much light the ‘control’ plants get.

Thank you for the suggestion. For good interpretability, we supply the separate graphs as Supplemental Figure 10.

-> This has not increased my trust of these data. The graph you show for Fig. S10a does not correspond to the data shown in Fig. 4a, in a number of different ways. 1) the peak light intensity in Fig. 4a is at 10 am, whereas the peak in Fig. S10a is at 1 pm. 2) the light intensity experienced by *C. rosea* is greater than that experienced by *C. multiflora* in Fig. S10a, whereas it is the other way round in Fig. 4a. Overall, these are clearly not the same data. Also, I don’t understand why your error bars for Fig. S10a go below 0, as this is not physically possible.

 We revised the data, removed figure S10 and provide the data as part of main figure 5.

*Finally, for the controlled growth chamber experiments (by the way a description of this is missing from the methods), why do the *C. multiflora* plants get less light than the *C. rosea* plants? And why is this the opposite of what was done in Fig. 4a?*

We apologize for not having included a description of this part in the methods. It must have slipped out in-between versions. We made sure to include these methods in section “Plant cultivation and physiological experimentation” and also list environmental conditions in the new Supplemental Table 8. Thank you for the thorough analysis.

In fact, the differences in light intensity in the controlled experiments have to be corrected. After discussing this, we revised the original raw data and realized that it was both an axis labelling error, caused by forwarding and copying the tables without the proper SI unit labels, and a data error (wrong sensor for *C. rosea*). We scaled the axes using a multiplier (10x for *C. multiflora*) so the actual value should be $400 \mu\text{M m}^{-2} \text{s}^{-1}$. For *C. rosea* the wrong values (PAR_{top} instead of PAR_{ambient}, see below) were supplied. The actual light intensities are in both cases at the same level ($\sim 400 \mu\text{M m}^{-2} \text{s}^{-1}$). Both plots are now corrected in Figure 4. We have also rechecked all other data to make sure that the data presented are absolutely correct.

-> thank you for correcting this. I suggest that you also tweak these graphs so that the x-axis for both 4c and 4f are the same length. This probably would involve having the x axis a little

longer than the true length of the gas exchange, but this extra time can just be left black. I think it would just make it easier to compare these two graphs, if their x-axis aligned. Also, as mentioned above, it would be useful to include a line where $y = 0$.

 The new gas exchange plots were redesigned (Fig. 1b,c).

Finally, Fig 4a and 4d shows the experimental conditions under which samples were collected. The rest of figure 4 doesn't have any mention of C. minor. But then in figure 5, data seems to have been collected for C. minor. Please can you explain why data is sometimes present for C. minor and sometimes absent?

Clusia spp. are quite difficult to work with on a molecular level. Their recalcitrant nature required us to adapt and improve almost every step of every protocol we used for both nucleic acid and protein extractions. To ensure comparability between species, we aimed to establish a protocol for each extraction that would work for all three plant species. In the case of the gDNA, we solved the contaminated DNA issues by pre-isolating nuclei. This allowed us to produce good assemblies for all three species. Even after rigorous testing, however, we were not able to apply the same protocol for RNA isolation to all three species. We have therefore settled on the protocol that worked well for both *C. multiflora* and *C. rosea*, but was not as efficient for *C. minor*. Unfortunately, we were not able to sequence the resulting cDNA library for *C. minor*, and have therefore lost the sample available. For metabolomics and proteomics, our extraction procedures worked quite well for all three species. We have decided to add the data obtained for *C. minor* as the species is quite interesting in terms of its C3+CAM behavior, and to publish its genome alongside the others. We believe that the metabolomics and proteomics data might be useful for better interpretation of other data in the manuscript and for the community.

-> This is very fair. Some mention of these considerations should be included in the methods section.

#####

5. *I think that a bit more detail/nuance is needed in your description of the phenotypes of the three species that you studied – especially for C. multiflora.*

In the case of C. rosea, it is worth noting that even this 'constitutive CAM' species does a fair amount of C3 (i.e. has relatively pronounced phases II and IV) under well watered conditions. For example, doi:10.1093/jxb/eru022 showed that C. rosea did more C3 assimilation than either C. hilariana or C. alata, suggesting that this species is somewhat on the edge of being considered a facultative CAM species. Also, as I mentioned above, evidence from carbon isotope surveys have suggested that some individuals in this species may primarily be fixing carbon via C3 (if these herbaria samples are believed to be identified correctly). This ambiguity should also be mentioned in your background, and then you could

show that under your growth conditions C. rosea displays a pretty strong CAM phenotype (clear phase I, no/little phases II or IV).

We made sure to mention this ambiguity in the text and cited the referenced article. In fact, we can confirm this observation. With this resubmission, we now provide 3–4-day continuous gas exchange measurements graphs highlighting the transition from cultivation conditions to stress treatment conditions to show the quite striking change in CO₂ uptake behavior. We performed such analyses also for *C. hilariana*. We wanted to discuss these data in a subsequent paper, but we understand that this may be critical for better understanding of the current manuscript. Therefore, we included the findings in the body of the text (lines 318-337) and as part of a new Supplemental Fig. 13.

-> I am really struggling to get the relevant information that is needed to critically assess your physiology data. As an example, I will describe the process I just went through to try to understand what you have done. I read in your comment, above, that you 'provide 3–4-day continuous gas exchange measurements graphs highlighting the transition from cultivation conditions to stress treatment conditions'. So this made me wonder if your gas exchange data in Fig. 4c and 4f, are under well-watered or drought-treated conditions. Are you saying that 3-4 days is enough time to induce a full drought response (it is probably not considering that it was not enough time in <https://doi.org/10.1016/j.jplph.2024.154185> to see any meaningful change). Anyway, I then took another look at fig S13, as your comment above was about *C. rosea*. I noticed that the gas exchange data trace did not resemble that presented in Fig. 4c. So I checked the materials and methods to see what you have said about the conditions for this experiment. But there is insufficient description here, and it sends me to supplementary table 8. When I check this table, there is no mention of any species name, just genus, which is *Clusia* for all experiments. Also, looking at this table makes it seem as though the sampling done for fig. 5 did not come from the 4 time points indicated in Fig. 4a (i.e. the stars) but was in fact done in several different experiments over the course of many years. Is this the case? I'm sorry to be so difficult, but I am finding the lack of clarity in this manuscript so obstructive that I really cannot understand what it is that you have done. It is extremely difficult for me to highlight all of the missing clarity, because you have loaded the manuscript with so many supplementary figures and tables, many of which are inconsistent with the main manuscript (e.g. S10 is different data to 4a; and the gas exchange traces in s13 do not correspond to those in Fig4). I think you need to heavily edit the entire thing, including the supplementary data and figures, because as it stands I am becoming increasingly aware that it is not possible to understand what you have done.

In the case of C. minor, it is worth noting that often even under well-watered conditions this species does a fair amount of weak CAM. (E.g. doi:10.1093/jxb/eru022). You don't provide gas exchange data for C. minor, so it is hard to tell exactly what this species is doing under your growth conditions. This makes interpreting the metabolic data a little trickier. It would be nice if you could add this analysis to your data set.

We also provide gas exchange data for *C. minor* as part of Supplemental Figure 13.

-> Why is this data so noisy, and what are the environmental conditions under which it was measured (i.e. when was it watered). This is particularly relevant for this graph, as it looks like assimilation has almost completely stopped for this plant.

Finally, I am not convinced by your phenotypic characterisation of C. multiflora, as a weak CAM species. I think that to characterise this species in this way, you must first provide compelling phenotypic data, and then the molecular (RNA-seq proteomic etc) data can come in as a reinforcement. At the moment, your physiology data is not sufficiently strong, despite the molecular signals pointing towards some form of weak CAM in this species.

We performed a couple of additional experiments to answer the questions raised point by point in the sections below, leading us to change the characterization to constitutive C3+CAM.

Your gas exchange does not show indications of weak CAM. Typically, when nocturnal respiratory CO₂ is being recycled, you see a small 'bulge' in the nocturnal CO₂ efflux. Sometimes, this 'bulge' pushes assimilation from a net negative to positive value for part of the night, but sometimes it just involves a reduction to CO₂ efflux, peaking at some point during the night. For example, Clusia pratensis often does this under well-watered conditions (see <https://doi.org/10.1093/jxb/eru063>, <https://doi.org/10.1016/j.jplph.2024.154185>, and <https://doi.org/10.1071/FP20268>) and also in other taxa (see <https://doi.org/10.1071/FP20151>, <https://doi.org/10.1071/FP20127> and <https://doi.org/10.1093/jxb/ery431>). The gas exchange data you show does not exhibit this tell-tale sign of weak CAM, which makes me question whether it really exists in C. multiflora.

While the mentioned bulge is more pronounced in e.g. Pilea peperomioides, we generally do not observe it prominently in Clusia. All species show relatively constant dark period CO₂ loss rates just like the C₃ and C₄ reference plants. Instead, the CAM phases II (dawn) and IV (dusk) of PEPC activity are much more decisive as covered by the continuous gas exchange experiment visualized in Supplemental Fig. 13. The proposed emergence of the typical CAM curve during transition is also prevalent in the mentioned papers of Clusia pratensis.

-> OK here I must confess that in my last review I missed an essential detail in figure 4F. Whilst there is no 'bulge' in the nocturnal net assimilation trace, there does seem to be a (small) net positive A value during the night, which is also indicative of weak CAM. I think that this would be much clearer if you included a line to indicate y=0 on this graph, as you have done in Fig. S13. I also think that you should explicitly explain that your gas exchange provides some evidence of weak CAM, and make reference to the other Clusia papers I mention above, in order to highlight that your evidence differs to much of the published understanding of weak CAM in this genus.

-> That being said, I would also like you to comment on the precision of your gas exchange instrument to pick up on small fluxes, when stomata are closed. How often did you match the IRGAs (and why is this info not included in the materials and methods)? In Fig. 4c, there is a net negative transpiration rate at the start of the night. This is also evident in many of the graphs in Fig S13. Do you think that this is real? And if it is an artifact, can you truly be confident that your assimilation data when fluxes are so small (e.g. C. multiflora at night) are robust?

In addition, we believe that the molecular data is amongst the most important pieces of evidence here. The molecular data provide evidence that C. multiflora behaves very much like a CAM plant. With several independent experiments (see below) we have now shown

that 1) *C. multiflora* accumulates malic acid during the night under controlled conditions and, 2) that organic acid accumulation acidifies the leaves, which is considered to be one of the most important phenotypic traits of CAM. Taking these observations and the molecular data together, we believe this is ample evidence to classify a plant as constitutive C3+CAM (weak CAM or CAM-cycling).

-> But these data are based on a flawed experimental design, under which each species * treatment combination received different treatments. Also, you have no gas exchange to go alongside these greenhouse sampling efforts, so we don't know what the plants were doing under those conditions.

-> also you put emphasis on the PCA plots in Fig. 4b and e being similar, as evidence that *C. multiflora* does some weak CAM. Firstly, these PCA plots are not so similar. Furthermore, the data largely clusters due to the time/treatment for each species, which just indicates that the experiment has had some impact on the transcript profile, in a consistent way (within each species data set). It doesn't indicate anything about the photosynthetic phenotype of *C. multiflora*.

*Furthermore, your malate content (Fig. 5e) shows some accumulation of malate that coincides with dawn. I have a few comments on this. First, it would be more useful to show malate as a concentration, on a per leaf area and per fw basis. Having both would give a more comprehensive view of these data. Note that in a related species *Clusia tocuchensis*, there is a considerable change in the RWC of the hydrenchma tissue over a 24h period (<https://doi.org/10.1111/pce.14539> – Fig. S8). So there is a possibility that only reporting metabolite concentrations on a FW basis could be biased by this phenomenon. If a combination of low hydraulic conductance and high hydraulic capacitance results in considerable water losses when stomata open in the morning, this could drive the water content of leaves down, and result in an increase in malate content on a fw-basis, without any actual increase to malate content on a leaf-area basis. Maybe this is not the case, but as it stands it is not possible to make that assessment.*

We always sampled the same leaf area. As described in our methods: “We used sterile punch pliers with a 5 mm diameter to punch four holes into one leaf of each plant.” With this in mind we went back to see whether the biomasses (FW/leaf disc) across a day changed significantly and could find no influence of daytime on the leaf FW for *C. multiflora* (see figure below).

*Secondly, it is unusual that the accumulation of malate only seems to occur right at the end of the night/around the beginning of dawn (Fig. 5e). These data imply that all the weak CAM activity is limited to the very end of the night, as there doesn't appear to be any difference between 7 pm and 4 am. This is quite interesting, but also makes me wonder whether malate accumulation has occurred in the dark (i.e. between 4 am and 6 am) or only in the first part of the light period (6am to 7am)? I can see from Fig. 4f, that the very beginning of the light period is characterised by an increase in assimilation. Could there be some PEPC-mediated carboxylation during this time? It is not impossible, because strong CAM species of *Clusia* tend to exhibit PEPC activity for a few hours into the morning. Seeing as your second sampling time (7 am) is 1h after dawn, could you be observing some flux*

through the CAM cycle that just during the morning?

As a matter of fact, this assessment is only true under non-stress conditions. If exposed to high-light and withheld watering there is a considerable increase in malic acid in *C. multiflora* between 7 pm and 4 am, even more pronounced than that between 4 am and 8 am (Fig. 5e, right side), which would further speak against the water loss hypothesis to explain the malic acid increase and shows that *C. multiflora* has the capacity to produce malic acid before dawn.

To address the comment in more rigorous way, we performed another set of experiments. We combined the measurements of malic acid in *C. rosea* and *C. multiflora* with pH and H⁺ measurements to assess the contribution of organic acid accumulation to leaf acidity. This time we did so under highly controlled conditions with the first sampling timepoint at 6 am (before dawn). The new results show consistently that there is considerable malic acid accumulation between 8 pm and 6 am in both control as well as treatment plants, which reduces during the day. In line with this observation, the pH values are low at 6 am and increase during the day. We incorporated these data into the new version of the manuscript as part of Supplemental Figure 12 and into the text in lines 311-317.

-> This isn't really performing a new set of experiments, but rather combining several data sets that were generated over multiple years and depicting them as a single experiment. Based on my comments above about the lack of transparency/clarity, this is a major issue for me. Were these different years of data collected under the same conditions? Were the plants the same size (this will effect their whole plant transpiration rate and thus their soil water status). Were the pots the same size? Why do the error bars fall below 0? What statistical tests have you done on these data? Considering that Fig. 12g depicts a tiny flux in malate, how confident can you be in the precision of your experiment? I think that combining data from different years is not an appropriate way to identify the presence of CAM when malate fluxes are this low, because these data will be so prone to small differences that could occur one year to the next. Also, why have you presented the 60 % LSD value for *C. multiflora* rather than the 90 % value as was done for everything else. Finally, and please forgive me if I have made a mistake here, but I took a ruler and looked at the error bars in each graph. It seems as though for each graph, every error bar is exactly the same size. This makes alarm bells go off for me, and these data needs to be checked. Regardless of this, unfortunately I think that the aforementioned issues are sufficiently great to make this graph lack any meaningful value in phenotyping *C. multiflora*.

-> My advice remains that to confidently conclude that *C. multiflora* does some CAM, you need to do a SINGLE experiment, under more controlled conditions, in which the sampling is done at a higher time resolution, for a full 24 hours. measurements of TA and malate for such an experiment would be strong evidence of whether *C. multiflora* does weak CAM. At the moment, your data is not strong enough, to make robust conclusions. Also, considering that your subsection entitled 'Diploidization of ancient polyploids underlies ecophysiological diversity of CAM' (starting line 391) relies on *C. multiflora* being C3-CAM for your current interpretation of your data, I would say that a lot of the conclusions in this manuscript are relying on weak data.

I think that based on the data presented so far, and also due to the complicated experimental design that I mentioned earlier, it is hard to be confident that C. multiflora really

is doing weak CAM, and I think you need some more data to reinforce this finding. I would advise you measure malate concentrations over a diel period (including earlier time points in the night). If it were me, I as close to the beginning of the light period as possible, and then 2 hours afterwards, to also see if malate accumulation is limited to the early morning. Also, I would advise that you do this under controlled growth conditions, with each species experiencing the same treatment. If there is a rationale for treating each species differently, this may be ok, but it must be more explicitly stated. I strongly suspect that you will find C. multiflora to be doing no CAM, but I would love to be proved wrong! However, as it stands I don't think your data is strong enough to conclusively show that this species does any weak CAM.

We hope that the new experimental data regarding malate, H⁺ and pH, the reclassification as constitutive C3+CAM instead of weak CAM, the lack of a water evaporation effect, and the continuous gas exchange measurement was sufficient to get the reviewer to agree with our assessment. We would also like to point out that the CO₂ compensation point of around 10 ppM (predawn, postdawn, noon, measured with A/ci curves) for C. multiflora is very low compared to C3 plants and represents another important phenotypic indicator of some degree of CAM. And finally, also the WUE (as calculated as net assimilation to H₂O stomatal conductivity relation) is with a 1:5 ratio a lot smaller than that of a C3 plant, estimated at around 1:15 (Supplemental Table 7, Supplemental Fig. 13).

We thank the referee for the very thorough and helpful review that allowed us to improve the manuscript!

 For clarity, we have chosen to respond to all comments from the final section in a single consolidated block. The reviewer expressed continued concerns regarding the adequacy of the data. In our attempt to address earlier feedback by providing additional data—as requested during the first revision—we introduced material derived from several years of working with these plants. However, we acknowledge that our integration of this material into the original manuscript structure was unsuccessful and resulted in confusion. It may have appeared as though we were overloading the reader with loosely connected data, and for that we sincerely apologize. We agree that this approach was ultimately unhelpful.

 As a result, we have now substantially restructured these sections and, more importantly, the way we present the data. We have removed all previously added material that lacked clarity or cohesion, including the gas exchange data from our initial submission. The revised manuscript now presents only two experiments—clearly separated to prevent any confusion between them.

 We continue to emphasize the importance and validity of the “open greenhouse” experiment. That said, we recognize that the gas exchange measurements associated with it were not sufficiently supported by detailed methods, and some figures were flawed. To address this, we have now repeated the gas exchange experiment under fully controlled environmental conditions. We subjected both C. major and C. rosea to the same CAM-inducing treatment. This updated experiment is described in detail in the revised Methods section. The results are shown in Figure 1, demonstrating CO₂ fixation during day for C. major and CO₂ fixation during night for C. rosea under highly controlled conditions.

 We have moved this section to an earlier point in the manuscript—now introduced with the species selection for genome sequencing (Supplemental File 1)—to clearly distinguish it from the later greenhouse experiment. The manuscript now presents two experiments only:

one under strictly controlled conditions and one under semi-natural conditions (see experimental parameters as part of a revised Supplemental Table 8).

 Importantly, we will retain the open greenhouse experiment. As stated previously, we are confident in its scientific integrity and value for interpreting our genomic and multiomics results. Beyond the substantial effort invested in generating these data (which remain only an adjunct to the genomics-centered narrative), we believe they provide meaningful biological context for the phenotypic diversity of the chosen *Clusia* species.

 Finally, we would like to note that the reclassification of *C. multiflora* to *C. major* carries important implications for the interpretation of our results and for the experiment itself. We acknowledge and appreciate the reviewer's valid concern regarding the initial classification of *C. multiflora* as a CAM species, and we are grateful for the rigor applied in highlighting this point. Given this taxonomic correction, the data presented here can be viewed in a new light. Specifically, it supports the possibility that CAM metabolism in the studied plant is indeed plausible. CAM has ancestrally developed in the revised clade and *Clusia major* has previously been described as a weak C₃/CAM intermediate. We believe this context strengthens the interpretation of our findings. We hope the reviewer will agree that this reclassification enhances the plausibility of our conclusions.

Minor comments

Line 54 – I'm not sure it is strictly true to say that CAM doesn't require structural changes in order to function. Even within *Clusia* there is a pretty substantial body of literature that shows that CAM is associated with anatomical adaptations. I think these need to be mentioned, or the text needs to be rephrased.

(See <https://doi.org/10.1093/jxb/eru022> ; <https://doi.org/10.1093/botlinnean/boab075> ; <https://doi.org/10.1093/aob/mcad035>)

-> Done

See

Line 431 – you need a reference to support this statement about environmental niches of these species.

-> Done

Line 75 – I think you should name the 'phenotype' of *C. minor* (i.e. facultative CAM) and also briefly mention that it does some weak CAM under well-watered conditions.

-> Done

Comments from a second opinion:

Start of message:

I can't say much about the genomics part of the manuscript, but I can say that the physiology part is not very strong, and this is worrying because it is largely the physiological experimentation that makes the authors conclude that *Clusia multiflora* shows some CAM. If true, this could have important implications for the understanding of CAM evolution in *Clusia*, because in none of the species of the *C. multiflora* complex has CAM been reported previously.

Given past misidentifications in the *Clusia* literature, at this point there is no guarantee that the authors' *C. multiflora* is indeed *C. multiflora*. Plants were received as a gift from another lab, and there is no information in the manuscript that the authors have confirmed species identities and have deposited pressed plant samples with flowers and fruits for future reference.

The presence of CAM typically results in nocturnal net CO₂ uptake and nocturnal increases in tissue acidity, i.e., of malic acid. As expected, *Clusia rosea* shows nocturnal net CO₂ uptake in Fig. 4c but rather unexpectedly doesn't show nocturnal net CO₂ uptake in Supplemental Figure 13c (?). *C. multiflora* doesn't show nocturnal net CO₂ uptake in both Fig. 4f and Supplemental Figure 13a suggesting that net CO₂ uptake is entirely via the C₃ pathway. Thus, evidence of presence or absence of CAM in *C. multiflora* relies almost entirely on measurements of malic acid levels. These were conducted between 4 am and 7 pm, but not over a full day-night cycle. Indeed, Fig. 5e demonstrates a decline in malic acid level from 1 pm to 7 pm, but it is not demonstrated that malic acid levels return to the initial 4 am/8am level during the following night. It is important to demonstrate this, because factors other than CAM can lead to a decline in leaf acidity during daytime. Similarly, declines in malic acid levels (with large error bars) are shown in Supplemental Fig. 12g, but again no data are presented that show that malic acid levels increased during the following night.

In some Figures the y axis extends to negative values for malic acid, probably to accommodate the large error bars, but also for PAR (Supplemental Fig. 13) which is not possible.

In summary, there seems to be a possibility of weakly expressed CAM in *C. multiflora*, but the manuscript does not provide convincing evidence.

End of message

Reviewer #3 (Remarks to the Author):

The authors addressed my comments/suggestions in the revision.

Reviewer #4 (Remarks to the Author):

The work is a notable genomic study on the genomic rearrangements underwent by several species of the *Clusia* genus. The authors have described how these species in fact slid back on the C3-CAM continuum providing both genomic and physiological evidence, and in the revised version of the manuscript, a plausible evolutionary explanation due to their current ecological niche.

The work is quite relevant as it in fact provides a more complex view of the C3-CAM evolution. Among others, the authors mention an instance that I would call C3-CAM-pseudoC3. This research showcases that CAM as every adaptation is not a culprit of evolution both rather an evolutionary state that can move further on to other adaptations depending on selective pressures and availability of ecological niches.

Quite an inspiring work!

REVIEWER COMMENTS

Reviewer #1 (Remarks to the Author):

The abstract advances an exciting claim: “Through a combination of phased chromosome-level assembly and annotation, comparative multi-omics, and physiological experiments, we demonstrate that diploidization of polyploids explains the physiotype diversity of CAM.” However, the manuscript does not yet provide a causal or even preferential link between diploidization and CAM. Whole-genome duplication followed by fractionation is the rule for angiosperms, so the burden is to show that CAM genes experienced a non-random fate relative to the genomic background and that those fates are tied to CAM phenotypes. At present, the results largely document fractionation/pseudogenization within CAM-related loci, but they do not test whether these patterns exceed genome-wide expectations or predict CAM function.

How does this compare to other paleotetraploids in terms of the fractionization of these CAM genes (even in non-CAM genomes since the GO terms in Supplemental Figure 5 are very similar to what is found in other genomes). I suggested in the last round to look at classic paleotetraploids like maize and soy (but there are many with great data). Or the authors could compare a host of pathways to see if the CAM pathway is preferentially impacted during the diploidization.

We thank the Reviewer for this thoughtful comment and for reiterating an important conceptual point regarding genome-wide diploidization dynamics. We appreciate the opportunity to clarify both the scope of our study and the logic underlying our interpretation.

The Reviewer raises the concern that, because whole-genome duplication followed by fractionation is widespread among angiosperms, a causal or preferential link between diploidization and CAM evolution would require demonstrating that CAM-related genes experienced a non-random fate relative to the genomic background or to other paleopolyploid species. While we agree that such comparative analyses would be highly informative, we respectfully note that they go beyond the central aim and scale of the present study.

Our primary objective is not to claim that diploidization uniquely targets CAM genes to the exclusion of other pathways, nor that CAM evolution required exceptional genome-wide behavior. Rather, our goal is to demonstrate that key CAM-associated genes did undergo fractionation and pseudogenization during diploidization, and that these changes are plausibly linked to observed CAM physiotype diversity. From a functional and evolutionary perspective, it is not necessary to show that CAM genes behaved differently from all other genes in the genome; it is sufficient to establish that diploidization affected CAM-relevant loci in ways that could influence CAM metabolism and regulation. Even if similar processes also impacted other pathways, this does not diminish their potential relevance for CAM evolution.

Importantly, we do provide evidence that not all CAM genes were equally affected. In Figure 4c, we show that specific, functionally central CAM-associated genes—most notably PWD and PGMP—exhibit stronger signatures of transposon insertions than the genomic background (z-scores of up to 15). These genes are focal to our physiological and experimental analyses, which directly demonstrate their contribution to CAM-related traits.

Regarding broader comparisons to other paleotetraploid genomes (e.g., maize or soybean) or pathway-wide enrichment analyses, we fully agree that such work would be valuable. However, conducting equivalent chromosome-level, gene-by-gene diploidization analyses

across multiple species would require substantial additional computational and analytical investment and would substantially expand the scope of the manuscript. Our study instead prioritizes depth over breadth: a high-resolution genomic analysis of a single system, coupled with experimental validation of key genes, to support a mechanistic hypothesis linking diploidization to CAM physiotype diversification.

We appreciate the Reviewer's engagement with this point and agree that the previous wording suggested a preferential link and implied a causal demonstration rather than a hypothesis. In response, we have revised both abstract and the overall text to remove causal language. The sentence in the abstract now reads:

"Through a combination of chromosome-level assembly and annotation, comparative multiomics, and physiological phenotyping, we identify a strong association between diploidization of polyploids and the physiotype diversity of CAM."

We hope this clarification makes clear that our conclusions are now intentionally framed as a hypothesis grounded in genome-scale patterns and functional validation, rather than as a claim that CAM genes uniquely escape general rules of post-WGD genome evolution. Ideally, this encourages others to discover remnants of the proposed mechanism within their CAM/CCM lineages, as diploidization appears to be underrepresented in plant sciences.

We are grateful that the reviewer raised these genomic comparisons, as we hope to pursue research in this direction in future work, possibly in collaboration with other groups.

A second issue is assembly terminology and phasing. The current reference is a haploid/pseudo-haplotype assembly, yet the text calls it "phased." Phasing, in the strict sense, requires haplotype-resolved sequences and metrics (e.g., switch-error rate). With a single collapsed haplotype, haplotype collapse or mixing could mislead subgenome and fractionation inferences. The rebuttal introduces a CANU build and "manual curation" as validation, but the procedures and criteria are not described, and the statement that CAM pathway genes are "accurate" is undefined.

In short, the study compellingly catalogs CAM-gene fractionation but does not yet show that diploidization specifically shaped CAM beyond background WGD dynamics, and the assembly status needs to be aligned with current standards or terminology.

We thank the reviewer for highlighting the ambiguity in assembly terminology and for requesting clarification on phasing, validation, and manual curation. We agree that the term "phased" was used incorrectly. The reference assembly represents a haploid (pseudo-haplotype) chromosome-level assembly rather than a fully phased (haplotype-resolved) genome. All occurrences of "phased assembly" have therefore been removed, and the text has been amended to consistently use appropriate terminology.

To investigate haplotype confusion and validate the integrity of identified pseudogenes, we used to screen for CAM-relevant regions solely in the genome browser. The assessment included the manual inspection of spiking/dipping read coverages and counts/qualities of alignments to the pseudo-haplotypes as well as to an independent CANU assembly. The latter in particular provides a solid, consistent indication of erroneous mixing, collapsing, or switching between haplotypes, especially when run with polyploid-aware parameters to minimize haplotype collapse (see Supplementary Code: Link).

To better address the reviewers concern regarding curation of pseudogenes due to FALCON's known issues of mis-assembly and haplotype confusion, we now implemented a new procedure with defined criteria and explicitly described the validation strategy in the

Methods (lines 561, 612-616, 667) and Supplementary Materials (see GitHub commit: Link). Briefly, pseudogenic loci lacking any alignment or exhibiting large deletions were excluded from downstream analyses. A revised main Fig. 3c highlights 739 identified false positives as a result of haplotype confusion, 6.6% of total pseudogenes. This led to the removal of three CAM-related pseudogenized genes and corresponding updates to figures and tables.

Changelog:

- Fig. 3c: New category for haplotype confusion, mentioned in the legend
- Fig. 4a & Suppl. Fig. 7: 3 CAM-related pseudogenes were removed
- Suppl. Table 4: List of pseudogenes now includes indel/filtering columns
- Suppl. Table 5: Signals of genic diploidization removed for filtered pseudogenes

While we do not claim a fully haplotype-resolved assembly, the revised terminology, explicit validation criteria, and conservative filtering ensure that the curated CAM gene set underlying our conclusions is robust.

Link to CANU config/parameters:

<https://github.com/hanneskramml/Clusia/blob/Manuscript1/assembly/canu/spec.hdusage.txt>

Link to GitHub commit:

<https://github.com/hanneskramml/Clusia/commit/4aa5fb94a607080761cd0dd936ed66c963ca9291>

Reviewer #2 (Remarks to the Author):

Clusia genomes shed light on the evolution and diversity of CAM physiotypes 3rd review

I was reviewer 2 in the first 2 rounds of peer review

Major comments –

I am not going to give responses to your responses to my responses, because this would be hard to read. Instead, I outline the 5 major points that were brought up in the first two rounds of peer review and give my responses below.

1. This was addressed in the last round of reviews

perfect

2. Fantastic! It is really great to have included a taxonomist in the analysis, and to have a record of what you have used for your genome sequencing – this will save a lot of headaches for future scientists who opt to use your genome as a resource. I am particularly impressed that you have deposited herbarium specimens and I agree that you have established “a new framework amid persistent taxonomic uncertainty”. Whilst you no longer have a C3 comparator in your study, I personally believe that this extra care and consideration in taxonomic identification actually adds more novelty than is lost from the lack of a fully C3 species. I know that there is some debate amongst CAM researchers as to the precise species identification of the first Kalanchoe genome that was assembled, so your study can now act as a baseline upon which future work can be based, to avoid such issues.

Thank you for the kind words

3. Please accept my apologies if I have been pedantic with regards to your use of English – this was not my goal in these comments – the manuscript is extremely well written. I feel that the updated manuscript, for the most part, does not overstate the link between WGD and CAM evolution in the problematic way that the last versions did. If I understand correctly, all of your species are ‘functional diploids’ and you are hypothesising that the WGD may have contributed to the evolutionary lability within Clusia, rather than genome size being directly proportional to the ability to evolve CAM? If this is what you are saying, then I can get behind this as a hypothesis. I would say that this could be spelled out a bit more clearly, as I am not entirely sure that that I have interpreted this correctly. May I suggest that you use the PhD thesis by B. Zambrano as a comparison in the paragraph starting line 414. Because this thesis shows that genome size does not correlate with CAM strength (Even if there are taxonomic misidentifications (very possible) in this thesis, this actually doesn’t make any difference for this point, because the author has generated an internally consistent estimate of the strength of CAM for these 9 species – so even if they are not the species that they are reported to be, the analysis still holds). But seeing as you have shown that all of your study species are functional diploids then it would follow that if WGD did promote the evolution altered CAM states, this would not necessarily result in a correlation between genome size and CAM strength. I think that if this is the hypothesis that you are trying to present, then it would help to make this a bit more explicit.

We thank the Reviewer for this clarification and for articulating the hypothesis so clearly. This interpretation is indeed exactly what we intend to convey.

All species examined in our study are functional diploids, and we do not propose a direct relationship between extant genome size or ploidy level and CAM strength. Rather, we hypothesize that ancient whole-genome duplication, followed by lineage-specific diploidization, may have increased evolutionary lability within Clusia, thereby facilitating diversification of CAM phenotypes without necessitating a present-day correlation between genome size and CAM expression.

We agree that this point benefits from being stated more explicitly. Accordingly, we have revised the paragraph beginning at line 414 (now 420) to clarify this conceptual framework and to distinguish it from genome-size-based interpretations of CAM evolution. As suggested, we now cite the PhD thesis of B. Zambrano here, which demonstrates a lack of correlation between genome size and CAM strength across multiple taxa. This observation is fully consistent with our hypothesis, given that all taxa in our study have undergone diploidization.

We believe these revisions make our intended interpretation clearer and align the manuscript more closely with the reviewer's understanding.

That being said, I still do feel that the abstract of your manuscript is overstating the role of WGD in the evolution of CAM. For example, in your current abstract, you say “we demonstrate that diploidization of polyploids explains the phenotype diversity of CAM” – but really you are proposing a hypothesis, rather than a demonstration.

We agree with the Reviewer that the previous wording in the abstract overstated the role of WGD by implying a causal demonstration rather than a hypothesis. In response, we have revised both abstract and the overall text to remove causal language. The sentence in the abstract now reads:

“Through a combination of chromosome-level assembly and annotation, comparative multiomics, and physiological phenotyping, we identify a strong association between diploidization of polyploids and the phenotype diversity of CAM.”

We believe these revised wordings more accurately reflect the nature of our findings and frame the role of WGD as an association consistent with a hypothesis, rather than a demonstrated causal mechanism.

Also I think that line 422 should be changed to ‘we hypothesize that...’ and ‘may have played a crucial role’ – to make this clear that it is a hypothesis.

Done.

In a related point, I think that your use of “evolutionary intermediates” (line 45) is problematic, because it implies that weak and/or facultative CAM species are on an evolutionary trajectory towards becoming strong CAM. But this might not be the case and it is also equally parsimonious (based on only 3 species) that a strong CAM phenotype has evolved into a weaker CAM phenotype. I think that you should not implicitly assume that there is a particular direction to the evolution of CAM, because you don't have data to support this.

We agree that the term “evolutionary intermediates” implies a directional evolutionary trajectory that we do not intend to assume. To avoid this implication, we have revised the

abstract to replace this term with language that emphasizes “phenotypic phases” without implying evolutionary direction, as already introduced for this particular case in ref.⁹⁰.

4. I appreciate the time that it would take to recreate the greenhouse experiment under controlled conditions, as it would be prohibitively time consuming.

My initial issue stemmed primarily from the fact that you had drawn conclusions about the photosynthetic phenotypes of these species based on a highly variable experimental design, which made it difficult to differentiate G from E when drawing conclusions. I much prefer the way you have framed the data now, with Fig. 1 as the controlled phenotyping and figs 5 and 6 as examples of how you could use your genome sequences to understand the physiology of these species under field-like conditions (i.e. where there will be more variation). I am struck by what you say in the last round of reviewer responses – “In fields where molecular biology and ecology intersect, some degree of environmental variability is expected and, in our view, acceptable” – and I totally agree with you. In fact, I think that once the phenotyping has been done (under controlled conditions) this is a great example of how your genome can provide insights into what is going on under more realistic conditions. That being said, I still think that the manuscript would be improved from more data in the initial phenotyping, under controlled conditions. At the moment, the only line of evidence that *C. major* is C3/CAM is that there is no detectable net CO₂ efflux at night in your gas exchange (a point that you don't seem to make – and is absent from the legend to Fig. 1 and elsewhere). But seeing as the nocturnal CO₂ assimilation rate is so low in Fig. 1B, it is hard to know that this falls within the precision of your instruments. If you were to measure titratable acidity, in all three species, grown under controlled conditions at dawn and dusk, under well watered and also droughted conditions, with the same light regime, you would be able to provide a second line of evidence that would help to confirm the photosynthetic physiology of the species you use in your study. This would then mean that the second physiology experiment could be used not to determine species genetic capacity for CAM, but how they are behaving under close-to-real world conditions. If you had this structure to the manuscript: i.e. 1) Phenotype the species under controlled conditions; 2) Identify the species ID using taxonomic techniques, 3) sequence the genome and assess WGD; and 4) use the genome in a more variable, close-to-real-world experiment to give direct evidence of the utility of the genome – this would be a really beautiful paper. As it stands it is just short of that, and the ambiguity of what *C. major* is doing leaves the reader with a sense of uncertainty in how to interpret the results presented in Fig 5 and 6, due to the combination of treatments employed in this experiment (i.e. drought and high light are always together).

Whilst I recognise that repeating the gas exchange for all of these plants would take too long, a titratable acidity experiment would be much more manageable, and in my opinion is the missing part to confirm that *C. major* is doing what you say it is. As you have pointed out in your manuscript, the prevalent misidentification of *Clusia* species in the literature makes it difficult to know that any previously published paper is definitely talking about the same species as you have. For this reason, it is integral that you have internally consistent phenotyping data conducted with a robust, highly controlled experimental design. Please note if you do this, that it is important to leave the plants under the controlled conditions to acclimate to those conditions for at least two weeks.

We thank the Reviewer for this detailed and constructive comment and fully agree that robust, internally consistent phenotyping under controlled conditions is essential, particularly given the prevalence of misidentification in *Clusia*.

In response, we have conducted the suggested titratable acidity (TA) experiment under highly controlled conditions for *C. major*, *C. minor* s.l. and *C. rosea*. Nine biological replicates were acclimated for six weeks under identical environmental conditions, and TA was measured three times over a full 24-hour cycle (five time points) under both well-watered and drought conditions two weeks later, without changes to the light regime nor microclimatic parameters (Supplemental Fig. 1). These data are now presented as a new panel for nocturnal acid fluctuations (Fig. 1c), with full TA time-course data for all species provided in Supplemental Fig. 1.

Consistent with the gas exchange measurements, *C. rosea* shows strong nocturnal acidification under both watering regimes ($\Delta TA \approx 100 \text{ mmol H}^+ \text{ g}^{-1} \text{ FW}$). In contrast, *C. major* shows no nocturnal acidification under well-watered conditions ($\Delta TA \approx 0$), but exhibits significant nocturnal acid accumulation under drought ($\Delta TA > 60 \text{ mmol H}^+ \text{ g}^{-1} \text{ FW}$), confirming CAM induction. Importantly, this TA assay was performed under the same controlled conditions as the gas exchange measurements shown in Fig. 1b.

We believe that the inclusion of this second, independent line of evidence resolves the ambiguity surrounding the photosynthetic physiotype of *C. major* and strengthens the interpretation of subsequent experiments (Figs. 5 and 6), which are now explicitly framed as genome-informed analyses of physiological behavior under more variable, field-like conditions.

5. Most of what I wanted to say about this is covered in my response to point 4. But I will just add that the subtitle “CAM-like gene expression and protein activity patterns are retained in the C3-type mode of photosynthesis” line 284 does not really hold up because you haven’t done sufficient phenotyping under controlled conditions – i.e. see my point above. The gas exchange trace gives some suggestion of weak CAM because there is no nocturnal respirational efflux of CO₂, but this is not robust enough to know for sure what is going on. For this reason you cannot determine that these plants are truly exhibiting a “C3-type mode of photosynthesis” or if they are constitutive weak CAM under well watered conditions. Considering that there is already considerable debate about the degree to which low level acid fluctuations occur in C3 plants (see <https://doi.org/10.1111/nph.17790>) it is tricky to be calling something “C3-type” when it might be doing weak CAM. I would suggest changing this subtitle to something like “RNA-seq and proteomics to explore the nature of CAM under close to real-world conditions”, and also adding the titratable acidity experiment I outlined above to Fig. 1.

We have revised the subtitle to remove any implication of photosynthetic classification and to focus instead on “CAM-associated molecular and metabolic signatures under close-to-real-world conditions”. This revised wording more accurately reflects the scope of the data and avoids assumptions about physiological mode.

Minor comments:

Line 78 – this statement is unsubstantiated and I don’t think it is correct. Whilst it is more than possible that ‘most’ and maybe even ‘all’ species of *Clusia* have some capacity of CAM, the data that currently exists in the literature has not shown this to be the case. For example isotopic analyses find many species that do not show CAM signatures (doi: 10.32615/ps.2022.018). Also a meta-analysis on photosynthetic modes in *Clusia* found that several species exhibited no upregulation of CAM even when drought stressed

(<https://doi.org/10.1071/FP20268>). I think that as it stands, there is no robust way to quantify the extent to which CAM (especially weak CAM) occurs across the genus. I would advise changing this opening statement to something more like “Extensive research throughout the 20th century has established that many *Clusia* species possess an inherent genetic capacity for CAM”.

done

Line 165 - Do you mean *C. major* here?

We do, the attention to detail is much appreciated

Line 288 – “In an ‘open greenhouse’ experiment, we exposed the plants to conditions mimicking 289 native habitats under canopy environments (shaded) as well as exposed and drought conditions, to see 290 how they cope with adverse conditions” – this makes it seem as though the shade and drought treatments were applied separately, whereas in reality the high-light plants were always experiencing drought – this should be made more clear in this sentence.

The description has been changed and now states: “In an ‘open greenhouse’ experiment, we exposed the plants to conditions mimicking native habitats under a) canopy environments (shaded+well-watered) as well as b) adverse conditions (high-light+drought), to see how they cope with variable environmental conditions”. This way it should be clear that there are only two experimental conditions and that high light and drought were always combined.

Line 438 – are you specifically referring to *Clusia* in this statement – if so you should make this clear

We amended the paragraph to reflect more precisely where we are referring to *Clusia* and where we refer to more generalizable principles (lines 447-452).

Fig. 1 – Can you really say that “*C. major* exhibits a clear C₃-type of photosynthesis” when there is no nocturnal CO₂ efflux? Does this not suggest a weak CAM phenotype where nocturnally respired CO₂ is recycled via the CAM cycle?

We agree with the reviewer that the absence of nocturnal CO₂ efflux alone argues against a strictly C₃ classification. While this point was discussed in the previous Supplementary File, we did not emphasize it in the main text at the time. However, in light of the combined gas-exchange data and the newly added titratable acidity (TA) measurements and overall development of the story, we fully agree that *C. major* cannot be described as exhibiting a clear C₃-type photosynthesis. We have therefore revised the figure legend accordingly as well as the main text to explicitly note the lack of nocturnal carbon efflux.

REVIEWERS' COMMENTS

Reviewer #2 (Remarks to the Author):

Peer review for *Clusia* Genome paper, March 2026

I was reviewer 2 in previous rounds of peer review. The editor has asked me to also assess whether the responses to reviewer 1's comments are satisfactory. Reviewer 1 wrote 2 paragraphs in the last round. Their first paragraph concerns the interpretation of WGD, and I have added comments to this in my 'major comments'. For the second paragraph of comments from reviewer 1, I do not have sufficient expertise to make a judgement of whether you have satisfied this comment.

Overall, I really enjoyed re-reading the manuscript, and feel that it is a significant contribution to the field.

Major comments:

A major difference in the manuscript is the addition of titratable acidity measurements to phenotype the species under controlled conditions. I thank the authors for adding these data, as I believe that they make a huge difference to the story. The manuscript no longer relies on data from the greenhouse experiment, where the environmental conditions are inherently noisy, to phenotype the three species.

I really appreciate that sufficient time was given to ensure that the plants were truly in a well-watered condition before the experiment began. My main criticism of these new data is that you have made the claim that $n=9$, when in reality $n = 3$. The use of multiple technical replicates from each biological rep definitely strengthens your data, but if you use $n = 9$, then this is pseudoreplication. Likewise, with the p values that you show – are these based on treating each of the 9 points as an individual replicate? Because if so then this needs to be changed.

We thank the reviewer for pointing this out. We have revised the plots to display only the three true biological replicates, after averaging the technical replicates for each sample. The figure legend now correctly indicates $n = 3$, and the statistical tests were recalculated using these three biological replicates only. The resulting p-values remain unchanged in their interpretation, and the biological conclusions are therefore unaffected.

In my last round of review, I mentioned that your writing, at times, depicts a directionality to the evolution of CAM – that the C3-CAM phenotypes are evolutionary intermediates between C3 and strong CAM. Whilst this might be the case, there is insufficient information to confirm this, as you are only looking at 3 species. It is also completely possible that *C. minor* and *C. major* evolved their 'intermediate' phenotypes from a strong CAM ancestor, rather than from a C3 ancestor. Because it is not possible to resolve this issue, with your data, more care is needed in your writing. Since the last review, you changed the phrasing in the abstract to say 'phenotypic phases'. However, I think you have not really dealt with the issue, but merely changed the phrasing to a different (more confusing) term. My point in the last review is that you cannot assume that the direction of evolution in your 3 species went C3 -> weak CAM -> strong CAM; because you don't have a strong enough phylogeny to resolve this. So you need to carefully reframe arguments that implicitly assume this direction. For example, line 66/67 you say "investigating intermediate phenotypes of CAM in close relatives can help gather snapshots of different evolutionary stages". But you don't know the direction that the phenotypes have evolved, so how can you conclude anything about 'evolutionary stages'? Also, line 73 you write "Our findings modernize our understanding of the "stepwise" evolution of CAM phenotypic diversity, the role of polyploidy in its, perhaps recurrent, origin..." - again, you don't know whether *C. minor* and *C. major* evolved their phenotypes from a C3 or a strong CAM ancestor, so you really cannot make any conclusions about the 'stepwise' nature of CAM evolution.

That being said, I do appreciate that you have engaged with this more directly later on (line 447). The issues are much more pronounced in the introduction and earlier parts of the manuscript.

We appreciate the reviewer's comments. While we believe that these passages primarily introduce the overall topic and do not draw explicit conclusions, we agree that this type of framing may nevertheless pave the way for unwarranted interpretations later in the manuscript. We therefore revised both sections mentioned and removed the terms "intermediate" and "stepwise". The former was amended to read: "...investigating different physiotypes of CAM in close relatives can help gather snapshots of different evolutionary stages from C3 to CAM, but also vice versa."

On a related note, one of the previous criticism I have outlined (as has reviewer 1) was the claim that WGD was important for the evolution of CAM. I think both reviewer 1 and I pointed out that this claim would need to show that there was a higher rate of WGD than would be expected in *Clusia* and that this has led to the evolutionary lability within this genus. I appreciate that the results/discussion section now does outline the WGD idea much more as a hypothesis than as a proven theory, but I do think that you could make this even more explicit, just so this paper doesn't get cited incorrectly. My suggestion is that you include a paragraph from your response letter:

"Our primary objective is not to claim that diploidization uniquely targets CAM genes to the exclusion of other pathways, nor that CAM evolution required exceptional genome-wide behavior. Rather, our goal is to demonstrate that key CAM-associated genes did undergo fractionation and pseudogenization during diploidization, and that these changes are plausibly linked to observed CAM physiotype diversity. From a functional and evolutionary perspective, it is not necessary to show that CAM genes behaved differently from all other genes in the genome; it is sufficient to establish that diploidization affected CAM-relevant loci in ways that could influence CAM metabolism and regulation. Even if similar processes also impacted other pathways, this does not diminish their potential relevance for CAM evolution."

My advice is that you include this after line 459. This will make it clear to the reader exactly the extent of your hypothesis, and will help them understand the nuance of what you are claiming/exploring.

We appreciate this suggestion and agree that explicitly clarifying the scope of our interpretation will help avoid overgeneralization of the results. Following the reviewer's recommendation, we incorporated a statement reflecting this point into the Discussion. The added text reads:

"Our objective is not to claim that diploidization uniquely targets CAM genes to the exclusion of other pathways, nor that CAM evolution required exceptional genome-wide behavior. Rather, our goal is to demonstrate that key CAM-associated genes did undergo fractionation and pseudogenization during diploidization, and that these changes are plausibly linked to observed CAM physiotype diversity."

For thematic coherence, we placed this statement at the end of the section "Intronic transposon insertions and homoeolog fractionation affect CAM photosynthesis and starch metabolism in *C. major*." We felt this location was most appropriate because the preceding paragraphs directly describe the diploidization patterns affecting CAM-related loci, and the statement therefore helps clarify the scope and interpretation of these findings.

Minor comments:

Line 31 – this sentence needs to be rephrased – maybe replace 'evolved' with 'evolution'?

According to editorial demands the Abstract needs to be shortened to 150 words, therefore the line by line comments regarding the abstract became irrelevant, we would, however, appreciate if the new short abstract could also be reviewed by reviewer 2

Line 53 – reference needed here (for the sentence that ends on line 53) – to support the evolutionary relevance and ecological role of CAM in Clusia

done

Line 54 – ‘transpiration’ should be ‘diurnal transpiration’

done

Line 65 – I think ‘conversion’ should be replaced by ‘evolution’ here.

done

Line 65 – I am not really sure why you make specific reference to ‘single cell C4’ here, or C4 at all? My advice would be to remove the part in the parenthesis, as it doesn’t add anything to your introduction.

done

Line 66/67 – see major comments

done

Line 70 – Where you say “use our data to reconstruct the evolutionary origins of subtypes of CAM in this particularly physiologically plastic plant genus” - I don’t know what you mean.

Replaced “plastic” with “diverse”

Line 73 – see major comments

Line 79 – it seems odd to me that you make an effort to highlight U Luttge in the parenthesis and then ref 21 is not one of his papers.

changed

Line 79/80 – have you really ‘evaluated’ the claim that many species in Clusia do CAM? I think it would be more appropriate to say ‘To analyse this genus with contemporary molecular tools...’.

done

Line 85 – all weak CAM species (and all strong CAM species for that matter) do C3 photosynthesis, so it is not really correct to describe C. major as “consistently exhibiting C3-like behavior in 86 combination with weak CAM”. It is better to just describe it as a weak facultative CAM species.

done

Line 86 – “This selection aligned well with existing physiological data, especially when the notoriously challenging taxonomy of Clusia species is considered” – I don’t really know what this means.

We revised the section for clarity

Line 289 – as a plant physiologist (with limited genomics experience) reading this manuscript, I feel that it would be nice if you could qualify why you make this hypothesis. Is it based on the details that you have outlined before? Is there any quantitative evidence (i.e. comparison of diploidisation extent in starch degradation vs a basal background rate)? Or is it just a guess? I cannot tell what leads you to end this paragraph with this hypothesis, and would benefit from this being explained explicitly.

We amended the text. It now says: "We hypothesize that genetic variability in the phosphorolytic starch degradation pathway is related to physiological changes in plastidic starch metabolism in *C. major* and may be important for carbon fixation and stomatal behavior"

Line 396 – what do you mean by "Given that CAM originated much earlier in the two best-studied lineages" – are you referring to Crassulaceae and Portulacineae here? I think this sentence could be tighter, as I am not sure what you are referring to.

Changed to just say: "The question arises whether ~~its~~ CAM evolution in *Clusia* resembles that in Crassulaceae or Portulacineae where CAM evolved monophyletically from a single C₃-photosynthesis ancestor ..."